# BOATSv2: new ecological and economic features improve simulations of high seas catch and effort

**Jerome Guiet**[1], **Daniele Bianchi**[1], **Kim J. N. Scherrer**[2], **Ryan F. Heneghan**[3], **and Eric D. Galbraith**[4,5]

[1]Department of Atmospheric and Oceanic Sciences, University of California Los Angeles, Los Angeles, CA, United States
[2]Department of Biological Sciences, University of Bergen, 5020 Bergen, Norway
[3]School of Environment and Science, Griffith University, Nathan, Queensland, Australia
[4]Department of Earth and Planetary Science, McGill University, Montreal, QC, Canada
[5]Institut de Ciència i Tecnologia Ambientals (ICTA-UAB), Universitat Autònoma de Barcelona, 08193 Cerdanyola del Vallès, Barcelona, Spain TS1

**Correspondence:** Jerome Guiet (jerome.c.guiet@gmail.com)

**Abstract.** Climate change and industrial fishing are having profound effects on marine ecosystems. Numerical models of fish communities and their interaction with fishing can help assess the biogeochemical and socioeconomic dynamics of this coupled human–natural system and how it is changing. However, existing models have significant biases and do not include many processes known to be relevant. Here we describe an updated version of the BiOeconomic mArine Trophic Size-spectrum (BOATS) model for global fish and fishery studies. The model incorporates new ecological and economic features designed to ameliorate prior biases. Recent improvements include reduction of fish growth rates in iron-limited high-nutrient low-chlorophyll regions and the ability to simulate fishery management. Features added to BOATS here for the first time include (1) a separation of pelagic and demersal fish communities to provide an expanded representation of ecological diversity and (2) spatial CE1 variation of fishing costs and catchability for more realistic fishing effort dynamics. We also introduce a new set of observational diagnostics designed to evaluate the model beyond the boundary of large marine ecosystems (66 commonly adopted coastal ocean ecoregions). Following a multistep parameter selection procedure, the updated BOATSv2 model shows comparable performance to the original model in coastal ecosystems, accurately simulating catch, biomass, and fishing effort, and markedly improves the representation of fisheries in the high seas, correcting for excessive high seas and deep-sea catches in the previous version. Improvements mainly stem from separating pelagic and demersal energy pathways, complemented by spatially variable catchability of pelagic fish and depth- and distance-dependent fishing costs. The updated model code is available for simulating both historical and future scenarios.

## 1 Introduction

Recent developments have enabled the formulation of size-based fish community models based on fundamental ecological principles (Heneghan et al., 2021). Instead of resolving linkages between species or functional groups within marine food webs, aggregated size spectrum models are based on properties that emerge at higher levels of organization. These models rely on macroecological principles to connect individual growth and metabolism (Brown et al., 2004; Kooijman, 2010; Hatton et al., 2021) to community-level production and biomass (Gascuel et al., 2011; Blanchard et al., 2012; Maury and Poggiale, 2013; Jennings and Collingridge, 2015; Petrik et al., 2019; Heneghan et al., 2020). By simplifying complex ecosystem dynamics into community-level biodiversity (Maury, 2010; Petrik et al., 2019) and regional variations in trophic efficiency (Du Pontavice et al., 2020) and other ecological variables, these models can project the response of global marine ecosystems to warming and shifts in primary production due to climate change (Lotze et al., 2019; Tittensor et al., 2021).

The BiOeconomic mArine Trophic Size-spectrum (BOATS) model is a size spectrum model that incorporates an explicit representation of commercial fishing effort (Carozza et al., 2016, 2017). The model's integration of ecological and economic dynamics enables a clear illustration of the profound effects of advances in fishing technology on historical changes in fish biomass compared to the impacts of climate change (Galbraith et al., 2017; Carozza et al., 2019). The ability to simulate how fish catches respond to dynamic fishing effort allows optimization of the model's ecological parameters against observational reconstructions of fish catches (Carozza et al., 2017). Based on this optimization method, BOATS provided new estimates of the global biomass and cycling rate of fish, indicating a non-negligible impact of fishing on carbon sequestration and biogeochemical cycles (Bianchi et al., 2021; Le Mézo et al., 2022). It also offered mechanistic insights into the historical progression of fisheries (Guiet et al., 2020). While the model was originally designed under the assumption of open-access fishing effort, subsequent developments enabled it to investigate the effects of regulatory measures on fish community dynamics and their response to long-term and abrupt climate perturbations (Scherrer and Galbraith, 2020; Scherrer et al., 2020).

These studies prove the usefulness of BOATS for exploring various aspects of global fisheries. Still, comparisons with observations have also revealed discrepancies that suggest limitations in the model's parameterizations and missing mechanisms. For instance, high-nutrient low-chlorophyll (HNLC) regions are characterized by relatively low primary production despite available macronutrients (Moore et al., 2013). These regions represent more than one-quarter of the open-ocean surface area and include the Southern Ocean, the eastern equatorial Pacific, and the subarctic North Pacific. In HNLC regions, comparison of simulated effort with global reconstructions suggested excessive fishing activity in BOATS, indirectly pointing to excessive biomass accumulation in the model (Galbraith et al., 2019). Similarly, while the model provides a realistic representation of coastal fisheries, catches in the high seas (here defined as the regions of the ocean beyond large marine ecosystems) appear to be much larger than recent observational reconstructions. Note that large marine ecosystems (LMEs) are 66 coastal ocean regions defined by ecological criteria (Sherman and Duda, 1999). Specifically, BOATS simulates 40 % of global catches beyond the boundary of LMEs by the 1990s. This is approximately 4 times larger than the value of 8 %–9 % from empirical estimates (Watson, 2017; Pauly et al., 2020). The large high seas catches coincide with excessive fishing in waters found above the deep seafloor. During the 1990s, the model's catch-weighted mean depth of waters where fishing occurs is 1698 m, contrasting significantly with the observational reconstruction range of 154–372 m. This discrepancy limits the model's applicability to study the interaction of industrial fishing with high seas and deep-ocean ecosystems

and suggests potential shortcomings in the representation of open-ocean food webs.

In parallel, recent studies have shed new light on large-scale aspects of global marine ecosystems and fisheries. Reconstructions of industrial fishing effort by Global Fishing Watch (GFW; Kroodsma et al., 2018) highlighted spatial variations in fishing costs (Sala et al., 2018) and revealed the importance of seamounts in concentrating fishing activity, especially for pelagic fisheries in the high seas (Kerry et al., 2022). New reconstructions of fishing effort that include artisanal and industrial sectors provide more nuanced insights on the development of regional fisheries (Rousseau et al., 2019, 2024). Regional catch reconstructions have revealed the importance of "straddling" species, which forage both within exclusive economic zones and in the high seas over the course of a year, thus disconnecting regions of fish biomass production from regions of biomass extraction (Sumaila et al., 2015). Analysis of catch data shows how energy inputs at the base of food webs determine the dominance of pelagic vs. demersal communities across latitudes (van Denderen et al., 2018), suggesting different temperature sensitivities of growth for these groups (van Denderen et al., 2020). Finally, harmonization and in-depth analysis of fishery-independent trawl data have begun to reveal large-scale fish biomass patterns with unprecedented accuracy (Maureaud et al., 2024).

Here, motivated by discrepancies between observations and simulations with the original BOATS model (BOATSv1) and insights from recent large-scale studies, we revise the model formulation to improve its representation of high seas vs. coastal fisheries and of pelagic vs. demersal communities, leading to a significant model update: BOATSv2. The rest of the paper consists of four main sections. Section 2 summarizes the main principles and formulation of BOATSv1. Section 3 details previous model developments and new features introduced in BOATSv2. Section 4 describes a revised model optimization procedure. Section 5 justifies the selection of an ensemble of five optimal parameters, compares the old and new model versions (highlighting improvements in the representation of global fisheries in BOATSv2), and discusses insights from the new formulation.

## 2 BOATSv1

The philosophy of BOATS is to ensure global applicability while including sufficient ecological and economic complexity to represent realistic first-order fishery dynamics. The model is designed to include a relatively small number of parameters and to be computationally efficient, facilitating objective parameter optimization. It uses vertically averaged habitat characteristics on a two-dimensional spatial grid to simulate the variability of fish communities, from small regions to the global ocean. In the following, we provide a brief overview of key model principles. We refer the interested

reader to Appendix A for all equations and to previous publications for detailed explanations (Carozza et al., 2016, 2017). A schematic that illustrates the model principles (adapted for BOATSv2 from Carozza et al., 2017) is shown in Fig. 1.

## 2.1 Ecological module

BOATSv1 simulates the evolution in time ($t$) of the biomass of commercial fish as a function of size, $f_k$ (in $\mathrm{g\,m^{-2}\,g^{-1}}$, where g represents grams of wet biomass; see also Table 1 for a list of variables and parameters), and its propagation along a spectrum of size classes $[m_0, m_\mathrm{u}]$, where $m$ is the biomass of an individual fish (in g), and $m_0$ and $m_\mathrm{u}$ are the minimum and maximum fish sizes represented by the model. To include a coarse representation of species diversity, BOATS simulates $n_k$ different fish "groups" distinguished by their asymptotic mass $m_{\infty,k} < m_\mathrm{u}$, labeled by the subscript $k$ (see an illustration of the small and large groups in the central panel of Fig. 1). The total biomass density, $B_k = \int_{m_0}^{m_{\infty,k}} f_k \, \mathrm{d}m$, is the sum of each group's biomass across individual size classes (in $\mathrm{g\,m^{-2}}$). The propagation of biomass in each size spectrum as a function of time $t$ is described by the McKendrick von Foerster equation:

$$\frac{\partial}{\partial t} f_k = -\frac{\partial}{\partial m} \gamma_{S,k} f_k + \frac{\gamma_{S,k} f_k}{m} - \Lambda_k f_k - h_k. \qquad (1)$$

The first term in Eq. (1) represents the rate of change in time of the fish biomass spectrum for each group. The second term is the divergence of the growth flux, i.e., the transfer of biomass to increasing size as fish grow. The third term encapsulates the biomass accumulation due to the increase in individual size as fish grow. The fourth and fifth terms represent losses from natural mortality and catch, respectively.

This first-order partial differential equation in time and size requires both a boundary condition, here prescribed at the smallest size class $m_0$ and representing recruitment, and an initial condition at $t = 0$, representing the initial biomass distribution for each group:

$$\begin{cases} \gamma_{S,k} f_k = R_{\mathrm{P},k} \dfrac{R_{\mathrm{e},k}}{R_{\mathrm{P},k} + R_{\mathrm{e},k}} & \text{for } m = m_0 \quad \text{(boundary condition)} \\ f_k = f_{k,m,0} & \text{at } t = 0 \quad \text{(initial condition).} \end{cases} \qquad (2)$$

In Eqs. (1)–(2) the size-dependent growth rate $\gamma_{S,k}$ (in $\mathrm{g\,s^{-1}}$; Eq. 3) controls the biomass propagation through size (white arrows in the central panel of Fig. 1), influenced by local water temperature ($T$ in Kelvin) and primary production $\Pi_\psi$ (in $\mathrm{mmol\,C\,m^{-2}\,s^{-1}}$). The natural mortality rate $\Lambda_k$ (in $\mathrm{s^{-1}}$; Eq. 4) represents the biomass losses within each size class from predation and natural mortality (gray line in the central panel of Fig. 1). The sink term $h_k$ (in $\mathrm{g\,m^{-2}\,g^{-1}\,s^{-1}}$) is the biomass harvest by fishing that couples the ecological module to the fishery dynamics module (see Sect. 2.2). Finally, $R_{\mathrm{e},k}$ and $R_{\mathrm{P},k}$ in Eq. (5) are respectively the biomass input potential at the recruitment size $m_0$ from egg production ($R_\mathrm{e}$) by mature individuals (see yellow arrows in Fig. 1) and the maximum biomass input potential at the recruit size given

the primary production ($R_\mathrm{P}$). Both modulate the total recruitment input, $\gamma_{S,k} f_k$ (in $\mathrm{g\,m^{-2}\,s^{-1}}$). Simulations start from an initial biomass distribution $f_{k,m,0}$ that approximates an ocean in the absence of fishing ("pristine"), estimated from environmental conditions (Sect. 3.3).

In BOATS, the growth rate at a given size occurs either at the maximum physiological rate when food is not limiting (gray area in the central panel Fig. 1) or proportionally to primary production $\Pi_\psi$ when food is limiting (green area in the central panel Fig. 1). Accordingly, the growth rate is proportional to the minimum of two quantities: (1) the energy provided by primary producers that reaches a given size class $\xi_{\mathrm{P},k}$, given trophic transfer across the food web, divided by the number of fish in that size class, and (2) the maximum production potential for a fish of that size based on an individual-level allometric growth rate that follows a von Bertalanffy formulation $\xi_{\mathrm{VB},k}$ (in $\mathrm{g\,s^{-1}}$):

$$\gamma_{S,k} = (1 - \Phi_k)\, \xi_{I,k} = (1 - \Phi_k) \min\left(\xi_{\mathrm{P},k}, \xi_{\mathrm{VB},k}\right)$$
$$= (1 - \Phi_k) \min\left(\frac{\phi_{\mathrm{C},k} \pi m}{f_k}, \mathrm{Am}^b - k_\mathrm{a} m\right). \qquad (3)$$

Here, the term $(1 - \Phi_k)$ accounts for a reduction of the biomass allocated to somatic growth, with a fraction $\Phi_k$ allocated to the generation of reproductive material, i.e., egg production (Eq. 5). Thus, when food is limiting, individual fish will grow according to $\pi = \Pi_\psi / m_\psi (m/m_\psi)^{\tau-1}$, which defines a spectrum of available energy from primary production as a function of size (in $\mathrm{g\,m^{-2}\,g^{-1}\,s^{-1}}$). Here, $\tau$ is the trophic scaling, and $m_\psi$ is a representative cell size for primary producers (i.e., phytoplankton) at the base of the food web. The trophic scaling parameter determines the efficiency of the propagation of production through the consumer size spectrum to increasingly larger sizes and higher trophic levels, following the framework of the metabolic theory of ecology Brown et al. (2004). The representative size $m_\psi$ is determined from the empirical phytoplankton size structure model of Dunne et al. (2005) and depends on temperature ($T$ in °C) and primary production $\Pi_\psi$. To ensure coexisting fish groups and because of the scarcity of data available to constrain resource allocation, primary production is equally partitioned across the groups, i.e., $\phi_{\mathrm{C},k} = 1/n_k = 1/3$. While this is a first-order assumption that allows realistic simulation of catches by group, it should be revised as new observational constraints become available. When food is in excess of what can be consumed by the standing fish biomass, fish grow as fast as physiologically possible, given an allometric scaling $b$, a temperature-dependent anabolism $\mathrm{Am}^b = A_0\, a_\mathrm{A}(T) m^b$, and catabolism $k_\mathrm{a} m = A\epsilon_\mathrm{a}\, m_{\infty,k}^{b-1} m$, where $A_0$ is a growth constant (in $\mathrm{g\,s^{-1}}$) and $\epsilon_\mathrm{a}$ an activity fraction. This formulation is inspired by an empirical allometric framework following the model of Von Bertalanffy (1949), where growth is determined by food intake after assimilation and standard metabolism, discounted from energy used in activity and reproduction. This maximum growth is temperature-dependent

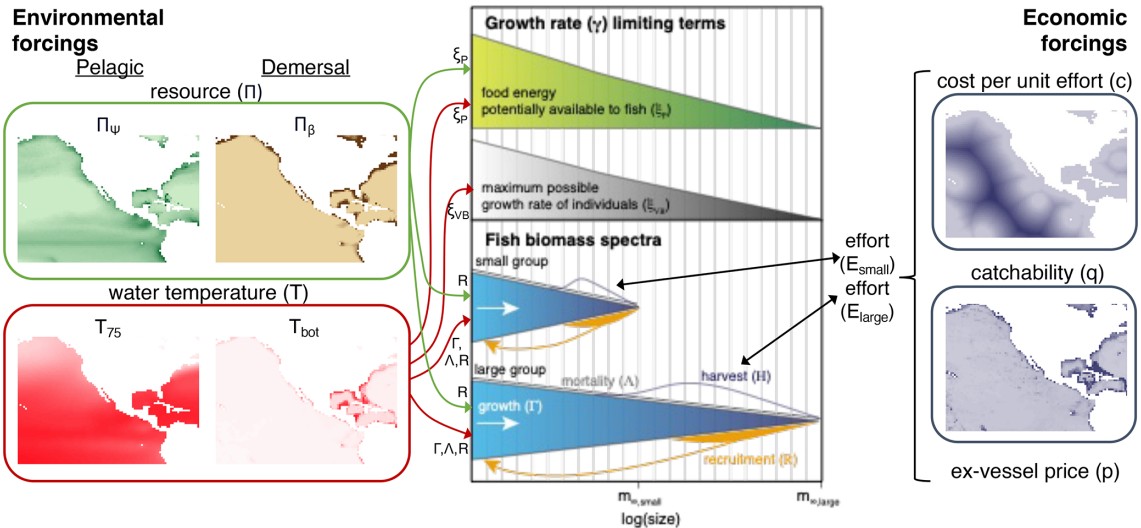

**Figure 1.** Schematic diagram of the main modules, components, and processes of BOATSv2. Environmental forcings, shown in the left panel ("pelagic" for BOATSv1; "pelagic" and "demersal" for BOATSv2), drive an ecological module that solves for the evolution in time of fish biomass as a function of fish size for multiple groups with different maximum size, shown in the central panel. These fish biomass spectra interact with the dynamic of fishing, controlled by an economic module and economic forcings, shown in the right panel. Economic forcings are spatially uniform in BOATSv1 but can be spatially variable in BOATSv2. Environmental forcings include the spatial distribution of resources at low trophic levels ($\Pi_\psi$ or $\Pi_\beta$) and representative habitat temperatures ($T_{75}$ or $T_{bot}$). Fish biomass spectra for multiple groups emerge from the balance of environmentally controlled growth ($\Gamma$, linked with $\xi_P$ or $\xi_{VB}$), recruitment ($R$), natural mortality ($\Lambda$), and fishing mortality ($H$). Economic forcings, which include spatially uniform ex-vessel prices ($p$) and spatially variable fishing costs ($c$) and catchability ($q$), influence the dynamic of fishing effort ($E$) for each fish group. Color shades of forcings illustrate spatial variations, from low (light) to high (dark) values. This figure is updated from the schematic for BOATSv1 in Carozza et al. (2017).

based on the factor $a_A(T)$ ($T$ in Kelvin), which follows a van't Hoff–Arrhenius curve controlled by a growth activation energy $\omega_{a,A}$ (in eV). The fish mortality is independent of variations in the growth rate.

The natural mortality rate (in units of $s^{-1}$) depends on both individual and asymptotic mass and represents biomass losses due to predation to organisms both within and outside of the resolved community size spectrum, as well as other natural causes. The natural mortality rate is based on an empirical parameterization (Gislason et al., 2010; Charnov et al., 2013):

$$\Lambda_k = e^{\zeta_1}\frac{A_0}{3}a_\lambda(T)m^{-h}m_{\infty,k}^{h+b-1},\tag{4}$$

where $h$ is an allometric scaling, and $\zeta_1$ (in $g\,s^{-1}$) a mortality rate parameter. As in Gislason et al. (2010), the natural mortality rate is linked to growth by means of the constants $A_0$ and $b$. To account for separate temperature dependencies between metabolism $a_A(T)$ and other processes such as predator–prey interactions, mortality varies with a distinct temperature dependence $a_\lambda(T)$, following a van't Hoff–Arrhenius curve controlled by a second activation energy $\omega_{a,\lambda}$ (in eV).

Recruitment provides the boundary condition, setting the flux of biomass at the lower mass boundary $m_0$. Recruitment is a function of the biomass production by mature individu-

als $R_{e,k}$ and a limit to the survival of recruits that depends on resource availability, proportional to primary production $R_{P,k}$:

$$\begin{cases} R_{e,k} = \phi_f s_e \frac{m_0}{m_e}\int_{m_0}^{m_{\infty,k}}\gamma_{R,k}(m)\frac{f_k(m)}{m}dm \\ R_{P,k} = \phi_{C,k}\pi(m_0)m_0. \end{cases}\tag{5}$$

Assuming that a fraction $\Phi_k$ of the input energy is allocated to reproduction, $\gamma_{R,k} = \Phi_k\xi_{I,k}$, the recruitment flux is determined by summing the contributions of all mature individuals across sizes $m$ for a fraction of females $\phi_f$, an egg survival probability $s_e$, and a mean egg mass $m_e$ (in g). Survival of recruits saturates towards a limit set by the energy available from primary production at the recruit size $m_0$. In high-biomass regions with large egg production rates, recruitment is thus generally constrained by $\pi(m_0)$.

With this formulation, for a given set of parameters (see the list in Table 1 and more details in Appendix A), the dynamics of commercial fish biomass in BOATSv1 are determined by two spatially and temporally varying environmental "forcings" (shown as "pelagic" forcings Fig. 1): local primary production $\Pi_\psi$ and epipelagic temperature $T = T_{75}$ (average temperature in the top 75 m in °C or K).

## 2.2 Economic module

BOATSv1 couples fish biomass and fishing effort $E_k$ (in $\text{W m}^{-2}$) to determine fish catch rates $C_k$ (in $\text{g m}^{-2}\,\text{s}^{-1}$) for each species group $k$ (see harvest in the central panel of Fig. 1). Fishing effort is typically initialized everywhere at negligible values, starting from an unfished ocean, and evolves independently in each grid cell under an open-access dynamic, proportional to the local net profit (the difference between revenue and cost) as

$$\begin{cases} \frac{\mathrm{d}}{\mathrm{d}t} E_k = \kappa_{\mathrm{e}} \frac{\text{revenue}_k - \text{cost}_k}{E_k} \\ E_k = 0 \text{ at } t = 0, \end{cases} \tag{6}$$

where $\kappa_{\mathrm{e}}$ (in $\text{W}^2\,\text{m}^{-2}\,\text{USD}^{-1}$) is a fleet dynamic parameter that sets the effort adjustment timescale for a given change in profit. This formulation assumes an absence of regulation so that fishers seek the greatest total catch in each grid cell. When profitable, revenues in Eq. (7) exceed costs in Eq. (9), and fishing develops continuously. In the presence of a continuous increase in catchability, this eventually leads to a peak in catch, overharvesting, and ultimately biomass collapse (see Carozza et al., 2017, and Galbraith et al., 2017, for details).

The rate of revenue for a time interval $\mathrm{d}t$ from a given location in the ocean (in $\text{USD m}^{-2}\,\text{s}^{-1}$) is determined as a spectrum $r_k$ (in $\text{USD m}^{-2}\,\text{s}^{-1}\,\text{g}^{-1}$) that is defined separately for each species group $f_k$. This represents the product of catch and the mass-specific price fishers are paid at port, integrated across size classes within each group:

$$\text{revenue}_k = \int_{m_0}^{m_{\infty,k}} r_k \mathrm{d}m \ \mathrm{d}t = p_k q_k E_k \mathrm{d}t \int_{m_0}^{m_{\infty,k}} \sigma_k f_k \mathrm{d}m, \tag{7}$$

where $p_k$ is the ex-vessel price (in $\text{USD g}^{-1}$) for each group. The catchability $q_k$ (in $\text{m}^2\,\text{W}^{-1}\,\text{s}^{-1}$) per unit of nominal fishing effort encapsulates the ability of fishing effort to extract fish biomass from the ocean. This quantity reflects the inherent characteristics of the fish group, as well as the fishing technology adopted (including gear, navigation instruments, sonars, and communication) and accrued knowledge (Palomares and Pauly, 2019). This formulation depends on the fraction of the fish biomass $\sigma_k f_k$ targeted by fishing, where $\sigma_k$ is a size-dependent selectivity of the fishing gear used to target group $k$. The selectivity plays a fundamental role by distributing fishing effort across size classes. A variety of functional forms exist, and all avoid the smallest sizes. These can be generalized as either dome-shaped (e.g., gill nets, longlines) or sigmoidal (e.g., trawls, seines, or dredges). Here, we parameterize the selectivity as a sigmoidal curve around a target threshold mass $m_{\Theta,k} = d_{m_{\Theta,k}} e_{m_{\Theta,k}} m_{\alpha,k}$, essentially reducing the fishing effort targeting the smallest size classes:

$$\sigma_k = \left[ 1 + \left( \frac{m}{m_{\Theta,k}} \right)^{-c_\sigma/3} \right]^{-1}, \tag{8}$$

with $c_\sigma$ a fishing selectivity slope. The target threshold mass is proportional to the maturity mass for each group $m_{\alpha,k}$, with the parameter $e_{m_{\Theta,k}}$ accounting for uncertainty around this mass and $d_{m_{\Theta,k}}$ set to select mainly mature individuals (i.e., $d_{m_{\Theta,k}} = 1$).

Net profits are determined by subtracting costs from revenues. Similar to revenue, the cost is expressed as the average expenditure rate per time over an area of the ocean (in $\text{USD m}^{-2}\,\text{s}^{-1}$). In reality, the cost of fishing includes the purchase and maintenance of capital, fuel costs for transit between fishing grounds and ports as well as during gear operation, and labor. In the model, cost is simply proportional to effort:

$$\text{cost}_k = c_k E_k \mathrm{d}t, \tag{9}$$

where $c_k$ is the cost per unit effort ($\text{USD W}^{-1}\,\text{s}^{-1}$).

When revenue exceeds costs, fishing effort in Eq. (6) increases. Any nonzero effort will lead to catches,

$$h_k \ \mathrm{d}t \ \mathrm{d}m = q_k \sigma_k \ E_k \ f_k \ \mathrm{d}t \ \mathrm{d}m, \tag{10}$$

which couple the economic and biological modules in Eq. (1). A catch limit is imposed for numerical stability (i.e., to prevent harvesting more fish than the biomass available in each grid cell). The total catch rate for each group is then given by $C_k = \int_{m_0}^{m_{\infty,k}} h_k \mathrm{d}m$ (in $\text{g m}^{-2}\,\text{s}^{-1}$). Note that when computing catches, but also costs and effort change, we set a lower limit on effort $\epsilon = 10^{-50}$ to allow the development of fishing and prevent division by zero in Eq. (6).

In BOATSv1, the ex-vessel fish price $p_k$ is generally assumed to be constant in space and time ($1.264 \times 10^{-3}\,\text{USD g}^{-1}$), since observations suggest small historical variations (Sumaila et al., 2007; Galbraith et al., 2017). Similarly, cost $c_k$ is also assumed to be constant ($1.852 \times 10^{-7}\,\text{USD W}^{-1}\,\text{s}^{-1}$). Catchability $q_k$ increases annually at a 5 % rate that accounts for sustained technological improvements and is the only temporally varying economic "forcing". Empirical studies have estimated an average annual rate of 2 %–8 % between fisheries and periods. We select an annual rate of 5 % increase as, after testing when other observed economic parameters are forced, it accurately reproduces the historical development of fisheries with BOATS (Galbraith et al., 2017; Scherrer and Galbraith, 2020). A list of economic parameters and quantities is provided in Table 1 TS2; additional details are provided in Appendix A.

## 3 BOATSv2

Here we describe the features of BOATSv2 that provide an update to the original BOATSv1. Two of these features were

**Table 1.** List of parameters and quantities of BOATSv2. For a full list and values see Appendix A.

| Parameter | Name | Units |
|---|---|---|
| $f_k$ and $B_k$ | Fish biomass spectrum and cumulative biomass of group $k$ | $(\text{g m}^{-2}\,\text{g}^{-1})$ and $(\text{g m}^{-2})$ |
| $m \in [m_0, m_\text{u}]$ | Biomass spectrum size range | (g) |
| $m_\text{e}$, $m_{\alpha,k}$, and $m_{\infty,k}$ | Egg, maturity, and asymptotic masses of group $k$ | (g) |
| $n_k$ and $\phi_{\text{C},k} = 1/n_k$ | Number of fish species size groups and fraction per group | Unitless |
| $\gamma_k$ | Size-dependent growth rate of group $k$ | $(\text{g s}^{-1})$ |
| $\Phi_k$ | Fraction of input energy allocated to growth of group $k$ | Unitless |
| $\xi_{p,k}$ or $\xi_{\text{VB},k}$ | Biomass input at individual level of group $k$ | $(\text{g s}^{-1})$ |
| $A_0$ | Growth constant | $(\text{g}^{1-b}\,\text{s}^{-1})$ |
| $\epsilon_\text{a}$ | Activity fraction | Unitless |
| $b$ | Growth scaling exponent | Unitless |
| $\pi$ | Fish production spectrum | $(\text{g m}^{-2}\,\text{g}^{-1}\,\text{s}^{-1})$ |
| $\tau = \ln(\alpha)/\ln(\beta)$ | Trophic scaling (trophic efficiency $\alpha$; predator–prey mass ratio $\beta$) | Unitless |
| $\Lambda_k$ | Natural mortality rate of group $k$ | $(\text{s}^{-1})$ |
| $\zeta_1$ | Mortality constant | Unitless |
| $h$ | Mortality scaling exponent | Unitless |
| $R_{\text{e},k}$ or $R_{\text{P},k}$ | Recruitment input of group $k$, from eggs or primary production | $(\text{g m}^{-2}\,\text{s}^{-1})$ |
| $\phi_\text{f}$ | Fraction of females | Unitless |
| $s_\text{e}$ | Eggs to recruit survival fraction | Unitless |
| $a_\text{A}(T)$ and $a_\lambda(T)$ | Growth and mortality van't Hoff–Arrhenius dependence | Unitless |
| $\omega_{\text{a,A}}$ and $\omega_{\text{a},\lambda}$ | Growth and mortality activation energies | (eV) |
| $h_k$ and $C_k$ | Fish catch spectrum and cumulative catch of group $k$ | $(\text{g m}^{-2}\,\text{g}^{-1}\,\text{s}^{-1})$ and $(\text{g m}^{-2}\,\text{s}^{-1})$ |
| $r_k$ and revenue$_k$ | Revenue spectrum and cumulative revenue of group $k$ | $(\text{USD m}^{-2}\,\text{g}^{-1}\,\text{s}^{-1})$ and $(\text{USD m}^{-2}\,\text{s}^{-1})$ |
| $c_k$ and cost$_k$ | Cost per unit effort and total cost of group $k$ | $(\text{USD W}^{-1}\,\text{s}^{-1})$ and $(\text{USD m}^{-2}\,\text{s}^{-1})$ |
| $E_k$ and $E_{\text{targ},k}$ | Fishing effort and effort target per group $k$ | $(\text{W m}^{-2})$ |
| $S$ | Effectiveness of regulation enforcement | Unitless |
| $\kappa_\text{e}$ and $\kappa_s$ | Fleet dynamics and regulation response parameters | $(\text{W USD}^{-1}\,\text{s}^{-1})$ and $(\text{m}^{-2}\,\text{USD}^{-1})$ |
| $p_k$ | Fish selling price of group $k$ | $(\text{USD g}^{-1})$ |
| $q_k$ | Fish catchability of group $k$ | $(\text{m}^2\,\text{W}^{-1}\,\text{s}^{-1})$ |
| $\sigma_k$, $c_\sigma$, and $m_{\Theta,k}$ | Fishing selectivity of group $k$, slope, and target threshold | Unitless and (g) |
| $d_{m_{\Theta,k}}$ and $e_{m_{\Theta,k}}$ | Parameter of the selectivity target threshold of group $k$ | Unitless |
| $T_{75}$ or $T_{\text{bot}}$ | Temperature, near surface or bottom | (°C) or (K) |
| $\Pi_\psi$ or $\Pi_\beta$ | Primary production, near surface or bottom | $(\text{mmolC m}^{-2}\,\text{s}^{-1})$ |
| $m_\psi$ or $m_\beta$ | Representative mass of primary producers or benthos | (g) |
| $\text{NO}_3^-$ and $k_{\text{NO}_3^-}$ | Surface nitrate concentration and constant controlling iron limitation | (µM) |
| pe$_{\text{ratio}}$ and $b_\text{a}$ | Export ratio and attenuation coefficient of particle flux | Unitless |
| $z_{\text{eu}}$ and $z_{\text{bot}}$ | Euphotic layer and seafloor depths | (m) |
| $d_{\text{coast}}$ | Distance to nearest coast | (km) |
| $x^*$ and $\delta$ | Parameters for cost profiles | (m or km) or $(\text{USD W}^{-1}\,\text{yr}^{-1}\,\text{m}^{-1}$ or $\text{km}^{-1})$ |
| $q_{\text{min}}$, or $x_{\text{max}}$ and $x_{\text{mean}}$ | Parameters for catchability profiles | Unitless or (m) |

introduced incrementally, in previously published work, in order to capture iron limitation in regions where iron is known to be scarce (Galbraith et al., 2019) and to represent management of fisheries (Scherrer and Galbraith, 2020; Scherrer et al., 2020). We provide a brief summary of these previous updates, before discussing the novel features added to the model in detail (see also Table 1 TS3 and Appendix A).

### 3.1 Previously published features

#### 3.1.1 Reduced growth rates in iron-limited regions

Iron limitation of phytoplankton growth is widely recognized in the ocean, most prominently in HNLC regions (Tagliabue et al., 2017). Less is known about iron limitation of higher

trophic levels in the ocean, including fish (Le Mézo and Galbraith, 2021). When satellite-based observational estimates of primary production are used as forcings, BOATSv1 overestimates fishing effort in HNLC regions, likely by simulating excessive biomass. Evidence of lack of adaptation by fish to low-iron regions suggests that low iron availability also significantly limits fish growth and could contribute to reducing fish abundance in large portions of the high seas (Galbraith et al., 2019).

Following Galbraith et al. (2019), we parameterize iron limitation of fish by reducing the trophic efficiency $\alpha$, which determines the fraction of biomass incorporated into new tis-

sues at each trophic step in HNLC regions:

$$\alpha^{v2} = \alpha \left( \frac{k_{NO_3^-}}{k_{NO_3^-} + NO_3^-} \right). \tag{11}$$

Here, the surface concentration of nitrate ($NO_3^-$, in µM) is taken as a proxy for iron limitation (Moore et al., 2013) and as an indicator of regions where fish are expected to be limited by the lack of iron, given the absence of other robust global estimates of surface iron concentrations or plankton iron contents. Note that here and in the following sections, the superscript "v2" indicates corrected quantities compared to the initial formulation in BOATSv1. This parameterization smoothly decreases the trophic efficiency as surface nitrate increases. The constant $k_{NO_3^-} = 5\,\mu M$ controls the strength of this effect and is constrained empirically. Nitrate concentrations are taken as the monthly minimum from the World Ocean Atlas (WOA) climatology (Locarnini et al., 2006) (see Appendix B). Although uncertainties persist regarding the impact of iron limitation on marine predators, this parameterization effectively reduces an overestimate in fishing effort in the Southern Ocean, North Pacific, equatorial Pacific, and, to some extent, North Atlantic in BOATSv1 (Galbraith et al., 2019).

### 3.1.2 Management with varying effectiveness

In BOATSv1, effort was generally assumed to follow an open-access dynamic equation (Eq. 6). This was modified to represent the influence of regulation by adjusting fishing effort to align with a prescribed target $E_{targ,k}$ (Scherrer and Galbraith, 2020) as

$$\left( \frac{d}{dt} E_k \right)^{v2} = \left( \frac{d}{dt} E_k \right) e^{-S} + \left( 1 - e^{-S} \right) \kappa_s \left( E_{targ,k} - E_k \right), \tag{12}$$

where $S$ is a non-dimensional parameter representing how effectively the target is enforced. When $S = 0$ the model follows open-access dynamics; when $S > 0$ the nominal effort is nudged towards the target at a rate proportional to the regulation response parameter $\kappa_s$ ($m^{-2}\,USD^{-1}$).

This feature showed that strong fishery regulation is required to prevent overfishing if technological progress keeps increasing, making management effectiveness a key factor in future scenarios (Scherrer and Galbraith, 2020). For the rest of the paper, we set $S = 0$ and focus on simulation of historical fisheries up to the time they reached maximum catches, for which the open-access dynamic was shown to be adequate (Galbraith et al., 2017; Guiet et al., 2020).

### 3.2 Newly added features

#### 3.2.1 Separate pelagic and demersal energy pathways

Variations in energy input at the base of marine food webs significantly affect biomass accumulation and cycling,

thereby altering the sensitivity of different fish communities to climate and environmental factors (Petrik et al., 2019). Pelagic communities are more tightly tied to surface planktonic production (i.e., net primary production $\Pi_\psi$), whereas benthic communities depend on the delivery of organic material to the seafloor (i.e., particle flux at bottom $\Pi_\beta$ in $mmol\,C\,m^{-2}\,s^{-1}$) (Stock et al., 2017; van Denderen et al., 2018). The two types of communities also experience different temperatures, with surface temperature (here, $T = T_{75}$) controlling the metabolic rates of pelagic fish and bottom temperature ($T = T_{bot}$) that of demersal fish.

To account for these ecological differences, we modified BOATS to resolve separate pelagic and demersal fish communities. Both communities are described by the same set of equations described above, Eqs. (1)–(12), but solved separately with independent sets of environmental forcings (see environmental forcing in Fig. 1). Pelagic fish are forced by surface conditions ($\Pi_\psi$ and $T = T_{75}$), while demersal fish are forced by bottom conditions ($\Pi_\beta$ and $T = T_{bot}$). Whereas the energy supply to the pelagic community remains dependent on surface net primary production (NPP), the particle flux to the bottom provides the energy input to the demersal community. The particle flux is modeled as a depth-dependent fraction of surface primary production:

$$\Pi_\beta = \Pi_\psi \times pe_{ratio} \times \left( \frac{z_{bot}}{z_{eu}} \right)^{b_a}. \tag{13}$$

This formulation assumes a power-law decrease in the flux of organic material below the euphotic layer, i.e., a typical Martin curve (Martin et al., 1987; Buesseler and Boyd, 2009). The attenuation coefficient $b_a = -0.8$ is selected within the range of plausible values (Gloege et al., 2017), and the euphotic layer depth $z_{eu} = 75\,m$ is assumed to be fixed, although both could be modeled to vary in space and time. Here, similar to prior work (Stock et al., 2017; van Denderen et al., 2018; Petrik et al., 2019), we focus on first-order variations in fish biomass in increasingly deep habitats, where food becomes progressively scarce. We calculate the term $(z_{bot}/z_{eu})^{b_a}$ using the high-resolution bathymetry $z'_{bot}$ from the ETOPO global surface relief at 1/10th degree **TS4** (Amante and Eakins, 2009), taking the average across 1° grid cells $(z_{bot}/z_{eu})^{b_a} = \overline{(z'_{bot}/z_{eu})^{b_a}}$. Note that when $z'_{bot}$ is shallower than $z_{eu}$, export is only determined by $pe_{ratio}$, which is taken as a function of surface temperature $T_{75}$ (in °C) and net primary production $\Pi_\psi$, following Dunne et al. (2005).

In the pelagic ocean, the typical size of phytoplankton, $m_\psi$, varies markedly between productive and oligotrophic regions. This variation affects both the length of the food web and the proportion of production accessible to fish communities (Ryther, 1969). We use an empirical phytoplankton size model to account for this variation (Dunne et al., 2005). In analogy with the pelagic ecosystem, we assume that the representative size of benthic organisms at the base of the demersal food web, $m_\beta$, influences the fraction of energy that

reaches demersal fish. For simplicity, we take $m_\beta$ to be globally uniform. Unlike $m_\psi$, for which empirical parameterizations exist, $m_\beta$ is poorly constrained. We keep most food web parameters the same for pelagic and demersal fish, with the exception of the activation energy for growth $\omega_{a,A}$ and mortality $\omega_{a,\lambda}$, since observations of growth rates suggest significant differences between the two communities (van Denderen et al., 2020).

### 3.2.2 Heterogeneous costs

Simulations with BOATSv1 suggest that variations in the cost per unit effort of fishing $c_k$ (in $\mathrm{USD\,W^{-1}\,yr^{-1}}$) played only a secondary role in the development of global fisheries (Galbraith et al., 2017). Yet, heterogeneous costs in the global ocean could modulate the spatial distribution of fishing effort and its evolution over time (Swartz et al., 2010; Anticamara et al., 2011; Lam et al., 2011). Reconstructions of fishing effort in the high seas suggest more than 2-fold average cost differences between distinct fishing gears and regions (see Sala et al., 2018, Kroodsma et al., 2018, and Appendix C).

To simulate the effect of heterogeneous fishing costs on the historical offshore expansion of fisheries, we replaced the constant costs per unit effort ($c_k = 5.85\,\mathrm{USD\,W^{-1}\,yr^{-1}}$) in BOATSv1 by spatially varying costs using a linear function of the distance to shore for effort targeting pelagic fish, $d_{\mathrm{coast}}$ (in km), and a linear function of seafloor depth $z_{\mathrm{bot}}$ (in m) for effort targeting demersal fish (see economic forcing Fig. 1):

$$c_k^{\mathrm{v2}}(x = d_{\mathrm{coast}}, z_{\mathrm{bot}}) = \begin{cases} c_k & \text{when } x \leq x^* \\ c_k + \delta(x - x^*) & \text{when } x > x^*, \end{cases} \quad (14)$$

where $x^*$ is a reference parameter that determines the boundary between coastal and high seas regions. For pelagic effort, $x^*$ identifies a coastal band over which transit costs are assumed to be small compared to other costs. Here we adopt $x^* = 370\,\mathrm{km}$ (or 200 nm), the limit of exclusive economic zones separating coastal regions and high seas. For demersal effort, $x^*$ identifies a depth threshold above which the cost of setting and hauling gears is negligible compared to other costs, and we set $x^* = 200\,\mathrm{m}$. The parameter $\delta$ is the proportionality constant for the increase in costs beyond these coastal bands (in $\mathrm{USD\,km^{-1}\,W^{-1}\,yr^{-1}}$ for pelagic effort and $\mathrm{USD\,m^{-1}\,W^{-1}\,yr^{-1}}$ for demersal effort).

For distance-dependent costs, we select $\delta = 7.9 \times 10^{-3}\,\mathrm{USD\,km^{-1}\,W^{-1}\,yr^{-1}}$ such that the average high seas fishing cost is $9.3\,\mathrm{USD\,W^{-1}\,yr^{-1}}$, comparable with an empirical upper mean value of $8.9\,\mathrm{USD\,W^{-1}\,yr^{-1}}$ (see Appendix C). For depth-dependent costs, the depth of the fishing grounds $z_{\mathrm{bot}}$ is determined from high-resolution (1/10th degree) bottom topography (Amante and Eakins, 2009), taking the shallowest depth in each 1° model grid cell. We set $\delta = 2.5 \times 10^{-3}\,\mathrm{USD\,m^{-1}\,W^{-1}\,yr^{-1}}$ such that the average high seas fishing cost is $9.9\,\mathrm{USD\,W^{-1}\,yr^{-1}}$ for exploitation of deep demersal stocks, comparable with an empir-

ical upper boundary of mean high seas trawling costs of $9.2\,\mathrm{USD\,W^{-1}\,yr^{-1}}$ (Appendix C). Given the uncertainty regarding whether distance or seafloor depth has a greater impact on costs in pelagic and demersal fisheries, we first tested distance- and depth-dependent costs separately and then added them to determine their combined impact.

### 3.2.3 Heterogeneous catchability

In BOATSv1, technological progress, represented by an exponential increase in the catchability $q_k$ at a rate of $5\,\%\,\mathrm{yr^{-1}}$, was shown to be a dominant driver of the development of fisheries (Galbraith et al., 2017). While a homogeneous increase rate approximates the first-order effect of technological progress well, heterogeneous technological efficiencies among fisheries could modulate this development across regions (Palomares and Pauly, 2019), especially as separate gears target distinct resources and are deployed in different ecosystems (Kroodsma et al., 2018). Similar to cost, spatially heterogeneous catchability could have influenced the spatial expansion of fisheries and/or the deepening of catches with time (Watson and Morato, 2013).

To simulate the effect of heterogeneous catchability, the exponential increase is spatially weighted (see economic forcing in Fig. 1):

$$q_k^{\mathrm{v2}}(x = z_{\mathrm{bot}}, y = \mathrm{P\ or\ D}) = q_k\,\mathrm{Pr}(x)\,\mathrm{Of}(y), \quad (15)$$

where $\mathrm{Pr}(x = z_{\mathrm{bot}})$ accounts for spatial variations in technological efficiencies with seafloor depth, and $\mathrm{Of}(y = \mathrm{P\ or\ D})$ is an offset between the catchability of pelagic and demersal resources.

Commercially exploited fish often aggregate near seamounts and other shallow features, resulting in the local establishment of fisheries (e.g., 57 % of longlining activity; Kerry et al., 2022). The coarse resolution of BOATS prevents a direct representation of seamounts. However, the presence of seamounts could increase both fish biomass density and profitability within a model's grid cell, as opposed to the case where resources were more homogeneously distributed across the grid cell. The profile $\mathrm{Pr}(x = z_{\mathrm{bot}})$ parameterizes the effect of seamounts and more generally an increase in the density of resources in shallow regions:

$$\mathrm{Pr}(x = z_{\mathrm{bot}}) = q_{\min} + (1 - q_{\min})\frac{\log_{10}(x_{\max}) - \log_{10}(x)}{log_{10}(x_{\max}) - \log_{10}(x_{\mathrm{mean}})}. \quad (16)$$

Here, $q_{\min} = 0.8$ is the minimum efficiency of gears targeting pelagic resources (see Appendix D), and $x_{\mathrm{mean}}$ and $x_{\max}$ at 2372 and 5750 m depth, respectively, are based on the median and deepest depths of seamounts where fishing activity occurs (Kerry et al., 2022). The depth of the fishing grounds $z_{\mathrm{bot}}$ is determined from ETOPO at 1/10th resolution (Amante and Eakins, 2009), coarsened by taking the shallowest depth in each 1° model grid cell, as described above.

The dominant gears used to target different communities (pelagic vs. demersal) are characterized by different effi-

ciencies (see Appendix D). We tested the effect of separate catchabilities for pelagic and demersal communities, setting $Of(P) = 1.4$ when $Of(D) = 1$. This offset was estimated from the technology coefficients of different gear targeting pelagic and demersal species, weighted by the fraction of global fishing effort for 16 different gears (Kroodsma et al., 2018; Palomares and Pauly, 2019).

## 3.3 Forcing and initialization

Forcing BOATSv2 requires surface temperature $T_{75}$, bottom temperature $T_{\text{bot}}$, and net primary production $\Pi_\psi$. Since we are interested in the recent ocean state, we use climatological observations, and, to assess improvements between BOATS versions, we adopt the same forcing as in Carozza et al. (2017). Surface temperature ($T_{75}$) and temperature at the seafloor ($T_{\text{bot}}$) are taken from the World Ocean Atlas 2009 (Locarnini et al., 2006). $T_{75}$ (in °C) is calculated as the mean temperature over the top 75 m on a 1° grid. $T_{\text{bot}}$ (in °C) is calculated by averaging temperatures at different depths, weighted by the fraction of each depth within a model grid cell as reported by the ETOPO 1/10 bathymetry dataset (Amante and Eakins, 2009). Recognizing that the resolution of observational temperature datasets such as the WOA decreases with depth, we select the layers closest to the bottom as indicative of the temperature near the seafloor. For $\Pi_\psi$ we take the average of three satellite-based estimates at 1° resolution (Behrenfeld and Falkowski, 1997; Carr et al., 2006; Marra et al., 2007). Note that $\Pi_\psi$ and then $\Pi_\beta$ are forced once converted to $\mathrm{g\,m^{-2}\,s^{-1}}$. Forcing BOATS with two-dimensional grids does not account for vertical positions along the water column but characterizes mean environmental conditions where many harvested fish live.

The model is initialized by a "pristine" ocean biomass distribution determined by spinning up the model without fishing for 300 years to reach a steady state. Then, the ecological and economic modules are run together with an increase in the catchability $q_k$ for another 300 years, starting with a small, globally uniform effort. In a given region, fishing begins once catches become profitable (i.e., revenue$_k$ > cost$_k$ in Eq. 6). The open-access dynamics generally first drive an increase in catch, followed by a peak and decline due to overfishing (Guiet et al., 2020). To align simulations with observations (see Sect. 4.1), we estimate the time of the peak catch integrated across LMEs and align it with the time of the observed peak catch, which occurs in the 1990s (Pauly and Zeller, 2016).

## 4 Parameterization procedure

In this section, we first describe the observations used for the evaluation of BOATSv2 and then detail the procedure used to parameterize the model, which is based on the following two steps (Fig. 2). (1) Ecological update: we start by focusing on coastal regions, where most of the observed catch originates and where economic parameterizations are more homogeneous. We parameterize separate pelagic and demersal pathways ($\Pi_\psi$ and $\Pi_\beta$) and growth limitation in HNLC regions for pelagic species ($\alpha^{\text{v2}}$), to determine the best parameter sets for 11 undetermined parameters of the ecological module (see Table 2). (2) Economic update: we then fine-tune the parameters of the economic module using spatially heterogeneous costs and catchability, considering the global ocean (i.e., $c_k^{\text{v2}}$ and $q_k^{\text{v2}}$). Results of this parameterization procedure are described in Sect. 5.1.

## 4.1 Observational data and diagnostics for model evaluation

We use multiple empirical data sources, including catch, biomass, and fishing effort, to tune and evaluate BOATSv2 (in gray Fig. 2). Comparisons are made on globally integrated quantities, quantities integrated across LMEs to assess regional variability in coastal regions, and quantities integrated beyond the boundary of LMEs, i.e., across high seas ecosystems (HSEs; see Appendix E), to assess variability in the open ocean away from coastal influences. We focus on observations around the peak catch in the 1990s but also include observations in the 1950s and 2000s for additional insight.

For fish catch, we use two catch reconstruction datasets: (1) the first is from The Sea Around Us project (SAU; Pauly et al., 2020), corrected for under-reported catch. For the SAU, catches by functional type allow separating pelagic (P) and demersal (D) species (see Appendix F). (2) The second is the database from Watson (2017) (hereafter WAT), including wild catch and corrected for illegal and unreported fisheries. When comparing catches by LME, we focus on 55 LMEs (out of 66) and ignore the Black Sea and a number of high-latitude regions to avoid errors caused by biases in satellite-based chlorophyll and the lack of representation of the effects of sea ice on the marine ecosystem (as in Carozza et al., 2017). We define two diagnostics to help correct for biases in BOATSv1: the fraction of catch in the high seas, $R_C = C_{\text{HS}}/(C_{\text{HS}} + C_{\text{CS}})$, and the catch-weighted mean depth of fishing $Z_C = (\sum_{\text{lat,lon}} C z_{\text{bot}})/(\sum_{\text{lat,lon}} C)$ (in m).

For biomass observations, we use the RAM Legacy Stock Assessment Database (Ricard et al., 2012). Stock assessment data are used to estimate mean catch to biomass ratios (C : B hereafter) in 25 LMEs where enough stock assessments are available, following Bianchi et al. (2021). We also compare historical changes in fish biomass to a global reconstruction based on stock assessments (Worm and Branch, 2012). Furthermore, we compare the model with two biomass databases derived from fishery-independent surveys: the first, encompassing demersal species across 14 large marine ecosystems (LMEs) in the Northern Hemisphere, ranging from the Bering Sea to northern Europe, is based on a recent synthesis of bottom trawl data (van Denderen et al., 2023; Mau-

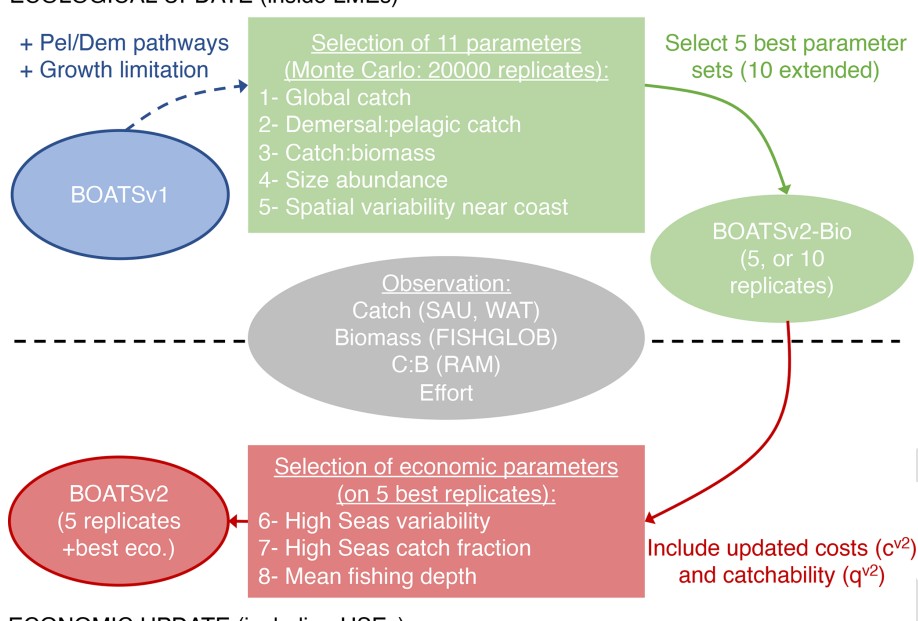

**Figure 2.** Schematic diagram of the parameterization procedure starting from BOATSv1 (in blue), with two steps. (1) Ecological update (in green): a Monte Carlo tuning procedure with five selection criteria is applied to a modified version of BOATSv1 that allows separate pelagic and demersal pathways and growth limitation in iron-limited regions. An ensemble of 20 000 simulations is carried out only for coastal regions with various parameter sets, and we identify a set of 5 (or 10 extended) best parameter sets. (2) Economic update (in red): with three selection criteria, we identify the best economic parameterizations applied to the optimized intermediate BOATSv2-Bio version to determine BOATSv2. We use simulations that include the high seas for the five best parameter sets. Observations used for the parameterization of both are shown in gray.

reaud et al., 2024). The second, focusing on pelagic species, is built on standardized trawls of coastal pelagic species in two LMEs along the North American west coast (Zwolinski et al., 2012).

Finally, we include a comparison with reconstructions of nominal effort for the global fishing fleet in both artisanal and industrial sectors to shed light on the regional development of fisheries (Rousseau et al., 2019, 2024). Similar to catch, we focus on a subset of 55 LMEs where model forcings are more suitable.

### 4.2 Ecological update: Monte Carlo ensemble

To calibrate the revised model based on BOATSv1 (in blue Fig. 2), we must specify the value of 11 poorly determined parameters (Table 2). (Note that the trophic scaling is a function of two free parameters and is thus completely determined by their values.) These parameters are not well constrained by the literature: the activation energies $\omega_{a,A-\lambda}$; the scaling exponents $b$ and $h$; the constants $A_0$ and $\zeta_1$; the trophic scaling $\tau = \ln(\alpha)/\ln(\beta)$ (Brown et al., 2004), itself a function of trophic efficiency $\alpha$ and predator–prey mass ratio $\beta$ (Barnes et al., 2010); the egg survival fraction $s_e$; the threshold mass for fishing selectivity $e_{m_{\Theta,k}}$; and the representative size of benthic organisms $m_\beta$. Following previous

work (Carozza et al., 2017; Bianchi et al., 2021), we adopt an ensemble Monte Carlo approach, running replicates with parameter sets randomly chosen from plausible ranges of values (see Table 2). Note that compared to the calibration of BOATSv1 (Carozza et al., 2017), we assume a subset of parameters ($k_E$, $\Pi^*$, and $c_\sigma$) to be relatively well constrained, since previous analysis showed that variation in these parameters had no significant effect. We also updated a few prior parameter ranges based on recent analyses like $\alpha$ (Stock et al., 2017; Eddy et al., 2020) and because previously optimized values were close to the boundaries of the ranges ($b$, $s_e$). We keep the same parameters for both pelagic and demersal communities, except for the temperature dependence of growth and mortality.

We run 20 000 simulations, each with a distinct combination of parameters, integrated with gradually increasing catchability over time, and select the best simulations according to global and local criteria. These criteria are updated from Carozza et al. (2017) to provide an evaluation of the model performance in reproducing the following features of pelagic and demersal communities.

1. *Global catch*. The best simulations must predict observed global fish catch, when integrated over the 55 LMEs, for the years of maximum catch in the 1990s,

i.e., $C_{\text{Globmax}}^{\text{SAU-WAT}} \simeq 100 \times 10^6\, \text{t yr}^{-1}$ (using tons of wet biomass). There are significant uncertainties around these reconstructions; furthermore, migratory species not represented in BOATS can influence the model's maximum yields. Therefore, we allow catches to be within the range $C_{\text{Globmax}} \in [70, 150] \times 10^6\, \text{t yr}^{-1}$.

2. *Demersal to pelagic catch*. At the global catch peak, simulations must capture the fraction of pelagic and demersal catch. Integrated over all 55 LMEs, pelagic catch and demersal catch respectively account for 45 % and 55 % of catches in SAU – that is, the ratio between demersal and pelagic catch at the peak is $R_{\text{Globmax,D/P}}^{\text{SAU}} \simeq 1.2$. Because of uncertainties around the SAU reconstructions, the presence of migratory species, and additional uncertainty in the allocation of pelagic vs. demersal catch (Appendix F) we allow this ratio to vary within the range $R_{\text{Globmax,D/P}} \in [0.8, 1.8]$.

3. *Catch to biomass*. To ensure that global catches are supported by realistic rates of fish biomass production, we compare the model catch to biomass ratio (C : B) averaged over 25 LMEs to the observational estimate from the RAM database (see Bianchi et al., 2021). We retain simulations for which a Kolmogorov–Smirnov test indicates that the modeled C : B ratios follow the same distribution as the stock assessment data, rejecting cases where distributions are found to be different at the 1 % significance level.

4. *Size abundance*. To preserve a realistic partitioning of fish catches by size groups, for the best simulations we constrain the catch of medium and large sizes to be in the observed range relative to fish in the small group, i.e., $0.3\, C_{\text{small}} < C_{\text{med}}$ and $0.1\, C_{\text{small}} < C_{\text{lrg}} < 0.8\, C_{\text{small}}$.

5. *Spatial variability near the coast*. Finally, we assess the regional variability of catch at the time of the global peak by computing Pearson correlation coefficients of simulated catch densities compared to observations ($r_{\text{LME90s}}^{\text{SAU}}$ or $r_{\text{LME90s}}^{\text{WAT}}$). We also compare simulated and observed maximum catch per functional type and LME, independently of the peak year, to estimate the model capability to reproduce maximum yields per group ($r_{\text{LMEmax}}^{\text{SAU P}}$ or $r_{\text{LMEmax}}^{\text{SAU D}}$).

Criteria (1) to (5) identify parameter sets (Table 2 and results Sect. 5.1.1) that best capture global properties of catches and of fish production per unit biomass for both pelagic and demersal species, focusing on well-sampled coastal regions, completing the ecological update to BOATSv2-Bio (in green in Fig. 2).

## 4.3 Economic update: sensitivity to cost and catchability

After improving the ecological realism of the simulations by tuning selected parameters with a Monte Carlo approach, we improve the economic realism by incorporating heterogeneous costs ($c_k^{\text{v2}}$) and catchability ($q_k^{\text{v2}}$) (in red in Fig. 2). We further evaluated the effect of considering different economic parameterizations and selected the best combination based on regional and global criteria. This evaluation compares the following.

6. *High seas variability*. Beyond LMEs, in HSEs, once heterogeneous costs or catchability are activated we compute Pearson correlation coefficients at peak $r_{\text{HSE90s}}$ to weigh improvements for predicted catch, similarly to step (5) in coastal regions.

7. *High seas catch fraction*. This constraint determines how costs and catchability influence the catches in the high seas and indirectly the historical offshore expansion of fisheries. We computed the high seas catch fraction in the 1950s ($\overline{R}_{\text{C50s}}$) and near the global catch peak of the 1990s ($\overline{R}_{\text{C90s}}$). We expect $\overline{R}_{\text{C50s}} \simeq 0.06$ to increase to only $\overline{R}_{\text{C90s}} \simeq 0.09$ at the global peak, while catch fractions for pelagic and demersal fish increase from $\overline{R}_{\text{C50s}} = 0.10$ to $\overline{R}_{\text{C90s}} = 0.11$ and $\overline{R}_{\text{C50s}} = 0.05$ to $\overline{R}_{\text{C90s}} = 0.07$, respectively.

8. *Mean fishing depth*. Finally, to better characterize the offshore expansion of fisheries, we computed the catch-weighted mean depth over which fishing occurs in the 1950s and 1990s, $\overline{Z}_{\text{C50s–90s}}$. For demersal catch ($\overline{Z}_{\text{C50s,D}} < 136\, \text{m}$ and $\overline{Z}_{\text{C90s,D}} < 206\, \text{m}$), the mean fishing depth reflects the historical deepening of fishing grounds (Watson and Morato, 2013). For pelagic catch ($\overline{Z}_{\text{C50s,P}} < 266\, \text{m}$ and $\overline{Z}_{\text{C90s,P}} < 546\, \text{m}$), it reflects an offshore expansion of fishing effort towards high seas regions with a deeper seafloor.

Criteria (6) to (8) allow refining our understanding of the regional variability of catch and their sensitivity to economic parameterizations (see results in Sect. 5.1.2). They are applied to simulation sets with optimum parameter sets, for multiple cost and catchability profiles, to best capture regional properties of catches, ultimately determining BOATSv2 (Fig. 2).

## 5 Results and discussion

### 5.1 Parameterization

#### 5.1.1 Ecological parameters

The Monte Carlo ensemble allows the identification of optimum ecological parameter sets (Fig. 2). All simulations span

**Table 2.** Model parameters and summary results for the Monte Carlo ensemble (update of Table 3 in Carozza et al., 2017). The sampling distribution of each parameter used in the Monte Carlo simulation are shown, where $X(p1, p2)$ represents the probability distribution ($N$ for normal, $U$ for uniform), and $p1$ and $p2$ are the mean and standard deviation of each parameter, respectively, for the pelagic P and demersal D communities when it applies. Opt. refers to the subset of optimized Monte Carlo simulations, N.O. to the subset of non-optimized simulations. SD is the standard deviation, and KS $p$ value is the $p$ value of the two-sample Kolmogorov–Smirnov test applied to the optimized and non-optimized sets. The three last variables are fixed compared to previous optimizations.

| Parameter | Name | Sampling distribution | Mean opt. | Mean NO | SD opt. | SD NO | KS $p$ value |
|---|---|---|---|---|---|---|---|
| $\omega_{a,A}$ | Growth activation energy | $U_P(0.45, 0.09)$ | 0.50 | 0.45 | 0.088 | 0.089 | $4.7 \times 10^{-3}$ |
| | | $U_D(< U_P, 0.09)$ | 0.37 | 0.30 | 0.14 | 0.13 | $2.6 \times 10^{-3}$ |
| $\omega_{a,\lambda}$ | Mortality activation energy | $U_P(0.45, 0.09)$ | 0.45 | 0.45 | 0.079 | 0.090 | 0.59 |
| | | $U_D(0.45, 0.09)$ | 0.45 | 0.45 | 0.096 | 0.090 | 0.57 |
| $b$ | Allometric scaling exponent | $N(0.55, 0.12)$ & $N(0.70, 0.12)^*$ | 0.72 | 0.63 | 0.06 | 0.15 | $1.4 \times 10^{-9}$ |
| $A_0$ | Allometric growth constant | $N(4.46, 0.50)$ | 4.7 | 4.46 | 0.47 | 0.50 | 0.053 |
| $h$ | Allometric mortality scaling | $N(0.54, 0.09)$ | 0.51 | 0.54 | 0.064 | 0.089 | $1.1 \times 10^{-3}$ |
| $\zeta_1$ | Mortality constant | $N(0.55, 0.57)$ | $-0.072$ | 0.54 | 0.38 | 0.57 | $3.6 \times 10^{-10}$ |
| $\alpha$ | Trophic efficiency | $U(0.23, 0.10)$ | 0.14 | 0.23 | 0.027 | 0.098 | $6.9 \times 10^{-14}$ |
| $\beta$ | Predator to prey mass ratio | $U(5000, 2500)$ | 4970 | 5000 | 2580 | 2510 | 0.94 |
| $\tau$ | Trophic scaling | $\ln(\alpha)/\ln(\beta)$ | $-0.24$ | $-0.19$ | 0.016 | 0.063 | $3.5 \times 10^{-17}$ |
| $s_e$ | Egg survival fraction | $U(0.05, 0.028)$ | 0.052 | 0.050 | 0.025 | 0.028 | 0.49 |
| $e_{m_{\Theta,k}}$ | Selectivity position scaling | $U(0.75, 0.2)$ | 0.77 | 0.75 | 0.20 | 0.20 | 0.54 |
| $\log_{10}(m_\beta)$ | Mean benthic size | $N(-6.5, 0.67)$ | $-6.4$ | $-6.51$ | 0.47 | 0.67 | 0.064 |
| $k_E$ | Eppley constant | – | 0.06 | – | – | – | – |
| $\Pi^*$ | Nutrient concentration | – | 0.35 | – | – | – | – |
| $c_\sigma$ | Selectivity slope | – | 17.8 | – | – | – | – |

$^*$ We merge two ensembles of 10 000 simulations each, with slightly different distributions for $b$. The first ensemble prompted re-selection of the parameter range for the second.

a total catch range of more than 6 orders of magnitude (see Fig. 3). We find that 12 % of simulations satisfy the first criterion of global catch (see Sect. 4.2). Then, for criterion (2), the ratio of demersal to pelagic catch varies by more than 3 orders of magnitude, with 20 % of the simulations with realistic ratios, leaving us with 3.0 % when combined with criterion (1) (Fig. 3). Among these, simulations capturing the observed catch to biomass ratio leave us with 0.8 % of all simulations, and acceptable size distributions ultimately lead to 0.2 % (42 simulations) of all simulations satisfying criteria (1) to (4).

Figure 4 shows the time series of catch, nominal effort, and biomass over all 55 LMEs from 1900 to 2050 for the 42 simulations that meet criteria (1) to (4). In each simulation, global catch increases until reaching a peak, beyond which biomass depletion limits recruitment and drives a fall in catch. Effort follows a comparable pattern, but with a consistent time lag. Biomass monotonically decreases from an initial, nearly pristine state. These features are comparable to observational reconstructions (Fig. 4a, $C^{SAU}$ and $C^{WAT}$). A delayed response of nominal effort is also consistent with observations (Fig. 4b), while the consistent decrease in biomass compares well with aggregated stock assessment data normalized to the pristine period (Fig. 4c, d).

Similar to prior work with BOATSv1, we focus the next analysis on the five best ensemble members selected to capture parameter uncertainty, while maintaining reasonable computational costs. These five parameter sets are selected

out of the 10 best of the 42 simulations based on their ability to reproduce regional variability in peak catches by LMEs – criterion (5). The peak catch is determined almost exclusively by ecological parameters, making it a valuable way to discriminate amongst them (Carozza et al., 2017). Accordingly, we rank the 42 simulations by the Pearson correlation coefficient of simulated vs. observed catch in the 55 LMEs ($r_{LMEmax}^{SAU}$; see Fig. 3) and select 5 ensemble members out of the top 10. These five chosen parameter sets comprise diverse shapes of catch, effort, and biomass histories, but, once averaged together, they provide an ensemble mean that matches the observed historical development of these quantities across LMEs ($r_{LMEmax}^{SAU} \in [0.63, 0.69]$; see Table 3 and Fig. 4, dark red lines).

Note that the Pearson correlation coefficients $r_{LME90s}$ between observed and simulated catch by LME at global peak are comparable with and without updated ecological features (see BOATSv1 compared to other model variants; Table 4; $r_{LME90s}^{SAU} \simeq 0.69$ and $r_{LME90s}^{WAT} \simeq 0.73$). However, the updated ecological features provide large improvements in the high seas (e.g., $r_{HSE90s}^{SAU}$ increases from 0.22 in v1 to 0.58). This improvement in the high seas is partly explained by the representation of iron limitation on fish growth ($r_{HSE90s}^{SAU}$ increases to 0.81 from 0.22 in v1), while along coastal regions, iron limitation alone is insufficient to explain catch (Table 4).

**Table 3.** List of parameter values for the selected five best simulations with updated ecological features.

|                              | 211         | 3773        | 14 028      | 14 349      | 15 436      |
|------------------------------|-------------|-------------|-------------|-------------|-------------|
| $r_{\text{LMEmax}}^{\text{SAU}}$ | 0.66        | 0.63        | 0.69        | 0.68        | 0.64        |
| $\omega_{a,A}$               | 0.30/0.16   | 0.42/0.33   | 0.42/0.17   | 0.43/0.12   | 0.47/0.25   |
| $\omega_{a,\lambda}$         | 0.53/0.61   | 0.47/0.49   | 0.39/0.61   | 0.43/0.57   | 0.42/0.51   |
| $b$                          | 0.73        | 0.75        | 0.75        | 0.80        | 0.70        |
| $A_0$                        | 4.35        | 4.49        | 4.35        | 5.04        | 4.49        |
| $h$                          | 0.49        | 0.55        | 0.46        | 0.46        | 0.55        |
| $\zeta_1$                    | −0.10       | −0.68       | −0.07       | −0.25       | −0.33       |
| $\alpha$                     | 0.14        | 0.09        | 0.10        | 0.12        | 0.12        |
| $\beta$                      | 2830        | 5890        | 8890        | 8510        | 6330        |
| $\tau$                       | −0.24       | −0.27       | −0.25       | −0.23       | −0.24       |
| $e_{m_{\Theta,k}}$           | 0.44        | 0.76        | 0.97        | 0.75        | 0.67        |
| $s_e$                        | $9.3\ 10^{-2}$ | $5.0\ 10^{-2}$ | $4.8\ 10^{-2}$ | $2.7\ 10^{-2}$ | $2.4\ 10^{-2}$ |
| $\log_{10}(m_\beta)$         | −6.6        | −6.0        | −6.1        | −6.6        | −6.0        |

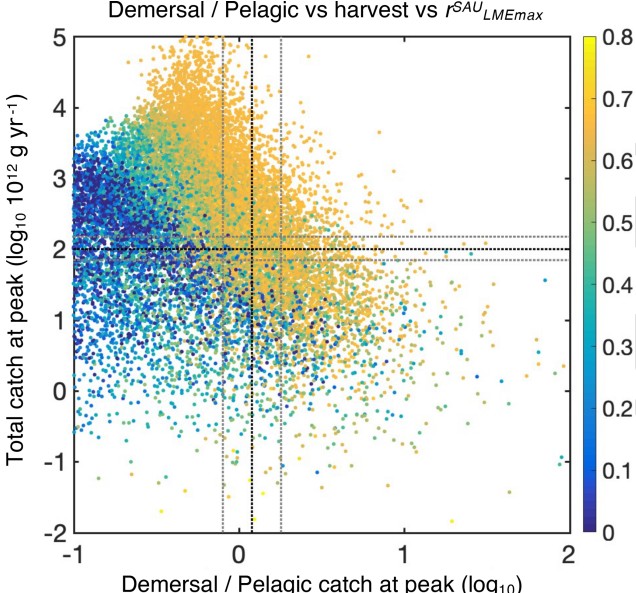

**Figure 3.** Simulations of global catch features from the BOATSv2 Monte Carlo ensemble. The total catch at the global peak of the 1990s $C_{\text{Globmax}}$ is shown as a function of the ratio between demersal and pelagic catch at the global catch peak, $R_{\text{Globmax,D/P}}$, for the 20 000 simulations in the ensemble. Colors show the Pearson correlation coefficient $r_{\text{LMEmax}}^{\text{SAU}}$ of simulated vs. observed (SAU) maximum catch in 55 LMEs, $C_{\text{LMEmax}}$, for each simulation. The dotted black horizontal line shows the reference global harvest, $C_{\text{Globmax}}^{\text{SAU-WAT}} = 100 \times 10^{12}\,\text{g yr}^{-1}$. The dotted black vertical line shows the observed ratio of demersal to pelagic catch at the global peak, $R_{\text{Globmax,D/P}}^{\text{SAU}} = 1.2$. The horizontal and vertical gray lines indicate the ranges within which the best simulations are selected.

### 5.1.2 Economic parameters

As summarized in Table 4, increased ecological realism improves the model's ability to reproduce high seas fisheries, in particular the fraction of high seas catch ($\overline{R}_{\text{C90s}}$ down to 0.16 from 0.40 in BOATSv1) and the catch-weighted mean depth of fishing ($\overline{Z}_{\text{C90s}}$ down to 694 from 1698 m). These improvements reflect growth limitation in HNLC regions ($\alpha^{\text{v2}}$) (Galbraith et al., 2019) and, to a greater extent, explicit separation of pelagic and demersal energy pathways ($\Pi_\psi$ and $\Pi_\beta+\alpha^{\text{v2}}$; hereafter BOATSv2-Bio; green line in Fig. 5a, b).

Since economic drivers could explain additional spatial variability, we test plausible heterogeneous economic parameterizations (Fig. 2). The heterogeneous costs and catchabilities have little impact on the coastal variability of catch at the time of the global peak ($r_{\text{LME90s}}^{\text{SAU}} \simeq 0.69$ and $r_{\text{LME90s}}^{\text{WAT}} \simeq 0.73$; Table 4). Note that the comparison reveals better correlations when comparing models with WAT catch reconstructions instead of SAU reconstructions. Most of the improvement is explained by higher mean catches in Australian LMEs (compare Fig. 5c and d), but the explanation for such discrepancy in the observational reconstructions remains unclear. We also compare the Pearson correlation coefficients of maximum catch by LME for pelagic and demersal catch separately. Heterogeneous costs or catchability show no effect on the variability of maximum pelagic and demersal catch yields: $r_{\text{LMEmax}}^{\text{SAUP}} \simeq 0.46$ vs. $r_{\text{LMEmax}}^{\text{SAUD}} \simeq 0.69$ (Table 4); these should instead influence the timing of the development of fisheries. Both correlations suggest that, along the coast, catches are independent of economic parameterizations and are instead controlled mainly by the environment.

To select economic parameterizations, criterion (6) and the catch variability in the high seas indicate only minor variations; economic parameters do not significantly enhance the accuracy of catches in the HSEs compared to the improvement from BOATSv1 to BOATSv2-Bio (see Table 4). How-

**Table 4.** Sensitivity of model variants for various combinations of ecological and economical processes. We report Pearson coefficients between mean simulated and observed catch across the 55 selected LMEs at the global peak $r_{LME90s}$ or for the historical maximum $r_{LMEmax}$ and across high seas ecosystems (HSEs) at global the peak $r_{HSE90s}$. We report the fraction of high seas vs. coastal catch $R_C$ and the mean depth of catch $\overline{Z_C}$ in the 1950s and near the global peak of the 1990s. The superscripts SAU and WAT indicate the kind of observation for pelagic (P) or demersal (D) communities with SAU.

| Observation or model version | $r^{SAU}_{LME90s}$ | $r^{WAT}_{LME90s}$ | $r^{SAUP}_{LMEmax}$ | $r^{SAUD}_{LMEmax}$ | $r^{SAU}_{HSE90s}$ | $r^{WAT}_{HSE90s}$ | $\overline{R_C}_{50s}$ (P; D) | $\overline{R_C}_{90s}$ (P; D) | $\overline{Z_C}_{50s}$ (P; D) | $\overline{Z_C}_{90s}$ (P; D) |
|---|---|---|---|---|---|---|---|---|---|---|
| SAU | | | | | | | 0.07 (0.10; 0.05) | 0.09 (0.11; 0.07) | 188 (266; 136) | 372 (546; 206) |
| WAT | | | | | | | 0.05 | 0.08 | 126 | 154 |
| BOATSv1 | 0.69 | 0.73 | | | 0.22 | 0.39 | 0.11 | 0.40 | 445 | 1698 |
| v1 + $\alpha$v2 | 0.50 | 0.62 | | | 0.81 | 0.71 | 0.09 | 0.31 | 386 | 1440 |
| v2-Bio* & $\Pi_\beta$ | 0.69 | 0.73 | 0.46 | 0.69 | 0.58 | 0.67 | 0.05 (0.13; 0.02) | 0.16 (0.32; 0.07) | 179 (273; 43) | 694 (1001; 316) |
| v2-Bio + $c^{v2}_k$ ($z_{bot}$) | 0.69 | 0.73 | 0.46 | 0.69 | 0.59 | 0.68 | 0.05 (0.13; 0.02) | 0.14 (0.31; 0.06) | 109 (280; 36) | 571 (991; 148) |
| v2-Bio + $c^{v2}_k$ ($d_{coast}$) | 0.69 | 0.73 | 0.46 | 0.69 | 0.51 | 0.63 | 0.05 (0.12; 0.02) | 0.14 (0.28; 0.07) | 110 (267; 43) | 555 (873; 259) |
| v2-Bio + $q^{v2}_k$ ($z_{bot}$, P) | 0.69 | 0.74 | 0.46 | 0.69 | 0.56 | 0.67 | 0.06 (0.11; 0.02) | 0.13 (0.27; 0.07) | 78 (124; 37) | 526 (827; 246) |
| v2-Bio + $q^{v2}_k$ ($z_{bot}$, P or D) | 0.68 | 0.73 | 0.46 | 0.69 | 0.51 | 0.64 | 0.05 (0.11; 0.02) | 0.10 (0.27; 0.05) | 47 (79; 28) | 315 (553; 103) |
| BOATSv2 | 0.68 | 0.73 | 0.46 | 0.69 | 0.51 | 0.64 | 0.06 (0.10; 0.02) | 0.11 (0.24; 0.06) | 75 (121; 34) | 420 (718; 141) |

* v2-Bio = v1 + $\alpha$v2 + $\Pi_\psi$.

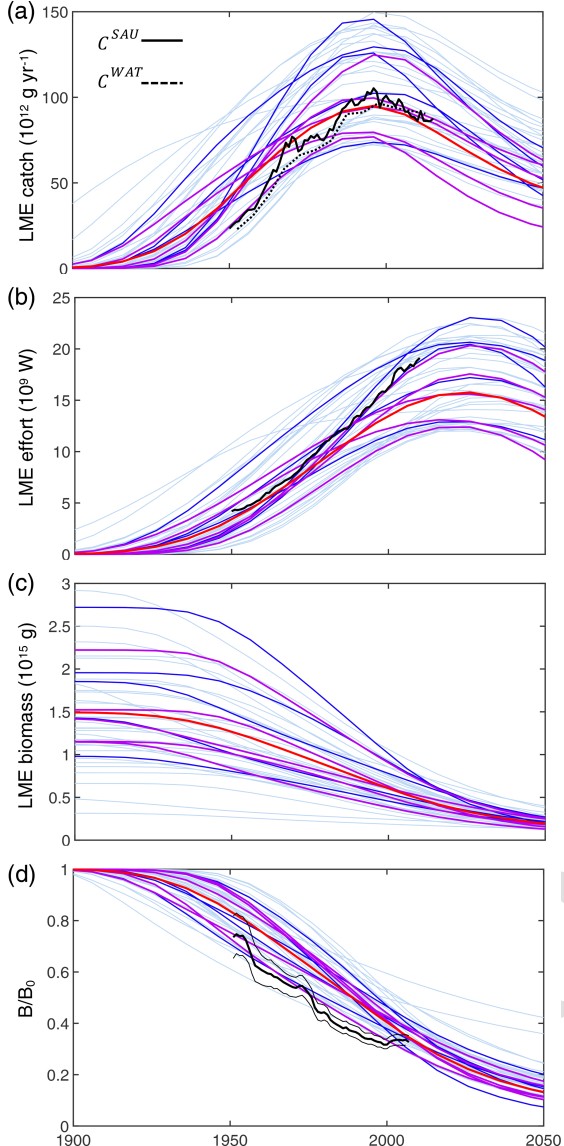

**Figure 4.** Historical simulations of catch, effort, and biomass for the best 42 simulations in the BOATSv2 Monte Carlo ensemble, forced only with exponentially increasing catchability over time. **(a)** Catch, **(b)** nominal effort, **(c)** biomass, and **(d)** biomass normalized to initial biomass in the selected 55 LMEs for the 42 best parameter sets from 1900 through 2050. Ensembles are aligned at the catch peak of the 1990s. The light blue lines show each parameter set; the dark blue lines show the 10 best simulations, out of which the 5 purple lines show the final best ensemble. The thick red line is the mean of the five best ensemble simulations. Black lines in panels **(a)**, **(b)**, and **(d)** show observational reconstructions, consisting of catch from SAU (black line in panel **a**, $C^{\text{SAU}}$) and the WAT database (dotted black line in panel **a**, $C^{\text{WAT}}$), effort (black line in panel **b**), and biomass from fish stock assessments normalized to the initial state (black line in panel **d**). Note that the simulations do not include the effects of climate change, environmental variability other than the seasonal cycle, and management.

ever, for criterion (7), economic parameterizations further reduce the high seas catch fraction (down to $\overline{R}_{\text{C90s}} = 0.10$ compared to $\simeq 0.09$; Table 4), and criterion (8) significantly reduces the mean fishing depth (down to $\overline{Z}_{\text{C90s}} = 315$ m; Table 4).

Heterogeneous costs and catchability parameterizations have unequal effects. Depth- and distance-dependent costs respectively reduce the offshore expansion of demersal ($\overline{Z}_{\text{C90s,D}}$ down to 148 m) and pelagic ($\overline{Z}_{\text{C90s,P}}$ down to 873 m) catches. Since both slightly improve aspects of the simulations, we retain both parameterizations in the final BOATSv2 update. Regarding catchability, accounting for the effect of seamounts also reduces the development of fishing over a deep seafloor. When applied to pelagic catches only, $\overline{Z}_{\text{C90s}}$ decreases to 526 m (Table 4). When this correction is applied both to pelagic and demersal communities, this is further reduced to $\overline{Z}_{\text{C90s}} = 315$ m, close to observational estimates. However, the maximum depth of the demersal catch becomes excessively shallow ($\overline{Z}_{\text{C90s,D}} = 103$ m). Therefore, we retain the heterogeneous catchability parameterization only for pelagic fishing effort in the final BOATSv2 update.

For the final reference simulations with BOATSv2, the inclusion of costs that increase with the distance from shore, costs that increase with seafloor depth for demersal fishing, and catchabilities that decrease with seafloor depth for the pelagic community leads to an improvement of the model's ability to reproduce the delayed development of high seas fisheries and the progressive deepening of catch (Table 4 and see red lines in Fig. 5a, b).

In summary, at the LME level, BOATSv2 and BOATSv1 have similar accuracy in their representation of regional catches (see $r_{\text{LME90s}} = 0.69$ or 0.73 when compared to SAU or WAT; Table 4 and Fig. 5c, d). This lack of improvement in the new model version is explained by limited accuracy in predicting pelagic catches across LMEs (see $r_{\text{LMEmax,P}} = 0.46$ vs. $r_{\text{LMEmax,D}} = 0.69$; Table 4 and Fig. 5e, f). Nevertheless, BOATSv2 better captures the large-scale variability of catches in the HSEs, which are approximately 1 order of magnitude smaller than in LMEs (see $r_{\text{HSE90s}} = 0.51$ or 0.64 compared to SAU or WAT; Table 4 and Fig. 5g, h), and better reproduces their historical offshore expansion ($\overline{R}_{\text{C90s}} = 0.11$ and $\overline{Z}_{\text{C90s}} = 420$ m; Table 4).

### 5.1.3 Model sensitivity

The best parameter sets selected by the Monte Carlo approach (Sect. 4.2) provide insights on the functioning of ecological communities. Of the 11 parameters that were optimized for, 6 have posterior distributions significantly different from the prior distributions ($p$ values $< 10^{-2}$; Table 2). The posterior distributions for these six parameters were also different when optimizing BOATSv1 (see Carozza et al., 2017), confirming their essential role in influencing the sensitivity of the model.

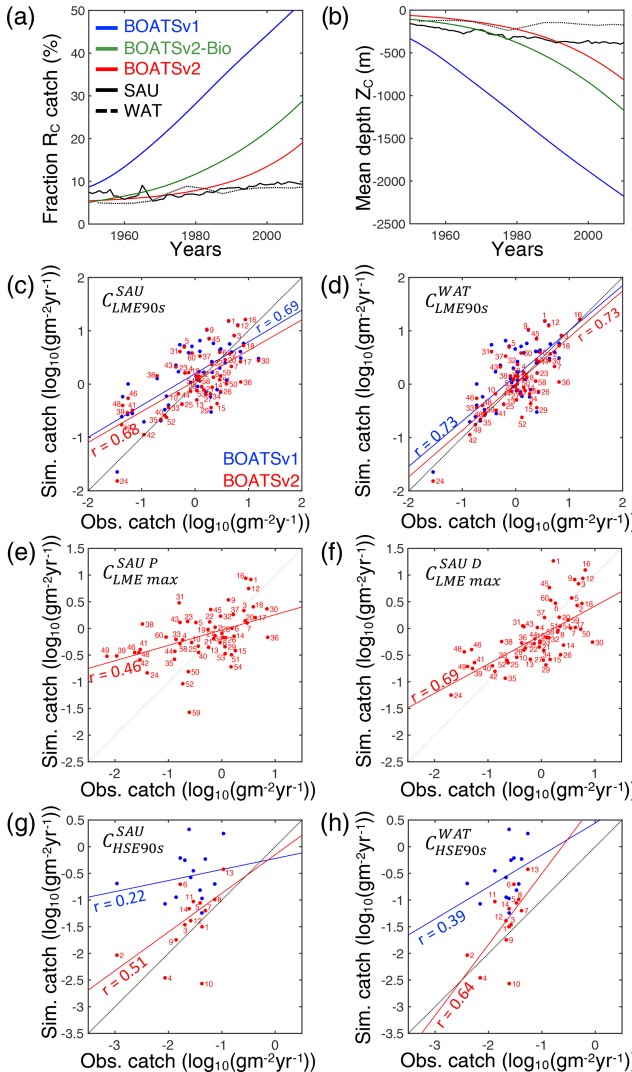

**Figure 5.** Evaluation of BOATSv2. (**a**) Observed and simulated historical development of high seas vs. coastal seas catch $R_C$. (**b**) Observed and simulated historical deepening of global catches $Z_C$. (**c, d**) Scatter plot of the observed vs. simulated catch at the global peak in 55 LMEs for SAU reconstructions $C_{LME90s}^{SAU}$ in (**c**) and WAT reconstructions $C_{LME90s}^{WAT}$ in (**d**). (**e, f**) Scatter plot of the observed vs. simulated maximum catch in LMEs for pelagic $C_{LMEmax}^{SAUP}$ in (**e**) and demersal catch $C_{LMEmax}^{SAUD}$ in (**f**). (**g, h**) Scatter plot of the observed vs. simulated catch at the global peak in HSEs for SAU reconstructions $C_{HSE90s}^{SAU}$ in (**g**) and WAT reconstructions $C_{HSE90s}^{WAT}$ in (**h**). In panels (**a**) and (**b**), SAU and WAT reconstructions are indicated by solid and dotted lines, respectively; BOATSv1 is indicated by the blue line, BOATSv2-Bio by green, and BOATSv2 by red. In panels (**c**)–(**h**) blue dots and lines show BOATSv1 and red dots and lines BOATSv2. Numbers next to each dot indicate the LMEs or HSEs (see regions in Appendix E).

First, the ensemble mean allometric scaling exponent of $b = 0.72$ (the range is 0.70–0.80 for the five best ensembles; see Table 3) is larger than the BOATSv1 value of 0.65 (Carozza et al., 2017) but in the middle of the expected range of 0.66–0.75 (Brown et al., 2004; Kooijman, 2010; Hatton et al., 2019). Second, the mortality rate parameter $\zeta_1$ was selected to be slightly negative ($-0.07$), different from the initial distribution (mean 0.54). Note that the negative value for $\zeta_1$ does not indicate a negative mortality, since $\Lambda_k \propto e^{\zeta_1}$ in Eq. (4). The mortality scaling (0.51) was also smaller than the mean value (0.54). To account for a large uncertainty in the trophic efficiency ($\alpha$) (Eddy et al., 2020), we expanded its prior range to [0.06, 0.4] (compared to previous estimates in Carozza et al., 2017). However, the optimization persistently selected values comparable to BOATSv1, with a mean of 0.14 (range 0.09–0.14; Table 3). Although separate pelagic and demersal communities could have different trophic efficiencies (Stock et al., 2017; Du Pontavice et al., 2020), for simplicity we adopt the same value here. The robustness of the optimized trophic efficiency suggests that sources of variability could be captured by other model parameterizations, e.g., the representative size of primary producers or the temperature dependence of growth and mortality. Lastly, growth activation energies ($\omega_{a,A}$) for pelagic (0.50) and demersal (0.37) communities are larger than the prior values. Although the temperature dependence of mortality ($\omega_{a,\lambda}$) is not significantly different from the initial values, the optimized values suggest a stronger sensitivity of growth compared to mortality for the pelagic community ($\omega_{a,A} - \omega_{a,\lambda} = +0.047\,\text{eV}$) and a stronger sensitivity of mortality for the demersal community ($-0.082\,\text{eV}$).

Covariations between parameters in the 42-member optimized ensemble reveal compensations between parameter pairs (see Fig. 6a). The most significant compensations are between parameters controlling the biomass flow through the size spectrum and biomass losses (see Fig. 6b–d). For instance, an increase in the trophic efficiency ($\alpha$) can be compensated for by a smaller predator–prey biomass ratio ($\beta$), which lengthens the food web (Fig. 6b; $r = -0.66$). When more biomass flows across trophic levels, longer food chains ultimately lead to greater losses over the food web and thus similar fish biomass production. Alternatively, an increase in the trophic efficiency can be compensated for by an increase in the mortality parameter ($\zeta_1$; Fig. 6c; $r = 0.42$). Conversely, the mortality parameter $\zeta_1$ decreases when the growth scaling exponent ($b$) increases (Fig. 6d; $r = -0.49$), instead of decreasing, because of indirect impacts on the asymptotic size ($m_\infty$).

Correlations between parameters that differ between pelagic and demersal food webs can also reveal trade-offs, particularly in how activation energies collectively affect the two communities (see Fig. 6e–g). For instance, an increased temperature sensitivity of growth for the pelagic community $\omega_{a,A}^P$ is matched by an increased sensitivity of growth

for the demersal community $\omega_{a,A}^{D}$ (Fig. 6e; $r = 0.74$) and a shift of the sensitivity of demersal mortality compared to demersal growth $\omega_{a,A-\lambda}^{D}$ (Fig. 6f; $r = 0.65$). Another relationship between communities is observed for losses. As the temperature dependence of mortality for the pelagic community ($\omega_{a,\lambda}^{P}$) increases, increasing losses, there is a concurrent decrease in the representative size at the base of the benthic food chain ($\log_{10}(m_{\beta})$). This extends the food chain length, increasing losses in the demersal community (Fig. 6g; $r = -0.40$).

While not exhaustive, this parameter analysis suggests trade-offs between biomass production and dissipation in pelagic and demersal communities (see Fig. 6a for further details).

## 5.2   Features of the simulated catch

### 5.2.1   Global catch

Relative to BOATSv1, BOATSv2 corrects for the overestimate of high seas catch (see Fig. 5) while maintaining similar skill in reproducing historical variations of LME fish catch (see comparison of model ensemble means with SAU and WAT reconstructions; Fig. 4). BOATSv2 also shows improved skill in hindcasting the spatial evolution of catch (see Fig. 7) and the offshore and equatorward expansion of fisheries (Swartz et al., 2010; Guiet et al., 2020). In the 1950s (Fig. 7a–c), higher-latitude shelf regions and productive upwelling regions contributed the most to global catch. In the 1990s (Fig. 7d–f), while high latitudes still produced large catches, subtropical regions were also significantly exploited, especially in shallow regions. Productive high seas areas also supported significant fishing. The expanded representation of ecological processes accounts for most improvements in the high seas, while updating economic processes only yields minor improvements (see Appendix G).

Despite the closer fit to observations, model biases remain, in particular low offshore catches in the western equatorial Pacific and excessive catches in the northern and southern Atlantic (see Fig. 7d). These biases are not improved by the economic update and are likely related to ecological factors (see Fig. G1 panel b vs. c). However, it remains unclear if biases could also result from historical interactions between ecosystems and fishing effort or from changing environmental conditions. Processes not included in the model, such as habitat alteration by bottom-trawling gear, additional constraints on habitats such as dissolved oxygen (Deutsch et al., 2020), fish migrations and movement (Barrier et al., 2023; Guiet et al., 2022; Lehodey et al., 2008; Watson et al., 2015), or management and regulation, likely play a role in these biases. While the two observational catch reconstructions used to calibrate the model show differences, likely related to different approaches (e.g., using bathymetry or not; compare Fig. 7e, f), biases in simulated catches are apparent when comparing with either reconstruction.

Figure 8 shows the differences (residuals) between maximum simulated and observed peak catches in each LME. While there is an overall improvement from BOATSv1 to BOATSv2 (compare Fig. 8a, b to c, d), areas of overestimated (e.g., Indian Ocean) or underestimated catches (e.g., northwestern Pacific) are correlated between the two model versions, suggesting structural biases in the model. It is possible that accounting for features of coastal habitats such as coral reefs and mangrove forests could reduce regional biases, especially in Southeast Asia (Tittensor et al., 2010). Representation of biodiversity also remains crude, and additional functional types with life histories that differ from those of fish, such as cephalopods, could be considered (Denéchère et al., 2024). Finally, some larger predators that dive to feed on the deep scattering layer experience environmental conditions that differ from those at the surface (Nuno et al., 2022; Braun et al., 2023). Accounting for this effect could help reduce model biases.

### 5.2.2   Pelagic vs. demersal catch

Separate pelagic and demersal energy pathways allow simulation of higher taxonomic diversity. At the global peak of the 1990s, a large fraction of simulated demersal catch is derived from high latitudes (Fig. 9), in general agreement with observations (van Denderen et al., 2018). At lower latitudes, modeled demersal catches are as abundant as pelagic catches in shallow regions or near seamounts, also consistent with observations. Significant biases remain, however, such as in the North Atlantic, where the simulated demersal catch fraction is lower than observed, and the eastern tropical Pacific, where the demersal fraction is overestimated. The latter bias could reflect the parameterization of iron limitation, which reduces accumulation of pelagic biomass in the eastern topical Pacific, an HNLC region (see Appendix B).

### 5.2.3   Deepening of the catch

The historical expansion of fisheries is associated with fishing in increasingly deep waters, i.e., a deepening of the catch (Morato et al., 2006; Watson and Morato, 2013). This can be attributed to the need to find new profitable fishing grounds beyond more accessible coastal regions, as well as improvements in fishing technology. The catch density per depth stratum from observational reconstructions reflects such expansion (see Fig. 10a and b for pelagic and demersal catch in SAU data). The deepening of demersal catches is consistent with increasingly deep fishing grounds, while the deepening of pelagic catches indicates an expansion of fishing effort towards deeper regions offshore.

In the model, decreasing biomass with depth slows the historical deepening of demersal catches (Fig. 10d). Similarly, higher costs and reduced catchability at greater depths delay the offshore expansion and deepening of pelagic catches (Fig. 10c). These factors collectively contribute to the slower

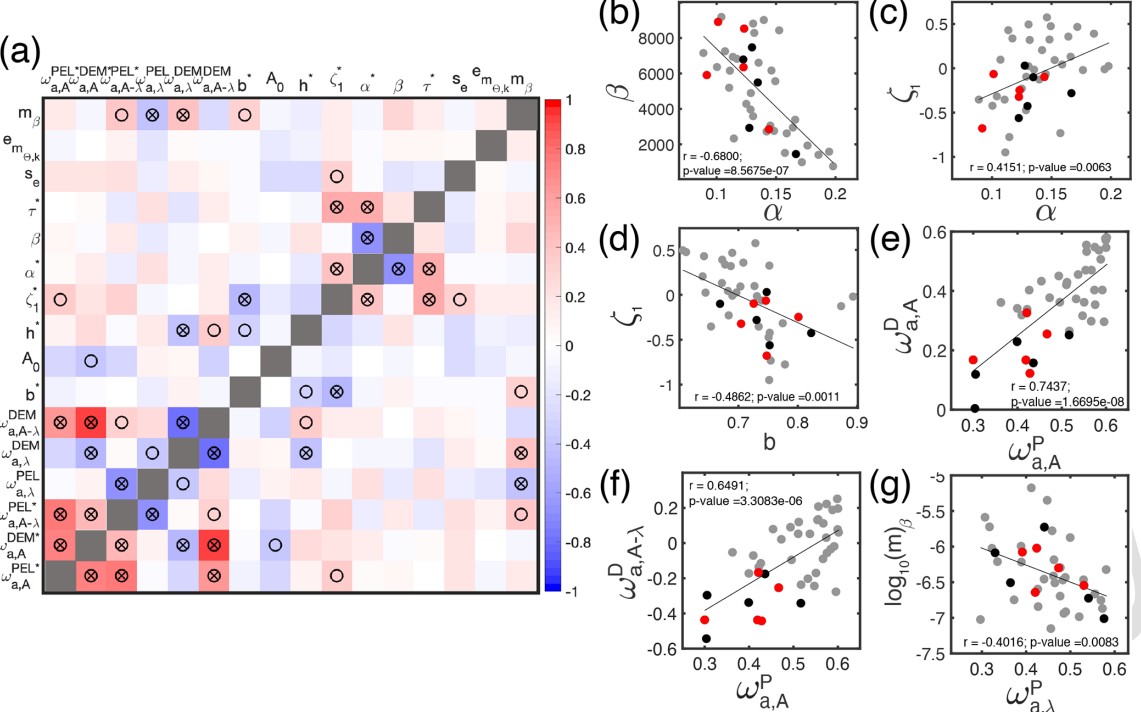

**Figure 6.** Emergent relationships between model parameters. **(a)** Pairwise correlations between model parameters for the 42-member optimized ensemble. **(b–g)** Scatter plots illustrating the relationships between the most strongly correlated parameters: **(b)** trophic efficiency ($\alpha$) and predator–prey mass ratio ($\beta$); **(c)** trophic efficiency ($\alpha$) and mortality parameter ($\zeta_1$); **(d)** growth scaling exponent ($b$) and mortality parameter ($\zeta_1$); **(e)** growth activation energy for the pelagic community ($\omega_{a,A}^P$) and demersal community ($\omega_{a,A}^D$); **(f)** growth activation energy for the pelagic community ($\omega_{a,A}^P$) and difference between growth and mortality activation energies ($\omega_{a,A-\lambda}^D$) for the demersal community; **(g)** mortality activation energy for the pelagic community ($\omega_{a,\lambda}^P$) and representative size of organisms at the base of the demersal food web ($m_\beta$). In panel **(a)**, circles indicate $p$ values $< 0.05$ and stars $p$ values $< 0.01$. In panels **(b)**–**(g)**, the lines show linear regressions for the 42 parameter values of the ensemble; Pearson correlation $r$ and $p$ values are reported in each plot. In panels **(b)**–**(g)**, red dots indicate the 5 final best parameter values for BOATSv2, while the black dots show the remaining 5 parameter values among the 10 best; gray dots indicate all other parameter values.

development of fisheries in deep waters and the reduced catch fraction from the high seas, consistent with observational reconstructions (compare with BOATSv1 in Appendix H).

## 5.3 Features of simulated fishing effort

The modeled nominal fishing effort aggregated across the 55 LMEs broadly matches observations (see observations in Fig. 4b), with a slightly earlier decline that falls within the uncertainty range. This could indicate that the model's effort responds to biomass depletion faster than observed or that the model underestimates the resilience of exploited stocks. It could also reflect the lack of management and subsidies in the model, which influence profitability and the progression of fishing effort.

The significant correlation between modeled and observed effort at peak catch across LMEs (see Fig. 11a; $r_{\text{LME90s}} = 0.57$) lends support to the model's assumption of open-access dynamics. However, significant deviations remain (Fig. 11b).

For instance, the model overestimates effort in highly productive shelf regions near the mouth of major rivers such as the Patagonian Shelf or North Brazil Shelf, suggesting development of fisheries that is too rapid compared to neighboring regions. Biomass redistribution by currents, or fish stock migrations, could correct this bias. At larger scales, the model underestimates effort across Southeast Asia, consistent with the underestimated peak catch. However, effort around Australia is also lower than observed, while the model overestimates catches there (see Fig. 8). This mismatch suggests regional differences in economic drivers or missing key habitats such as mangroves or reefs. Efficient management might also play a role, although it is unlikely to be the sole driver across the entire region.

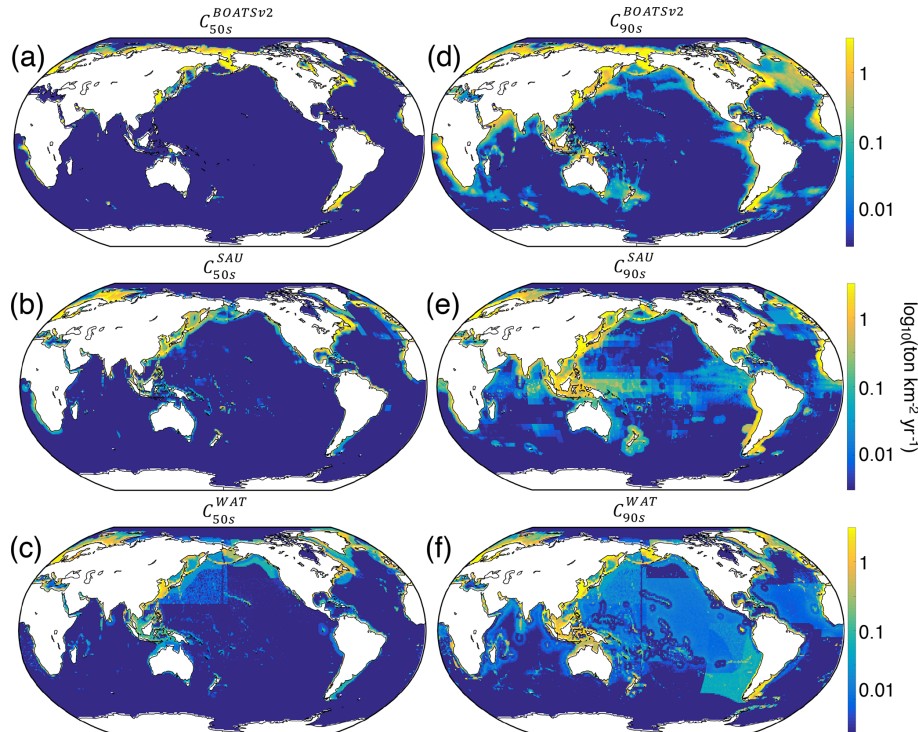

**Figure 7.** Observed and simulated catch in the 1950s and 1990s, $C_{50s-90s}$ (in $\log_{10}$; $t\,km^{-2}\,yr^{-1}$). **(a, d)** BOATSv2 simulation forced with exponentially increasing catchability over time compared to **(b, e)** SAU and **(c, f)** WAT observations.

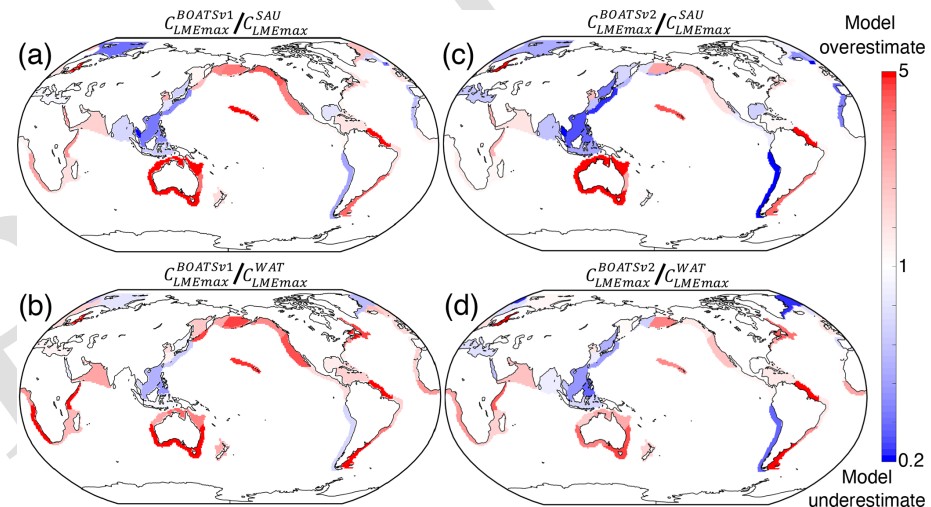

**Figure 8.** Residuals of observed vs. simulated maximum catch per LME, $C_{LMEmax}$ ($\log_{10}(C^{sim}/C^{obs})$ TS6). **(a, b)** BOATSv1 simulation compared to SAU and WAT observations. **(c, d)** BOATSv2 simulation compared to SAU and WAT observations.

## 5.4 Features of simulated biomass

### 5.4.1 Global biomass

In the absence of fishing, BOATSv2 estimates a commercial fish biomass of 1.9 Gt aggregated over LMEs, slightly larger than previous estimates from BOATSv1 (1.6 Gt in Bianchi et al., 2021). However, because the biomass in the high seas is lower, the biomass in LMEs accounts for 68 % of the global biomass (2.8 Gt). This is significantly larger than the 50 % of BOATSv1 (3.3 Gt in Bianchi et al., 2021). Thus, BOATSv2 suggests a 10 %–15 % smaller pristine biomass than BOATSv1. When fishing is included and forced by the historical catchability increase, the BOATSv2 commercial fish biomass aggregated across LMEs declines by about 50 %

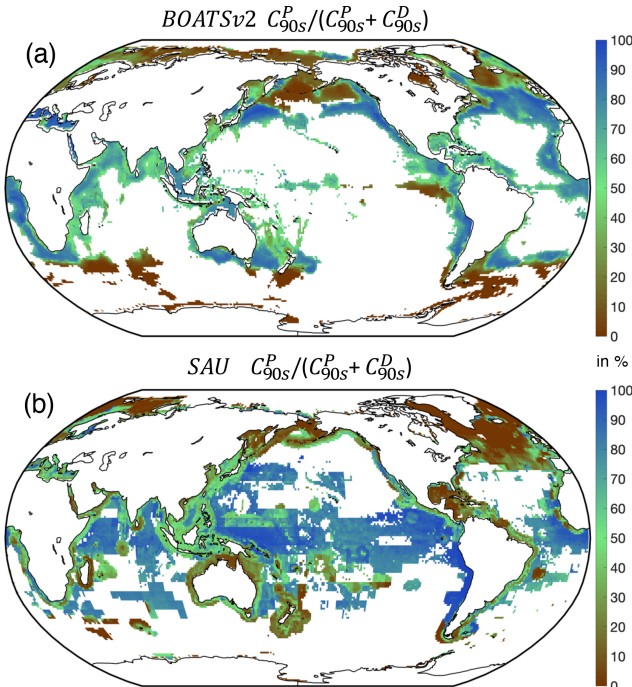

**Figure 9.** Observed and simulated pelagic catch fraction $C_{90s}^P / (C_{90s}^P + C_{90s}^D)$ (in %). **(a)** BOATSv2-simulated fraction compared to **(b)** SAU observations. Very low catch levels are masked (i.e., below $10^{-3}$ g m$^{-2}$ yr$^{-1}$).

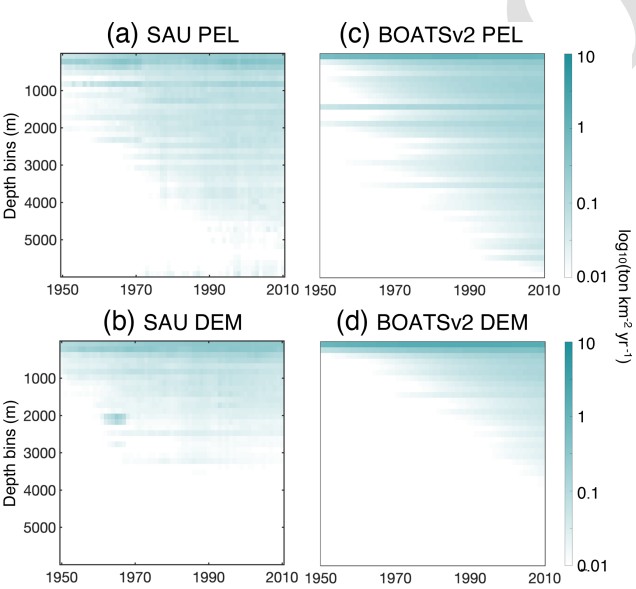

**Figure 10.** Fishing over increasingly deep seafloors. **(a, b)** Observed mean catch density (in $\log_{10}$; t km$^{-2}$ yr$^{-1}$) over depth strata for pelagic **(a)** and demersal **(b)** catch from SAU. **(c, d)** Simulated mean catch density over depth strata for pelagic **(c)** and demersal **(d)** catch in BOATSv2 (see Appendix H for comparison with BOATSv1).

from 1950 to 2000. This is consistent with both BOATSv1 simulations and global observational estimates (Fig. 4d; also compare with Worm and Branch, 2012). Interestingly, at peak catch, the LME–HSE difference between model versions is compensated for by differences in fishing effort so that both LMEs and HSEs hold approximately 50 % of the global biomass at this point (0.9 Gt within LMEs and 0.8 Gt in HSEs with BOATSv2, similar to 0.6 and 0.5 Gt, respectively, with BOATSv1; Bianchi et al., 2021). The optimized parameters of BOATSv2 suggest that the global fish biomass is about 40 % pelagic and 60 % demersal, a partitioning which could be relevant for the biogeochemical cycling and carbon export effects of fish. Comparing BOATSv1 and BOATSv2, the similar relative biomass distribution at peak harvest and the similar magnitude of pelagic biomass would suggest comparable estimates of export and sequestration by sinking fecal pellets (Bianchi et al., 2021). However, further analyses is needed to differentiate the roles of pelagic and demersal communities and their historical depletion in carbon and nutrient sequestration (Cavan and Hill, 2022).

### 5.4.2 Regional biomass distributions

In the model, shallow shelves and upwelling systems sustain on average 3 times more biomass per unit area than the high seas (10. vs. 2.9 g m$^{-2}$ within and outside the LMEs, respectively; see also Fig. 12a for local biomass gradients). Validating these predictions is challenging because of observational limitations; however, recent compilations provide a new way to assess them.

For demersal fish, scientific trawl compilations are now available from the Northern Hemisphere at locations ranging from Alaska to Europe (Maureaud et al., 2024; van Denderen et al., 2023). Figure 12 shows that BOATSv2 accurately simulates the average biomass across these LMEs and captures the biomass increase from the Gulf of Mexico–Florida (GM–FL) to Europe (EU) and the North American west coast (NA–W) (see circles Fig. 12b). However, the model underestimates the observed range of variability: while observations vary over almost 2 orders of magnitude, simulated biomasses vary only over 1 order of magnitude. The model also overestimates biomass for the North American east coast (NA–E).

These biases might reflect temporal offsets in the depletion of fish biomass over time due to exploitation (see also Fig. 4d), with the model failing to capture relative differences across LMEs. Indeed, regions where the model overestimates fish biomass still have relatively high simulated rates of biomass decline in the 2000s (e.g., LMEs 1, 5, 12). This can eventually deplete biomass to the observed levels. Conversely, regions where the model underestimates fish biomass (e.g., LMEs 10, 14, 60) are areas in which simulated fishing effort caused an early biomass decline (see Appendix I). These temporal mismatches could reflect regional differences in the rate of development of fisheries that are not captured by the simple globally homogeneous exponential

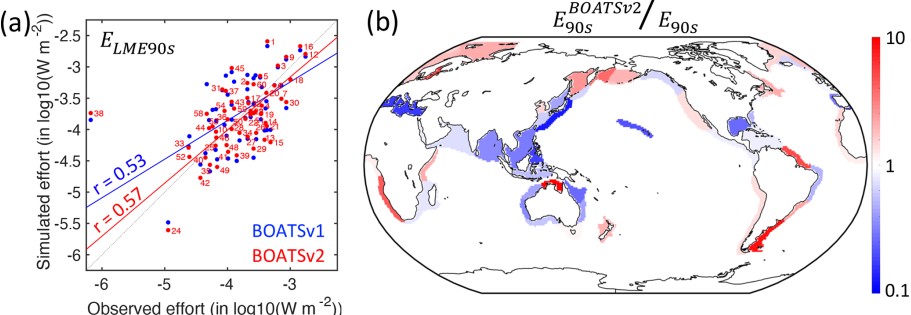

**Figure 11.** Observed and simulated nominal effort in the 1990s, $E_{90s}$. **(a)** Simulated effort density per LME compared to observations, $E_{\mathrm{LME90s}}$. **(b)** Ratio between modeled and observed effort by LME at the global catch peak in the 1990s. In panel **(a)**, red dots and lines show BOATSv2 output, and blue dots and lines show BOATSv1 output; numbers indicate the LMEs (see Appendix E).

increase in technology or by the open-access assumption and lack of management. For example, effective management in Alaskan fisheries has prevented the phase of overfishing that has been common in industrial fisheries worldwide (Worm et al., 2009). Alternatively, considering biodiversity could help explain these differences. For instance, in NA–W, the dominance of semi-pelagic Alaskan pollock may lead to an underestimation of our exclusively demersal biomass. Conversely, in NA–E, shifts from demersal to pelagic communities due to fishing can explain the overestimation of demersal biomass (Choi et al., 2004). Our approach does not capture these interactions between pelagic and demersal communities. Finally, discrepancies may be exacerbated because observations cover only a portion of each LME (compare mean biomass densities at LME level vs. grid cells where simulations overlap with observations in Appendix I).

Aggregated biomass observations for pelagic stocks are scarcer than for demersal stocks. We compare model output with scientific trawl data for coastal pelagic species in the California Current and Gulf of Alaska (shown by squares in Fig. 12b) (Zwolinski et al., 2012). The model simulates overall higher biomass densities than observed, showing a wider ranges of values. Similar to observations, simulated pelagic biomass densities are lower compared to demersal biomass. A caveat to this comparison is that estimates of pelagic biomass remain significantly uncertain due to challenges in sampling the three-dimensional oceanic environment, variability and aggregation in fish populations, uncertainty in sampled depth ranges, net avoidance by pelagic fish, and the limited selectivity of pelagic trawls (Kaartvedt et al., 2012; Zwolinski et al., 2012). In addition, fish migrations can redistribute fish biomass across life stages in ways not captured by the model.

### 5.5   Implications of the model update

The inclusion of distinct energy pathways and spatially variable economic drivers in BOATSv2 has a limited impact on the evolution of coastal fisheries over time but has a

large impact on simulated high seas fisheries. All else being equal, BOATSv1 and BOATSv2 ensembles show very similar LME-level catch at the global peak (see $C_{\mathrm{LME90s}}^{\mathrm{SAU\text{-}WAT}}$ Fig. 5c, d) and comparable progression across LMEs from 1950 through 2000. Both are in good agreement with observations (Fig. 13). The key improvement of BOATSv2 is the representation of high seas fisheries, where catches are delayed and greatly reduced during the historical period, bringing the model closer to observations (see dashed lines in Fig. 13). However, as fisheries keep developing, BOATSv2 still overestimates fishing in the high seas (compare the dashed red line with observations in Fig. 13 or the increasing trend in simulations in Fig. 5a). This discrepancy suggests either an improper representation of the historical rate of catchability increase in the simulations or missing mechanisms, such as horizontal migrations that redistribute biomass from the high seas to the coast.

Finally, the separation between pelagic and demersal communities influences fish production rates because these communities respond to different environmental drivers (van Denderen et al., 2021; Fredston et al., 2023). Compared to BOATSv1, this change could influence the resilience of fisheries to fishing and/or climate change. It could also alter the response to regulation, although we anticipate similar dynamics as in previous work (Scherrer and Galbraith, 2020; Scherrer et al., 2020). A separation of pelagic and demersal energy pathways is likely to impact the effects of fish on biogeochemistry (Bianchi et al., 2021; Le Mézo et al., 2022).

### 6   Conclusions

We introduce BOATSv2, an expanded version of the BOATS model that includes multiple added features, including separation of demersal from pelagic communities, and improves simulation of high seas fisheries. New model features have a limited impact in coastal regions so that BOATSv2 simulates dynamics and variability in catch and biomass over LMEs that are similar to BOATSv1. The expanded represen-

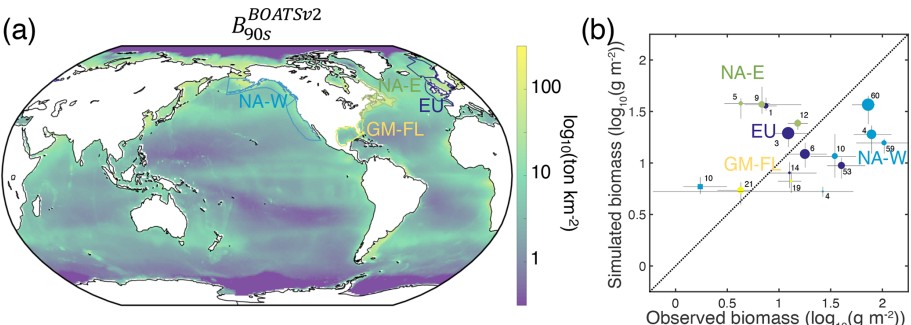

**Figure 12.** Observed and simulated biomass density. **(a)** BOATSv2 global mean biomass density in the 1990s, $B_{90s}$. **(b)** Demersal biomass density from bottom trawl surveys (circles) versus simulations and pelagic biomass density from surface coastal pelagic species trawl data (squares) versus simulations. Biomass densities are averaged across LMEs for the reference decade of the 2000s. In panel **(b)**, colors indicate the four different regions shown by the LME boundaries in panel **(a)** and numbers the specific LMEs (see Appendix E). In panel **(b)**, the dots indicate median values, and horizontal or vertical lines the 25–75th percentile range; the size of the dots indicates the relative size of the surface area sampled. Appendix I provides a further comparison of biomass time series by LME.

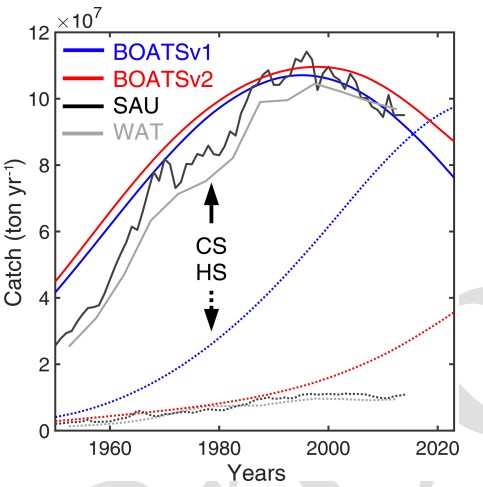

**Figure 13.** Time series of catch in the global ocean. Historical annual catch in the high seas (HS) vs. coastal seas (CS). Gray lines show observational reconstructions from Pauly et al. (2020) (SAU, dark gray) and Watson (2017) (WAT, light gray). Blue lines show output from BOATSv1 and red lines from BOATSv2, forced only with exponentially increasing catchability over time. Coastal (solid lines) and high seas (dotted lines) catches are shown separately.

tation of functional and taxonomic diversity allows more detailed comparisons with observations. In some cases, this reveals new model biases, such as in the simulation of demersal catches and biomass in the western North Atlantic.

We attribute improvements in the simulation of high seas fisheries to the separation of pelagic and demersal energy pathways, supporting the importance of distinguishing these communities (Blanchard et al., 2012; Petrik et al., 2019; Du Pontavice et al., 2020). We also introduced parameterizations of spatially heterogeneous economic drivers, i.e., fishing costs and catchability, which further improves the match to observations in the high seas. However, choosing between different formulations for these drivers (i.e., depending on distance from the coast, $d_{coast}$, or depth of the seafloor, $z_{bot}$) was only possible by testing plausible functional forms and retaining those leading to the largest improvements against empirical data. While this selection was not exhaustive, our final formulation is consistent with a variety of new observational constraints, such as the increase in fishing effort over seamounts (Kerry et al., 2022) and the historical deepening of fishing as technology progresses (Watson and Morato, 2013). We acknowledge that the specific choice of functional forms for these parameterizations is not well constrained and will likely require future refinement against observational diagnostics.

Because of the more accurate representation of high seas fisheries in BOATSv2 relative to BOATSv1, the fraction of catch that takes place in the High Seas at the time of the global catch peak is reduced from 31 % to 11 %, bringing it closer to the observed 8 %–9 %. Similarly, the mean depth of the catch declines from 1698 m in BOATSv1 to 420 m in BOATSv2, aligning it more closely with the empirical estimate of 154–372 m. This update should help reduce model uncertainties in future projections (Galbraith et al., 2017; Lotze et al., 2019; Tittensor et al., 2021) and provide a more accurate representation of the role of fish in global biogeochemical cycles (Bianchi et al., 2021; Le Mézo et al., 2022). Future model improvements could include a representation of the migration of fish stocks (Sumaila et al., 2015), the role of diverse coastal environments such as mangroves, reefs, and lagoons (Tittensor et al., 2010), and a representation of distinct mesopelagic communities (Irigoien et al., 2014; St. John et al., 2016; Hidalgo and Browman, 2019).

## Appendix A: BOATSv2 governing equations and parameters

**Table A1.** List of ecological and economic parameters in BOATSv2 for pelagic (P), demersal (D), or both communities. Parameters selected by the Monte Carlo procedure are reported by the range of best selected values. See more details in Carozza et al. (2016, 2017).

| Parameter | Name | Value (ranges in brackets) | Unit |
|---|---|---|---|
| $m_0$ | Lower bound of smallest mass class | 10 | g |
| $m_u$ | Upper bound of largest mass class | 100 000 | g |
| $n_k$ | Number of fish size groups | 3 | Unitless |
| $m_{\infty,k}$ | Asymptotic mass of group $k$ | (0.3, 8.5, 100) | kg |
| $T_r$ | Reference temperature of $a(T)$ | $10 + 273.15$ | K |
| $k_B$ | Boltzmann's constant | $8.617 \times 10^{-5}$ | $eV\,K^{-1}$ |
| $\omega_{a,A}$ | Growth activation energy of metabolism | P: [0.30, 0.47]; D: [0.12, 0.33] | eV |
| $\omega_{a,\lambda}$ | Mortality activation energy of metabolism | P: [0.39, 0.53]; D: [0.49, 0.61] | eV |
| $b$ | Allometric scaling exponent | [0.70,0.80] | Unitless |
| $A_0$ | Allometric growth constant | [4.35,5.04] | $g^{1-b}\,s^{-1}$ |
| $\epsilon_a$ | Activity fraction | 0.8 | Unitless |
| $c_s$ | Slope parameter of $s_k$ | 5 | Unitless |
| $\eta$ | Ratio of mature to asymptotic mass | 0.25 | Unitless |
| $\alpha$ | Trophic efficiency | [0.09,0.14] | Unitless |
| $\beta$ | Predator to prey mass ratio | [2830,8890] | Unitless |
| $\tau$ | Trophic scaling | $[-0.27, -0.23]$ | Unitless |
| $m_L$ | Mass of large phytoplankton | $4 \times 10^{-6}$ | g |
| $m_S$ | Mass of small phytoplankton | $4 \times 10^{-15}$ | g |
| $k_E$ | Eppley constant for phytoplankton growth* | 0.06 | $°C^{-1}$ |
| $\Pi^*$ | Nutrient concentration* | 0.35 | $mmol\,C\,m^{-3}\,d^{-1}$ |
| $m_\beta$ | Representative mass of benthos | $[8.3 \times 10^{-7}, 1.6 \times 10^{-6}]$ | g |
| $b_a$ | Martin curve attenuation coefficient | $-0.8$ | Unitless |
| $z_{eu}$ | Reference euphotic layer depth | 75 | m |
| $k_{NO_3^-}$ | Nitrate concentration constant | 5 | μM |
| $\zeta_1$ | Mortality constant | $[-0.68, -0.07]$ | Unitless |
| $h$ | Allometric mortality scaling | [0.46, 0.55] | Unitless |
| $\phi_f$ | Fraction of females | 0.5 | Unitless |
| $\phi_{C,k}$ | Fraction of NPP allocated to a group $k$ | 1/3 | Unitless |
| $s_e$ | Eggs to recruit survival fraction | [0.024, 0.093] | Unitless |
| $m_e$ | Egg mass | $5.2 \times 10^{-4}$ | g |
| $\kappa_e$ | Fleet dynamics parameter | $10^{-6}$ | $W\,USD^{-1}\,s^{-1}$ |
| $\kappa_s$ | Regulation response parameter | $4 \times 10^8$ | $m^2\,s^{-1}$ |
| $S$ | Societal enforcement strength (here deactivated) | 0 | Unitless |
| $c_\sigma$ | Fishing selectivity slope | 17.8 | Unitless |
| $d_{m_{\Theta,k}}$ | Selectivity mass adjustment | (1, 1, 1) | Unitless |
| $e_{m_{\Theta,k}}$ | Selectivity mass scaling | [0.44, 0.97] | Unitless |
| $(\delta_z, \delta_d)$ | Rate of cost increase with depth − distance | $(2.5 \times 10^{-3}, 7.9 \times 10^{-3})$ | $(USD\,m^{-1}\,W^{-1}\,yr^{-1}, USD\,km^{-1}\,W^{-1}\,yr^{-1})$ |
| $(z_{ref}, d_{ref})$ | Reference variables for cost profiles | (200, 370) | (m,km) |
| $q_{min}$ | Minimum gear efficiency | 0.8 | Unitless |
| $(z_{mean}, z_{max})$ | Reference depths for catchability profile | (2372, 5750) | m |

* Estimation of the fraction of large phytoplankton production following Dunne et al. (2005).

**Table A2.** Variables and governing equations for the ecological module of BOATSv2 for pelagic (P), demersal (D), or both communities. See more details in Carozza et al. (2016, 2017).

| Variable and functions | Formulation | Unit |
| --- | --- | --- |
| Size (mass) of fish | $m$ | g |
| Time | $t$ | s |
| Temperature | $T(t) \begin{cases} \text{P}: & T_{75} \\ \text{D}: & T_{\text{bot}} \end{cases}$ | K or °C |
| Surface nitrate concentration | $NO_3(t)$ | µM |
| Net primary production | $\Pi_\psi(t)$ | $\text{mmol C m}^{-2}\,\text{s}^{-1}$ |
| Bathymetry | $z_{\text{bot}}$ | m |
| Fraction of large phytoplankton production[1] | $\Phi_L(t)$ | – |
| Particle export ratio[1] | $\text{pe}_{\text{ratio}}(t)$ | – |
| Fish biomass spectrum of group $k$ | $f_k(m,t)$ | $\text{g m}^{-2}\,\text{g}^{-1}$ |
| Cumulative biomass of group $k$ | $B_k(t) = \int_{m_0}^{m_{\infty,k}} f_k\,dm$ | $\text{g m}^{-2}$ |
| Fish catch spectrum of group $k$ | $h_k(m,t)$ | $\text{g m}^{-2}\,\text{g}^{-1}\,\text{s}^{-1}$ |
| McKendrick von Foerster model | $\frac{\partial f_k}{\partial t} = -\frac{\partial \gamma_{S,k} f_k}{\partial m} + \frac{\gamma_{S,k} f_k}{m} - \Lambda_k f_k - h_k$ | – |
| Recruitment at $m = m_0$ | $\gamma_{S,k} f_k = R_{P,k} \frac{R_{e,k}}{R_{P,k}+R_{e,k}}$ | $\text{g m}^{-2}\,\text{s}^{-1}$ |
| Individual growth rate | $\gamma_{S,k} = (1-\Phi_k)\xi_{I,k}$ | $\text{g s}^{-1}$ |
| Fraction of input energy allocated to growth | $\Phi_k = s_k \frac{1-\epsilon_a}{(m/m_{\infty,k})^{(b-1)}-\epsilon_a}$ | – |
| Individual level total energy input | $\xi_{I,k} = \min\left[\frac{\phi_{C,k}\pi m}{f_k}, Am^b - k_a m\right]$ | $\text{g s}^{-1}$ |
| Growth constant | $A = A_0 a_A(T)$ | $\text{g}^{1-b}\,\text{s}^{-1}$ |
| Mass-specific investment in activity | $k_a = A\epsilon_a m_{\infty,k}^{b-1}$ | $\text{s}^{-1}$ |
| Fish production spectrum | $\pi = \begin{cases} \text{P}: & \frac{\Pi_\psi}{m_\psi}\left(\frac{m}{m_\psi}\right)^{\tau-1} \\ \text{D}: & \frac{\Pi_\beta}{m_\beta}\left(\frac{m}{m_\beta}\right)^{\tau-1} \end{cases}$ | $\text{g m}^{-2}\,\text{g}^{-1}\,\text{s}^{-1}$ |
| Representative mass of phytoplankton | $m_\psi = m_L^{\Phi_L} m_S^{1-\Phi_L}$ | g |
| Particle flux at bottom | $\Pi_\beta = \Pi_\psi\,\text{pe}_{\text{ratio}}\left(\frac{z_{\text{bot}}}{z_{\text{eu}}}\right)^{b_a}$ | $\text{mmol C m}^{-2}\,\text{s}^{-1}$ |
| Mass structure of energy to reproduction | $s_k = \left[1 + \left(\frac{m}{m_{\alpha,k}}\right)^{-c_s}\right]^{-1}$ | – |
| Mass of maturity | $m_{\alpha,k} = \eta\,m_{\infty,k}$ | g |
| Natural mortality rate | $\Lambda_k = \lambda m^{-h} m_{\infty,k}^{h+b-1}$ | $\text{s}^{-1}$ |
| Mortality constant | $\lambda = e^{\zeta_1}\left(\frac{A_0}{3}\right) a_\lambda(T)$ | $\text{g}^{1-b}\,\text{s}^{-1}$ |
| Primary-production-determined recruitment | $R_{P,k} = \phi_{C,k}\pi(m_0)m_0$ | $\text{g m}^{-2}\,\text{s}^{-1}$ |
| Egg-production-determined recruitment | $R_{e,k} = \phi_f s_e \frac{m_0}{m_e}\int_{m_0}^{m_{\infty,k}} \gamma_{R,k}(m)\frac{f_k(m)}{m}dm$ | $\text{g m}^{-2}\,\text{s}^{-1}$ |
| Energy allocated to reproduction | $\gamma_{R,k} = \Phi_k \xi_{I,k}$ | $\text{g s}^{-1}$ |
| van't Hoff–Arrhenius equation | $a_{a,\lambda}(T) = exp\left[\frac{\omega_{a,\lambda}}{k_B}\left(\frac{1}{T_r} - \frac{1}{T}\right)\right]$ | – |
| Corrected trophic scaling[2] | $\tau = \frac{ln\left(\alpha\,\frac{k_{NO_3^-}}{k_{NO_3^-}+NO_3^-}\right)}{ln(\beta)}$ | – |

[1] Estimated from net primary production and surface temperature (in °C) following Dunne et al. (2005). [2] Correction of trophic scaling when reduced growth in iron-limited regions is activated.

**Table A3.** Variables and governing equations for the economic module of BOATSv2 for pelagic (P), demersal (D), or both communities. See more details in Carozza et al. (2016, 2017).

| Variable and functions | Formulation | Unit |
|---|---|---|
| Fish catchability of group $k$ | $q_k(t)$ | $\mathrm{m}^2\,\mathrm{W}\mathrm{s}^{-1}$ |
| Ex-vessel fish price for group $k$ | $p_k(t)$ | $\mathrm{USD}\,\mathrm{g}^{-1}$ |
| Cost per unit effort for group $k$ | $c_k(t)$ | $\mathrm{USD}\,\mathrm{W}^{-1}\,\mathrm{s}^{-1}$ |
| Bathymetry | $z_{\mathrm{bot}}$ | m |
| Distance to shore | $d_{\mathrm{coast}}$ | km |
| Societal target for fishing effort[1] | $E_{\mathrm{targ},k}(t)$ | $\mathrm{W}\,\mathrm{m}^{-2}$ |
| Fish catch spectrum of group $k$ | $h_k(m,t)$ | $\mathrm{g}\,\mathrm{m}^{-2}\,\mathrm{g}^{-1}\,\mathrm{s}^{-1}$ |
| Cumulative catch of group $k$ | $C_k(t) = \int_{m_0}^{m_{\infty,k}} h_k \, dm$ | $\mathrm{g}\,\mathrm{m}^{-2}\,\mathrm{s}^{-1}$ |
| Fishing effort of group $k$ | $E_k(t)$ | $\mathrm{W}\,\mathrm{m}^{-2}$ |
| Fishing effort model | $\frac{dE_k}{dt} = \left(\kappa_e \frac{\mathrm{revenue}_k - \mathrm{cost}_k}{E_k}\right)e^{-S} + (1 - e^{-S})\kappa_S(E_{\mathrm{target},k} - E_k)$ | - |
| Revenue from fishing | $\mathrm{revenue}_k = q_k \, E_k \, dt \int_{m_0}^{m_{\infty,k}} p_k \sigma_k(m) f_k(m) \, dm$ | $\mathrm{USD}\,\mathrm{m}^{-2}\,\mathrm{s}^{-1}$ |
| Size-dependent selectivity of catch | $\sigma_k = \left[1 + \left(\frac{m}{m_{\Theta,k}}\right)^{-c_\sigma/3}\right]^{-1}$ | – |
| Threshold mass for catch | $m_{\Theta,k} = d_{m_\Theta,k} \, e_{m_\Theta,k} \, m_{\alpha,k}$ | g |
| Cost of fishing | $\mathrm{cost}_k = c_k \, E_k \, dt$ | $\mathrm{USD}\,\mathrm{m}^{-2}\,\mathrm{s}^{-1}$ |
| Corrected depth-dependent cost profile[2] | $c_k(z_{\mathrm{bot}}) = c_k + \delta_z(z_{\mathrm{bot}} - z_{\mathrm{ref}})$ | $\mathrm{USD}\,\mathrm{W}^{-1}\,\mathrm{s}^{-1}$ |
| Corrected distance-dependent cost profile[2] | $c_k(z_{\mathrm{dist}}) = c_k + \delta_d(d_{\mathrm{coast}} - d_{\mathrm{ref}})$ | $\mathrm{USD}\,\mathrm{W}^{-1}\,\mathrm{s}^{-1}$ |
| Corrected depth-dependent catchability profile[2] | $q_k(z_{\mathrm{bot}}) = q_k \, \mathrm{Pr}(z_{\mathrm{bot}}) \, \mathrm{Of}(P, D)$ | $\mathrm{m}^2\,\mathrm{W}\mathrm{s}^{-1}$ |
| Depth-dependent catchability weight | $\mathrm{Pr}(z_{\mathrm{bot}}) = q_{\mathrm{min}} + (1 - q_{\mathrm{min}})\frac{\log_{10}(z_{\mathrm{max}}) - \log_{10}(z_{\mathrm{bot}})}{\log_{10}(z_{\mathrm{max}}) - \log_{10}(z_{\mathrm{mean}})}$ | – |
| Catchability offset between communities | $\mathrm{Of}(P) = 1.4\,\mathrm{Of}(D)$ | – |

[1] Not detailed in the present model description; see Scherrer and Galbraith (2020). [2] Correction of catchability or cost when spatial economic parameterization is activated.

## Appendix B: Global variability in surface nitrate

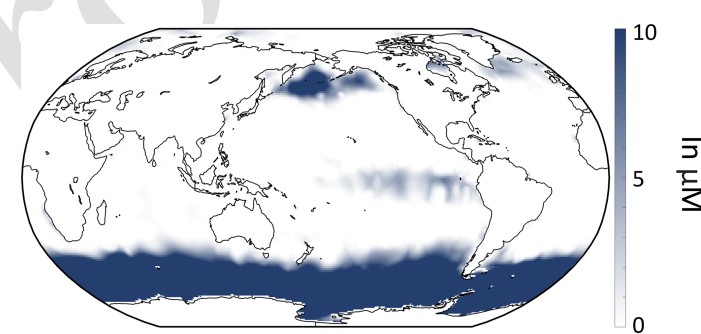

**Figure B1.** Minimum monthly sea surface nitrate concentration (in µM) from the World Ocean Atlas (Locarnini et al., 2006).

## Appendix C: Spatial variation of costs

The cost of fishing varies by fishing gear and by targeted fish community (Lam et al., 2011). To best constrain spatially variable costs we use estimates of these separate fishing costs in the high seas for the main gear types (98 % of total effort) following data reported by Sala et al. (2018). Table C1 summarizes these estimated costs. These compare with the BOATS default fishing cost of 5.85 USD W$^{-1}$ yr$^{-1}$ (Carozza et al., 2017; Galbraith et al., 2017). We defined spatially variable costs as a function of distance to the coast $d_{\mathrm{coast}}$ and depth of the seafloor $z_{\mathrm{bot}}$. Figure C1a illustrates the profile of distance-dependent costs and Fig. C1b the profile of depth-dependent costs (in USD W$^{-1}$ yr$^{-1}$).

**Table C1.** Cost of fishing the high seas based on estimates from Sala et al. (2018) for the year 2016.

| Gear type | Effort in kWh (fraction of total) | Cost range in USD | Cost per unit effort in USD W$^{-1}$ yr$^{-1}$ |
|---|---|---|---|
| Trawlers | $979 \times 10^6$ (15 %) | $[750 \times 10^6 – 1030 \times 10^6]$ | $[6.7–9.2]$ |
| Longliners | $3719 \times 10^6$ (55 %) | $[2523 \times 10^6 – 3023 \; 10^6]$ | $[6.0–7.1]$ |
| Purse seiners | $394 \times 10^6$ (6 %) | $[702 \times 10^6 – 1188 \times 10^6]$ | $[15.7–26.0]$ |
| Squid jiggers | $1490 \times 10^6$ (22 %) | $[1308 \times 10^6 – 1616 \times 10^6]$ | $[7.7–9.5]$ |
| Range for all gears | (98 %) | – | $[6.94–8.87]$ |
| BOATS default | – | – | 5.85 |

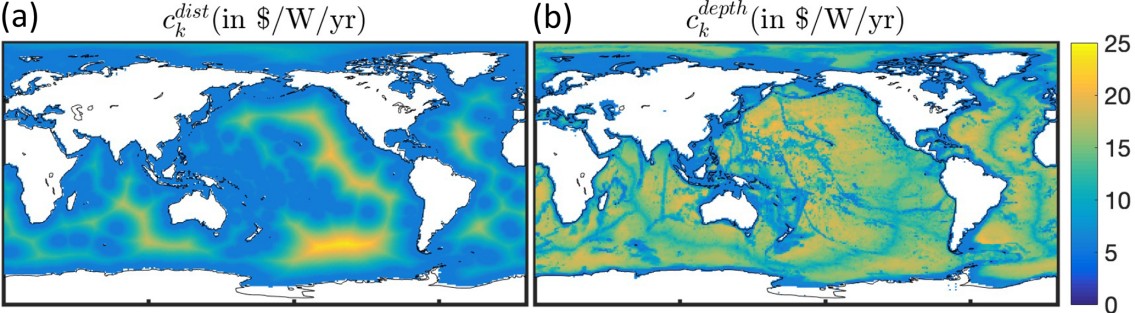

**Figure C1.** Cost per unit effort profiles $c_k^{\mathrm{coor}}$ in the global ocean (in USD W$^{-1}$ yr$^{-1}$) **(a)** as a function of distance to the nearest coast $c_k^{\mathrm{coor}} = c_k^{\mathrm{dist}}$ and **(b)** as a function of depth of the seafloor $c_k^{\mathrm{coor}} = c_k^{\mathrm{depth}}$.

## Appendix D: Spatial variation of catchability

Technology coefficients vary between gears (Palomares and Pauly, 2019), and gears are predominantly used in separate regions of the global ocean (Kroodsma et al., 2018; Kerry et al., 2022). Ultimately this can lead to spatially heterogeneous catchability of fish resources.

In order to better constrain the catchability, we use the reported difference of technology coefficients by gear estimated in 1995 and a coarse estimation of the contribution of each gear to the global fishing effort from 2015 through 2020 as reconstructed by Global Fishing Watch (GFW; see Kroodsma et al., 2018). Depending on the functional type predominantly targeted by a gear, pelagic vs. demersal, we estimate the mean technology coefficients for pelagic species to be $Of(P) = 1.3$ compared to $Of(D) = 0.9$ for demersal species (see Table D1).

Based on the observation that a dominant part of fishing effort on pelagic species by longliners occurs near seamounts (Kerry et al., 2022), we adjust the spatial catchability as a function of the depth of the seafloor such that it varies from a minimum of $Pr(z_{bot}) = 0.8$ over deep seafloors (e.g., for tuna seiners; Table D1) to $Pr(z_{bot}) = 2.4$ in shallow regions (such that the global mean is 1.3). Figure D1 illustrates the reference profile of technology coefficients $Pr(z_{bot})$ used for the analysis.

**Table D1.** Technology coefficient per fish community. The coefficients per gear are based on reported values in Palomares and Pauly (2019). Each gear is linked to the dominant resource it targets, pelagic (Pel) or demersal (Dem), and associated with the fraction of global fishing effort from 2015 through 2020, as reported by Global Fishing Watch. We reported the mean technological coefficient weighted with effort by gear, when available.

| Gear type (fraction of GFW effort) | Dominant target (Pel vs. Dem) | Technology coefficient 1995 (normalized) |
|---|---|---|
| Super trawlers (–) | Pel | 1.3 |
| Tuna seiner (1.1 %) | Pel | 0.8 |
| Freeze trawler (–) | Pel | 1.0 |
| Tuna longliner (–) | Pel | 1.2 |
| Purse seiner (2.2 %) | Pel | 1.0 |
| Stern trawler (–) | Pel/Dem | 1.0 |
| Longliner (19 %) | Pel | 1.4 |
| Multipurpose vessel (–) | Pel/Dem | 1.3 |
| Shrimp trawler (–) | Dem | 1.1 |
| Trawler (48 %) | Dem | 0.9 |
| Gill netter (6 %) | Dem | 0.8 |
| Fast potter (0.7 %) | Dem | 0.7 |
| Other (23 %) | Pel/Dem | – |
| Mean pelagic (22 %) | Pel | 1.3 |
| Mean demersal (55 %) | Dem | 0.9 |

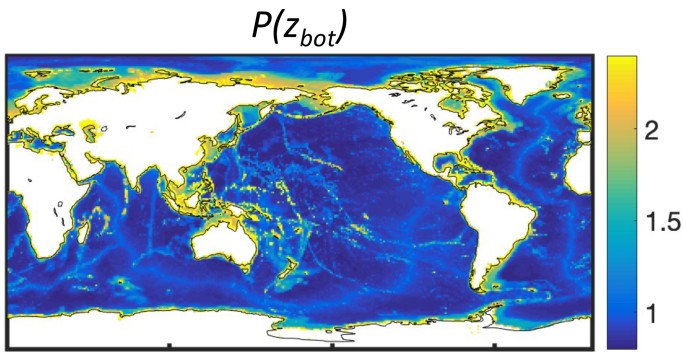

**Figure D1.** Relative technology coefficient profiles $\mathrm{Pr}(z_{\mathrm{bot}})$ in the global ocean.

## Appendix E: Large marine ecosystems and high seas ecosystems

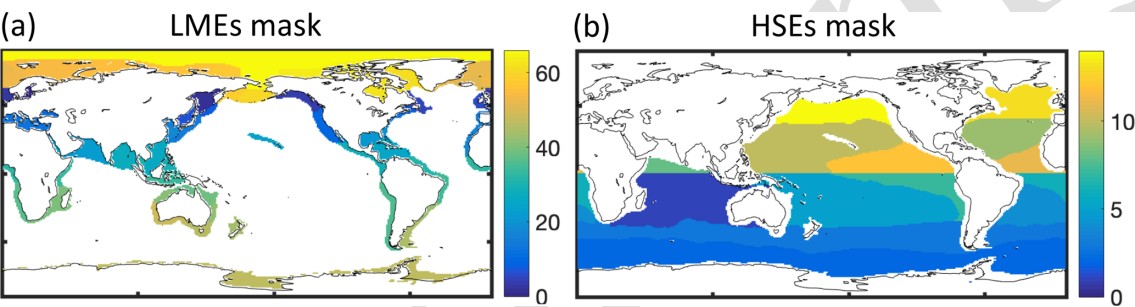

**Figure E1.** Regional masks used to compare observations and simulations. **(a)** Large marine ecosystems. **(b)** High seas ecosystems adapted from Weber et al. (2016).

## Appendix F: Pelagic and demersal catches

**Table F1.** The Sea Around Us (Pauly et al., 2020) functional types associated with pelagic and demersal catches.

| Catch type | SAU functional types |
| --- | --- |
| Pelagic | pelagic s/m/l<br>bathypelagic s/m/l<br>cephalopods |
| Demersal | demersal s/m/l<br>reef-associated s/m/l<br>benthopelagic s/m/l<br>bathydemersal s/l<br>shark s/l<br>flatfish s/l<br>ray s/l<br>shrimp<br>lobster and crab<br>other demersal invertebrates |

## Appendix G: Global catch distribution between BOATS versions

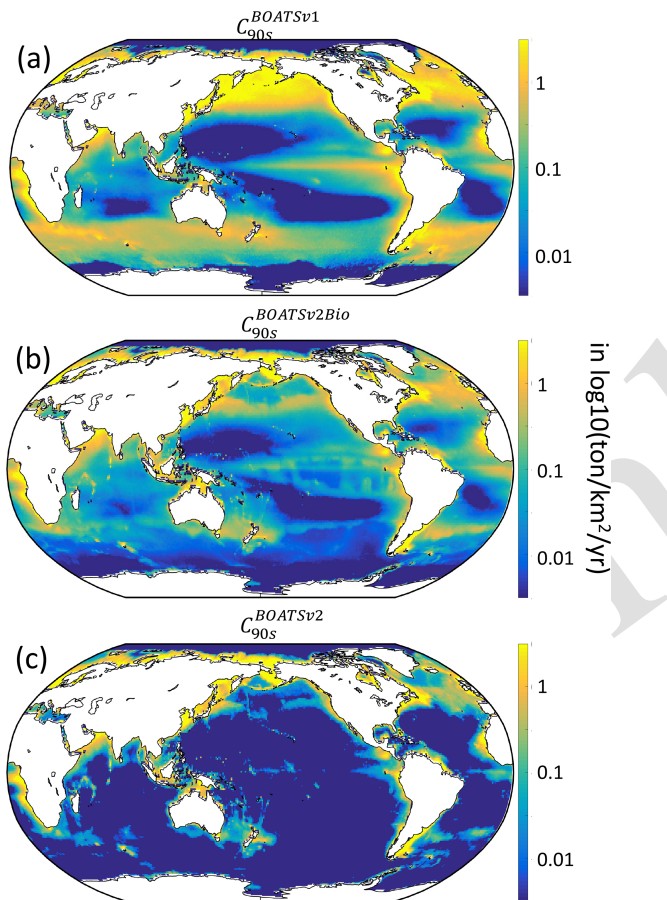

**Figure G1.** Simulated catch in the 1990s, $C_{90s}$ (in $\log_{10}$; $\mathrm{t\,km^{-2}\,yr^{-1}}$). **(a)** BOATSv1. **(b)** Updated version with improved ecology – BOATSv2-Bio. **(c)** Final update including improved economics – BOATSv2.

## Appendix H: Historical catch deepening

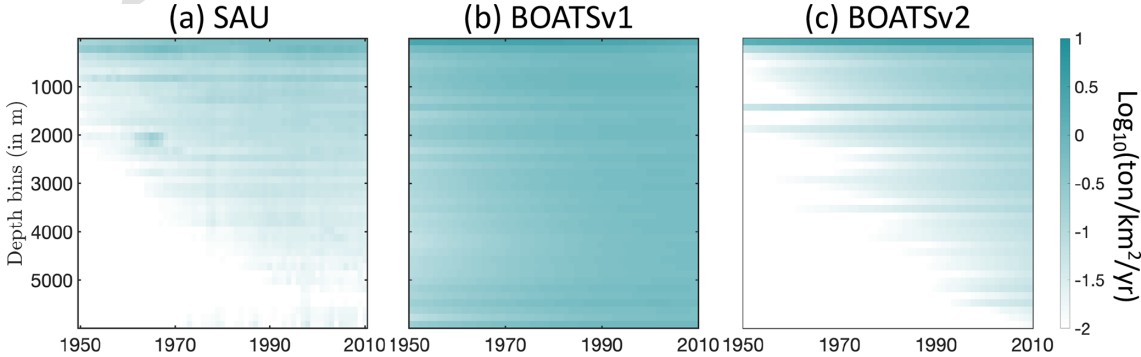

**Figure H1.** Fishing over increasingly deep seafloors. **(a)** Observed and **(b, c)** simulated mean total catch density (in $\log_{10}$; $\mathrm{t\,km^{-2}\,yr^{-1}}$) over depth strata. Compared to observations **(a)**, BOATSv1 **(b)** fails to capture the deepening, while BOATSv2 corrects it.

## Appendix I:  Historical biomass variation in selected LMEs

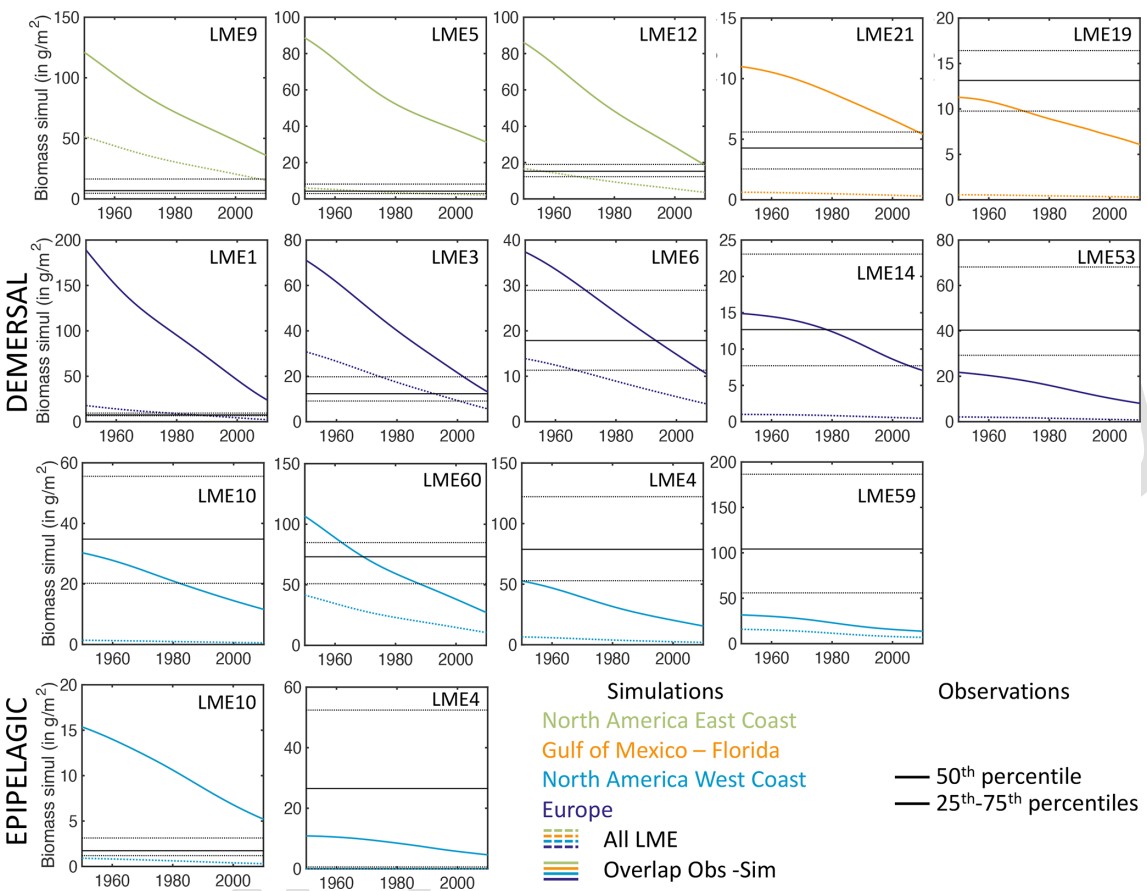

**Figure I1.** Observed and simulated biomass decline in LMEs for demersal and pelagic biomass. Each panel shows the simulated historical fish biomass density (in g m$^{-2}$) decline, averaged across the selected LME (dotted line) or averaged over 1° grid cells where observations are available (plain line). These are compared to the range of observed biomass density per LME over years in the 2000s, indicated by the median value (plain black lines) and the 25th and 75th percentiles (dotted black lines). Colors indicate neighboring LMEs, North American LMEs along the east coast (green), North American LMEs along the west coast (light blue), the Gulf of Mexico and Florida LMEs (in orange), and European LMEs (dark blue).

*Code and data availability.* The code of the model, forcing to complete reference simulations, and observations to assess the model are available through Zenodo (DOI: https://doi.org/10.5281/zenodo.11043334, Guiet et al., 2024).

*Author contributions.* JG, DB, KJNS, RFH, and EDG discussed and designed the new model developments and analysis. JG implemented the code updates and merged past developments from KJNS, DB, and EDG. JG performed the simulations and analysis. JG wrote the first draft and prepared the figures and tables in collaboration with DB. All authors provided critical feedback on the results and contributed to the final manuscript.

*Competing interests.* The contact author has declared that none of the authors has any competing interests.

ther geographical representation in this paper. While Copernicus Publications makes every effort to include appropriate place names, the final responsibility lies with the authors.

*Acknowledgements.* Daniele Bianchi and Jerome Guiet acknowledge support from the US National Aeronautics and Space Administration (NASA). Computational resources were provided by the Expanse system at the San Diego Supercomputer Center from the Advanced Cyberinfrastructure Coordination Ecosystem Services and Support (ACCESS) program, which is supported by National Science Foundation. Eric D. Galbraith was supported by the Canada Research Chairs Program. Kim J. N. Scherrer was supported by the Norwegian Research Council.

*Financial support.* This research has been supported by the National Aeronautics and Space Administration (grant no. 80NSSC21K0420), the San Diego Supercomputer Center (grant no. TG-OCE170017), the National Science Foundation (grants nos. 2138259, 2138286, 2138307, 2137603, and 2138296), the Canada Research Chairs Program (grant no. CRC-2020-00108), and Norges Forskningsråd (grant no. 326896).

*Review statement.* This paper was edited by Andrew Yool and reviewed by two anonymous referees.

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

**Remarks from the language copy-editor**

CE1   Please note the slight edits (a comma would not be grammatically correct).

**Remarks from the typesetter**

TS1   Please confirm all affiliation codes and affiliations.
TS2   Please check table reference.
TS3   Please confirm table reference.
TS4   Please confirm.
TS5   Please confirm.
TS6   Please give an explanation of why this needs to be changed. We have to ask the handling editor for approval. Thanks.
TS7   Please confirm addition.
TS8   Please provide date of last access.
TS9   This should be the correct abbreviation for the journal (see https://images.webofknowledge.com/WOK46P9/help/WOS/ D_abrvjt.html)
TS10   Please check URL; does it lead to a different paper? Please provide date of last access.