# Peer review of "BOATSv2: New ecological and economic features improve simulations of High Seas catch and effort"

_Geoscientific Model Development, 2024_

## Author Comment (AC1)

**We thank both reviewers for their constructive comments. In the following our responses (**in blue**) to the reviews (**in black**). Our responses detail or include changes and additions to the text.**

**Answer to reviewer #1:**

Guiet et al. present recent developments in the BOATS global ecosystem model that improve ecological and economic processes to improve the fit of model results to biomass, catch and fishing effort data. BOATS is a size-spectrum global marine ecosystem model in which they mainly incorporated three new features: 1) a demersal guild in addition to the already existing pelagic guild, 2) a better representation of iron limitation and 3) a modified assumption of fish accessibility for fisheries including spatially variable fishing costs and catchability. The results show multiple improvements in the evaluation of various indicators, validating the necessity of these recent developments and the ability of the newly included mechanisms to reproduce the observed patterns.

I would like to emphasize the quality of the article, which is well written and overall clear on the description of the model's equations and assumptions. Well done to the authors, who have done a tremendous job in this respect. In addition, I very much appreciated the approach of developing and testing new mechanisms in the model by successive implementations. Thus, I think the article is timely as we need in marine modeling science to:

(i) Continue the development of existing models by including new robust mechanisms to understand and anticipate current and future human pressures on marine ecosystems

(ii) Have a clear description and documentation of model assumptions and equations to improve model usability, transparency and reproducibility. For these reasons, I fully support the authors' request to publish their paper in Geoscientific model development.

Thank you for this encouraging feedback, we perfectly agree.

I also very much appreciated the ingenuity of the model parameterization and the selection of the best set of parameters: their approach could be more widely used for other marine ecosystem models in a context where model validation and calibration methods are still under discussion and development.

The structure of the document is unusual in that it includes a section on the sensitivity analysis of the model to certain parameters between the "Materials and methods" section and the results section and presents some of the results of the sensitivity analysis in this section. The need for this information to better appreciate the results makes it acceptable.

Furthermore, the results and discussion are brought together in one, which I find very appropriate in this type of technical paper.

My main criticism concerns certain parts of the discussion. I found that some results for which the model does not perform well are never discussed, whereas a discussion accompanied by a hypothesis about potential missing processes in the model or experimental data could help validate the model and understand the gap that remains between the data and the model. One of the aims of the paper was to improve the representation of iron-limiting zones. Are the zones better represented in the iron-limited zone of Boatsv2? How do you explain the areas where the model performs less well? Similarly, when the model underperforms in v1 and v2, can you speculate on the reasons for this underperformance? Some global hypotheses are mentioned at the end of the discussion, but specific hypotheses for the highlighted area would be useful (North Atlantic, Eastern Pacific, etc.). Maybe other global ecosystem models (Apecosm, DBEM…) with different assumptions perform well in different region: it could be helpful to formalize. For more examples, see the detailed line-by-line commentary.

We understand these criticisms. In this revised version, we expanded the discussion of plausible mechanisms behind remaining model limitations, and potential approaches to address them, as these could be useful for other marine ecosystem modelers. See details in the following answers.

My second point concerns certain parameter settings. I noticed that some values/hypotheses were given without explanation. Even if they are assumed or derived empirically, I think it would be useful to specify them for greater clarity. See also the line-by-line commentary for more information.

Thanks. Several aspects regarding the parameter settings were less detailed because they were primarily discussed in previous BOATS publications. However, we understand that the reader might need this information while reading this manuscript, so we followed your advice and included more information; see the following.

Lastly, I found some minor typo error that needs to be corrected (see lines by lines)

To close, I wish to emphasize that I consider this research is already of great quality. My criticisms are simply intended to be helpful to developing/precising it.

Thanks again for your supportive feedback. See our point-by-point answer below.

Specific comments:

**L37: Add a source exploring multiple aspects of global fisheries dynamics**

**Changed.** This statement refers to all the analyses described and referenced in the previous paragraph. We reformulated the sentence to make it clearer and avoid repeating the references.

L38: "*These studies prove the usefulness of BOATS for exploring various aspects of global fisheries. Still, comparisons with observations…*"

**L39: the example of an ecosystem defined as HNLC could help non-specialists of this ecosystem to identify the type of ecosystem you are referring to.**

**Changed.** We added the following sentences and reference to clarify:

L40-44: "*For instance, high-nutrient low-chlorophyll (HNLC) regions are characterized by relatively low primary production despite available macronutrients (Moore et al. 2013). These regions represent more than one-quarter of the open ocean surface area, and include the Southern Ocean, the Eastern Equatorial Pacific, and the Subarctic North Pacific. In HNLC regions, comparison of simulated effort with global reconstructions suggested excessive fishing activity in BOATS, indirectly pointing to an excessive biomass accumulation in the model (Galbraith et al. 2019).*"

**L70: how is the vertical position of communities/captures estimated if the grid is 2D?**

**Changed.** The model is designed to represent epipelagic and demersal communities but does not explicitly represent vertical positions along the water column. For resources at the base of each community, we select vertically integrated NPP and sinking organic matter flux at the seafloor for epipelagic and demersal communities, respectively. As a first approximation, we keep these communities separate. To constrain metabolic rates, we use the mean temperature in the upper layers (top 75m) for the epipelagic community, where photosynthesis takes place and where many harvested fish live, and the bottom temperature for the demersal community. Both forcing sets are meant to characterize the mean habitat characteristics for each community, which we believe determines to first order the variability at the community level. We clarified this by adding the following:

L76-77: "*It uses vertically averaged habitat characteristics on a 2-dimensional spatial grid to simulate the variability of fish communities, from small regions to the global ocean.*"

L335-336: *"Forcing BOATS with 2-dimensional grids does not account for vertical positions along the water column but characterizes mean environmental conditions where many harvested fish live."*

**L80: First equation: The first term of the equation is a growth term, but it is negative. Is it the biomass that exceeds the group size threshold due to growth? If so, a sentence explaining this would be useful.**

**Changed.** The first term on the right-hand side is the divergence of the growth flux. (This can be thought as an advection of biomass in size space, i.e., growth transfers biomass to increasing sizes). For a given size class, if the flux leaving the size class is larger than the flux entering, this term is positive and needs to be turned into a sink to account for the decrease of biomass within the size class. The second term on the right-hand side is also an expression of growth that arises from writing the conservation of fish biomass in size space. Consider a fixed number N of fish that grow over time (i.e., exclude mortality) and reach a certain size class. While the number of individuals remains constant, their total biomass increases as they grow, hence the source term. We have clarified the contribution of each term:

L91-94: *"The first term in Eq. (1) represents the rate of change in time of the fish biomass spectrum for each group. The second term is the divergence of the growth flux, i.e., the transfer of biomass to increasing size as fish grow. The third term encapsulates the biomass accumulation due to the increase of individual size as fish grow. The fourth and fifth terms represent losses from natural mortality and catch respectively."*

**L86: If the minimum requirement is not met, this has no impact on mortality, why?**

**Changed.** The reviewer is correct: mortality is not affected by the availability of food for growth. In BOATS, when resources are abundant, we cap the maximum growth rate to account for the fact that growth cannot exceed a biological maximum rate. Otherwise, it is proportional to NPP and becomes negligible if NPP is very low. But this is independent of mortality. We now clarify:

L109-114: *"In BOATS, the growth rate at a given size occurs either at the maximum physiological rate when food is not limiting (gray area in the central panel Fig. 1) or proportionally to primary production $\Pi_\psi$ when food is limiting (green area in the central panel Fig.1). Accordingly, the growth rate is proportional to the minimum of two quantities: (1) the energy provided by primary producers that reaches a given size class, given trophic transfer across the food-web $\xi_{P,k}$ ,divided by the number of fish in that size class, and (2) the maximum production potential for a fish of that size, based on an individual-level allometric growth rate that follows a von Bertalanffy formulation $\xi_{VB,k}$ (in g s$^{-1}$)"*

L132-133: "The fish mortality is independent of variations of the growth rate.

L134-136: "The natural mortality rate (in units $s^{-1}$) depends on both individual and asymptotic mass, and represents biomass losses due to predation to organisms both within and outside of the resolved community size spectrum, as well as other natural causes. The natural mortality rate is based on the empirical parameterization of (Gislason et al., 2010; Charnov et al., 2013)."

Note that Section 2.1 has been largely rewritten following comments of reviewer 2.

**L95: "Primary production is equally distributed between groups". Why is this?**

**Changed.** A fixed partitioning allows coexisting fish groups; excess NPP, which would result from the growth limitation of one group, is not available to other groups and is assumed to be utilized by other communities not represented by the model, such as non-commercial mesopelagic fish, planktonic invertebrates, and microbial communities. But this parameterization also reflects the scarcity of appropriate data constraints; beyond fixed partitioning, how this NPP is distributed between species, equally or not, could be revised as we gain understanding.

L123-126: "*To ensure coexisting fish groups, and because of the scarcity of data available to constrain resource allocation, primary production is equally partitioned across the groups, i.e., $\varphi_{C,k} = 1/n_k = 1/3$ . While this is a first-order assumption that allows realistic simulation of catches by group, it should be revised as new observational constraints become available.*"

**L103 and 104: The typography of the letter "phi" is different in this line and the next than in the rest of the text.**

**Changed.** The typography differences ($\Phi_*$ or $\varphi_*$) reflect different parameters, and we selected these out of consistency with the previous publication detailing the governing equations of BOATSv1 (Carozza et al. 2016). Note that the description of recruitment has now been rewritten, it includes detailed equations and has been expanded (Eq. 5, L143-145).

**L107: Equation 3: same A0 as in the anabolism equation? If so, mention it either afterwards, or in the table of parameters.**

**Changed.** Yes, it is the same parameter as the growth constant A0, we clarified accordingly:

L138-139: "*…where h is an allometric scaling, and $\zeta_1$ (in g $s^{-1}$) a mortality rate parameter. As in Gislason et al. (2010), the natural mortality rate is linked to growth by means of the constants $A_0$ and b.*"

**L121-122: Do you think that using only the first 75 to estimate the temperature faced by the pelagic community is sufficient? Or could this be responsible for a bias in the representation of the community? If it is the second option, please discuss it**

**Changed.** Selecting the upper 75m of the water column is a reasonable assumption because it represents the average euphotic zone where net organic matter production takes place, and where many exploited pelagic fish spend most of their time, especially smaller, abundant forage species. Some larger predators (e.g., shark and billfish) experience cooler temperatures on average, since they occupy deeper habitats when feeding on the deep scattering layer during the day; however, given the current large uncertainty on the patterns and frequency of these vertical dives (Nuno et al. 2022, Braun et al. 2023) as well as the comparatively smaller biomass for large predators (Hatton 2021), we neglected these differences at this stage. We included a comment in the discussion to address this point.

L576-578: "*Finally, some larger predators that dive to feed on the deep scattering layer experience environmental conditions that differ from those at the surface (Nuno et al. 2022; Braun et al. 2023). Accounting for this effect could help reducing model biases.*"

**L136-138: I did not find the parameter em_o,k in the equation preceding (5). Explain why this information is given here or delete it.**

**Changed.** The description of this parameter here was unclear; it is one of the free parameters of the Monte Carlo ensemble. Hence, it is helpful to mention. We clarified the definition of the fishing selectivity function now reporting its equation (Eq. 8) and adding more description (L163-168):

L175-178 "*A variety of functional forms exist, all avoid the smallest sizes. These can be generalized as either dome-shaped (e.g. gillnets, longlines) or sigmoidal (e.g. trawls, seines or dredges). Here, we parameterize the selectivity as a sigmoidal curve around a target threshold mass $m_{\Theta,k} = dm_{\Theta,k}\, em_{\Theta,k}\, m_{\alpha,k}$ , essentially reducing the fishing effort targeting the smallest size classes:*"

L180-182: "*The target threshold mass is proportional to the maturity mass for each group $m_{\alpha,k}$, with the parameter $em_{\Theta,k}$ accounting for uncertainty around this mass and $dm_{\Theta,k}$ set to select mainly mature individuals (i.e., $dm_{\Theta,k}$ =1).*"

**L146: Since EK (t = 0) = 0, how do the dynamics of Ek begin?**

**Changed.** We set a lower limit on effort, epsilon=10^(-50), when calculating harvest, cost, and effort change. This prevents division by zero in Equation 6 and allows the development of fishing under open-access dynamics. We clarified:

L157-158: "*Fishing effort is typically initialized everywhere at negligible values, starting from an unfished ocean and evolves independently in each grid cell…*"

L193-194: "*Note that when computing catches, but also costs and effort change, we set a lower limit on effort $\varepsilon = 10^{-50}$ to allow the development of fishing and prevent division by zero in Eq. (6).*"

**L147: Table 1: Predator to prey mass ratio: this ratio is very high. Do you have a source that confirms this? How can the trophic scale parameter be interpreted biologically?**

**Changed.** Regarding the predator-to-prey mass ratio, we think that the values used are reasonable and in line with the literature. We recognize however that there is variability in this ratio, and predator-to-prey mass ratios tend to decrease for small organisms as compared to larger ones. For example, predator-to-prey mass ratios can be as low as 10 (for microzooplankton) and as high as 10,000 for large fish predators (Hansen et al. 2004, Barnes et al. 2010). There is also substantial variability for any given predator size class, e.g., depending on the feeding mode. For the fish size range considered here, the selected range of predator-to-prey mass ratios is aligned with observational estimates, such as Barnes et al. 2010. We added this reference when mentioning β L383.

The trophic scaling parameter indicates how efficiently energy is transferred through food webs. It encapsulates the efficiency with which energy and biomass propagate from primary producers to increasingly larger sizes and higher trophic levels, following the framework of the metabolic theory of ecology (Brown et al. 2004). We completed:

L120-122: "*The trophic scaling parameter determines the efficiency of propagation of production through the consumer size spectrum, to increasingly larger sizes and higher trophic levels, following the framework of the metabolic theory of ecology (Brown et al., 2004).*"

**L157: "qk increases annually at a rate of 5%" : Where does this value come from? Is it realistic?**

**Changed.** Empirical studies have estimated an average rate of increase of 2–8% /yr in diverse fisheries and periods. In previous studies (e.g., Galbraith et al., 2017), we tested a range of increase rates within this range when forcing other observed economic parameters (i.e., costs and price). The rate of increase in BOATS that accurately reproduces the observed development of global fisheries was found to be around 5% /yr  (see Galbraith et al., 2017, for model sensitivity to different rates). We recognize that there is substantial uncertainty in this quantity. However, adopting slightly different values in the range would not change the overall dynamics of fisheries, besides leading to faster or slower development. Given the reasonable match with observations (Fig. 4), we keep this rate for now. See also Scherrer et al. 2020 for more details on the role of catchability. We added a sentence to clarify:

L198-201: "*Empirical studies have estimated an average annual rate of 2-8% between fisheries and periods. We select an annual rate of 5% increase as, after testing when other observed economic parameters are forced, it accurately reproduces the historical development of fisheries with BOATS (Galbraith et al. 2017, Scherrer and Galbraith 2020).*"

Note that the increase in catchability is the only driver of historical variability in BOATS simulations. We stressed by clarifying results shown in figures 4, 7 and 13, by adding "… *forced only with exponentially increasing catchability over time*".

**L171: Why is the reduction in primary production not enough to explain the change in fish growth? Do you have any arguments in favor of a change in trophic efficiency? How do they explain the mechanism in Galbraith et al?**

**Changed.** Iron limitation reduces the growth of phytoplankton. The hypothesis behind including iron limitation for fish is that this limitation would also apply to fish and ultimately lead to significantly lower biomass in iron-limited regions.

Before correcting for iron limitation on fish, the model was already forced with observed primary production and thus included reduced primary production in iron-limited regions. However, this still led to excessive fish biomass and hence fishing effort in iron-limited regions. Therefore, lower primary production was not enough to explain the change in fish growth.

Instead, by observing Fe:C ratios in fish, Galbraith et al. show that fish's lack of adaptation to low iron regions prevents them from thriving in such environments, ultimately impacting biomass accumulation. In the model, a simple but effective way to account for this effect is to reduce trophic efficiency (i.e., the efficiency with which energy is transferred to higher trophic levels) in low iron regions (HNLC regions). This parameterization largely improved the match between observed and simulated fishing efforts in HNLC regions (see Galbraith et al., 2019). Yet we acknowledge that our approach is very simple and meant to improve model biases with a very simple biogeochemically-based parameterization. This will need to be re-evaluated and refined, especially as we improve Fe understanding and representation in models.

We included the following sentences to clarify these points:

L213-216: "*When satellite-based observational estimates of primary production are used as forcings, BOATSv1 overestimates fishing effort in HNLC regions, likely by simulating excessive biomass. Evidence of fish lack of adaptation to low iron regions suggests that low iron availability also significantly limits fish growth and could contribute to reducing fish abundance in large portions of the High Seas (Galbraith et al., 2019).*"

Including iron limitation of fish growth ultimately contributes to improve the representation of the High Seas catch variability in BOATSv2:

L469-471 "*This improvement in the High Seas is partly explained by the representation of iron limitation on fish growth ($r_{HSE90s}^{SAU}$ increases to 0.81 from 0.22 in v1), while along coastal regions, iron limitation alone is insufficient to explain catch (Table 4).*"

**L174: (NO-3 , in μM) is considered an indicator of iron limitation. Why is?**

**Changed.** Iron is significantly more difficult to measure than nitrate, and has a complex cycle. To date, accurate maps of Fe limitation in the surface ocean that could be used towards a parameterization do not exist. Instead, we rely on the long-standing biogeochemical observation that, under Fe-limitation of phytoplankton, macronutrients such as nitrate accumulate in the surface ocean (e.g., Moore et al., 2013). Therefore, observed surface nitrate maps can be used as a simple proxy to indicate limitation by other micronutrients, and experiments have shown that iron is generally the limiting factor in these high-nutrient, low-chlorophyll regions (Moore et al., 2013). We clarified the sentence:

L220-222: "*...is taken as a proxy for iron limitation (Moore et al. 2013) and as an indicator of regions of growth limitation given the absence of other reliable globally resolved estimates of surface iron concentrations or plankton iron contents.*"

**L199 and L278: What thickness is used to estimate Tbot? Does the thickness vary with bottom depth?**

**Changed.** The bottom temperature is estimated from the temperature values of the deepest depth layers. Given that these layers are not of homogeneous thickness in observational datasets such as the WOA and models, they are instead the closest temperature value found in the vicinity of the seafloor. We clarified:

L331-333: "*Recognizing that the resolution of observational temperature datasets such as the WOA decreases with depth, we select the layers closest to the bottom as indicative of the temperature near the seafloor.*"

**L208: How did you assume -0.8?**

**Changed.** This parameter is uncertain since it can vary heterogeneously in space; best estimates indicate the range -[0.7,0.98] (Gloege et al. 2017). We selected -0.8 in this range:

L256-258: "*The attenuation coefficient $b_a$=-0.8 is selected within the range of plausible values (Gloege et al. 2017), and the euphotic layer depth $z_{eu}$=75m is assumed to be fixed, although both could be modeled to vary in space and time.*"

**L221: Specify the activation energy of "growth and mortality" to help identify that these are 2 parameters.**

**Changed.** We clarified this L270.

**L231: When fisheries target demersal species, do we agree that cost increases with distance from shore and depth? If so, I don't find this clear in the equation for the demersal community. If not, why not?**

**Changed.** We agree, but given that we were unsure if both would apply or if instead one would be dominant over the other, we tested them separately first, hence the description of the processes independently. We clarified:

L294-296: "*Given the uncertainty over whether distance or seafloor depth have a greater impact on costs in pelagic and demersal fisheries, we first tested distance and depth-dependent costs separately, and then added them to determine their combined impact.*"

**L322: 12 is not fixed in Table 1. Is this a reason not to recalibrate the trophic scale?**

**Changed.** The trophic scaling is just a function of 2 of the other 11 free parameters (trophic efficiency and predator-to-prey mass ratio), so it is determined from the value of those two model parameters. We just report it because it is an important quantity that is useful to interpret the sensitivity of the model.

L380: "*(Note that the trophic scaling is a function of two free parameters, and thus is completely determined by their values.)*"

**L324: (h ζ1) becomes (h and ζ1),**

**Corrected**, thanks

**L355: "5 parameters (6 including the trophic scale)". Why do you make a distinction here?**

See previous comment. Note that this sentence has been removed to address a comment of reviewer 2 and improve the readability of the manuscript.

**L386: Figure 3: Acceptable range in addition to mean harvest could be useful ([70,150]x10^6) + Why is there an overall overestimation of pelagics?**

**Changed.** We added the range of acceptable catches and ratios in Figure 3.

The overall estimation of pelagics can reflect the lower food input at the base of the demersal food web (from the flux of detritus reaching the bottom) that the parameterization must compensate for to allow the accumulation of comparable pelagic and demersal biomass. But our main point is that among all simulations, a subset is able to reproduce observations, and these actually also show good LME-level coefficients of determination.

**Figure 4c: why the change in variability over time?**

This variability change must indirectly relate to our selection method of best ensembles, especially the criteria that selected ensembles reproduce the observed peak catch in the 1990s. Given the dynamic of BOATS, this catch maximum occurs when new fish biomass production cannot compensate for the development of fisheries anymore, and ecosystems

start being depleted beyond a maximum sustainable yield, after which catch declines. This explains the spread of biomass at a pristine state (~1900s) that is reduced as ecosystems are exploited to reach more comparable values around the catch peak of the 1990s.

**Table 3: why does v2-Bio\* & Πβ seem to be the best model? \*v2-Bio = v1 + αcorr + (Πψ : delete parenthesis**

**Changed.** The model v2-Bio\* & Πβ is better in many aspects, especially in capturing characteristics of LME-level catch. However, it leads to too much high-seas catch and poorer values for $R_C$ and $Z_C$, a bias that we attempted to correct with BOATSv2 (see also Fig 5a,b, compare the green "v2-Bio\* & Πβ" and red lines "v2"). Economic parameterizations allow improvement for $R_C$ and $Z_C$ that we keep for the definition of the BOATSv2 version. Note that the description of the economic parameter selection has been reformulated (see Section 5.1.2, L472-510) to strengthen the main message. We now stress that (1) economic parameters have little influence on the fidelity of simulations to observation along the coast, (2) they mainly improve global indicators, i.e. High Seas catch fraction and mean fishing depth, with modulations between pelagic and demersal. This helps us pick the best combination of spatial cost and catchability parameterizations kept for BOATSv2.

**L389: These Australian LMEs: are they deep or iron-limited zones?**

**Changed.** Based on ETOPO, Australian LMEs include relatively large continental shelves that would qualify as shallow and based on the map of sea surface nitrate concentration (see Supplementary Material Fig. B1), they do not seem to be significantly iron-limited. Here, our point is not about our simulations but instead about catch observations. SAU vs. Watson et al. catch reconstructions provide different estimates for these particular LMEs; the latter leading to a better match with our simulations. However, the assumptions that lead to significant differences in the reconstructed catch in Australian LMEs in data from Watson et al. vs SAU remain unclear. We clarified:

L480-483: "*Note that the comparison reveals better correlations when comparing models with WAT catch reconstructions instead of SAU reconstructions. Most of the improvement is explained by higher mean catches in Australian LMEs (compare Fig. 5c and d), but the explanation for such discrepancy in the observational reconstructions remains unclear.*"

**L445: How can mortality be negative? Perhaps more information on this term in the additional parameters table might help to understand where it comes from.**

**Changed.** Sorry for the confusion. This parameter is not a mortality constant but a parameter that contributes to the mortality constant in exponential form (exp(-0.072) = 0.9305), along with a temperature-dependent and growth-dependent contribution (see equation 4). It follows the definition of mortality in Gislason et al. 2010. We stressed

throughout the manuscript that $\zeta_1$ is a parameter in the definition of the mortality constant, and added the sentence:

L520-521: "*Note that the negative value for $\zeta_1$ does not indicate a negative mortality, since $\Lambda_k \propto e^{\zeta_1}$, Eq. (4).*"

**Figure 5a: How do you explain the increasing trend in the model for the 2 versions, which is not observed in the data? Is it linked to the exponential response to temperature? If so, it could be interesting to discuss what other temperature responses could have been used, and how they might impact the model.**

**Changed.** We don't think a changing environment drives this trend – since all versions include the same exponential response to temperature. Instead, this is linked to the exponential increase of catchability that drives the development of fisheries in the model. Here, we show that we can strongly correct the ratio $R_C$ by delaying the development of fishing in the High Seas or reducing the exploitable biomass in the High Seas. However, this correction is still imperfect, especially regarding the continuously increasing trend beyond the 1990s. Future tests that could help resolve this discrepancy are:
1. Consider changes in the rate of catchability increase.
2. Address additional processes that further influence the relative distribution of biomass between high seas and coastal regions. A main process to consider is the migration of fish stocks between inshore and offshore regions, which redistributes biomass.

We included a comment on this and plan to address these in the future:

L668-672: "*However, as fisheries keep developing, BOATSv2 still overestimates fishing in the High Seas (compare red dashed line with observations in Fig. 13, or the increasing trend in simulations in Fig. 5a). This discrepancy suggests either an improper representation of the historical rate of catchability increase in the simulations, or missing mechanisms, such as horizontal migrations that redistribute biomass from the High Seas to the coast.*"

**L453: "Although the temperature dependence of mortality ($\omega_{a,\lambda}$) is not significantly different from the initial values, the optimized values suggest a stronger sensitivity of growth compared to mortality for the pelagic community ($\omega_{a,A} - \omega_{a,\lambda}$ = +0.047 eV), and a stronger sensitivity of mortality for the demersal community (−0.082 eV).". Is it supported by experimental studies of fish thermal responses?**

While we are aware of analyses pointing toward a lower temperature dependence of growth for demersal communities compared to pelagic communities (van Denderen et al. 2020), we are unaware of analyses addressing this difference with mortality between communities. Further study is required to verify this emergent sensitivity in BOATS.

**L464: "indirectly allows larger asymptotic sizes (m∞) that are exposed to greater natural mortality; however, since m∞ is fixed". There's a contraction in that sentence, isn't there? If not, it needs to be explained differently.**

**Changed.** Here, we were describing the sensitivity of the mortality rate to parameters (see eq. 4), but the sentence was indeed unclear. We now report the relationship between parameters without further analysis of plausible mechanisms:

L537-539: "*Conversely, the mortality parameter $\zeta_1$ decreases when the growth scaling exponent (b) increases (Fig. 6d, r = -0.49), instead of decreasing, because of indirect impacts on the asymptotic size ($m_\infty$).*"

**L488: For ecological or economic reasons?**

**Changed.** It is unclear what dominates, but given that the biases are already visible before modifying the spatial parameterization of economic processes, we suggest an ecological cause (see Appendix G). We adapted accordingly:

L562-564: "*These biases are not improved by the economic update, and are likely related to ecological factors (see Appendix G panels b vs. c). However, it remains unclear if biases could also result from historical interactions between ecosystems and fishing effort, or from changing environmental conditions.*"

**L495: Can you add a hypothesis about the reasons?**

**Changed.** We added suggestions on what might help reduce these biases. However, a clear answer will only be possible after a robust comparison of the improvements that each new process could allow. We plan to address this in future model developments. We also added comments regarding the biomass under/overestimations (cf Fig 12).

L573-578: "*It is possible that accounting for features of coastal habitats such as coral reefs and mangrove forests could reduce regional biases, especially in South East Asia (Tittensor et al. 2010). Representation of biodiversity also remains crude, and additional functional types with life histories that differ from those of fish, such as cephalopods, could be considered (Denechere et al. 2024). Finally, some larger predators that dive to feed on the deep scattering layer experience environmental conditions that differ from those at the surface (Nuno et al. 2022, Braun et al. 2023). Accounting for this effect could help reducing model biases.*"

L648-653: "*Alternatively, considering biodiversity could help explain these differences. For instance, in NA-W, the dominance of semi-pelagic Alaska pollock may lead to an underestimation of our exclusively demersal biomass. Conversely, in NA-E, shifts from demersal to pelagic communities due to fishing can explain the overestimation of demersal*

*biomass (Choi et al. 2004). Our approach does not capture these interactions between pelagic and demersal communities.*"

**L540: What are the expectations in terms of the impact on carbon sequestration?**

**Changed.** We haven't completed this analysis accurately as this requires additional computations beyond the scope of the present study. However, given that at peak catch, the relative biomass distribution is the same between versions and the total biomass of pelagic fish in BOATSv2 is largely comparable with biomass estimates in BOATSv1, we would expect comparable carbon sequestration at peak catch between versions. Further analysis will be required, in particular to differentiate effects from pelagic and demersal communities. We included a sentence on this:

L625-628: "*Comparing BOATSv1 and BOATSv2, the similar relative biomass distribution at peak harvest, and the similar magnitude of pelagic biomass would suggest comparable estimates of export and sequestration by sinking fecal pellets (Bianchi et al. 2021). However, further analyses is needed to differentiate the roles of pelagic and demersal communities and their historical depletion in carbon and nutrient sequestration (Cavan et al. 2022).*"

**Table A1: n_k = 3. Why is this so? Is the model sensitive to this parameter?**

This subdivision is a crude representation of biodiversity devised to best fit the binning of observed catch by SAU and WAT, meaning three groups: a small group representing species of asymptotic sizes smaller than 30cm L<30cm; a medium group representing species of asymptotic sizes smaller than 90cm 30<L<90cm; and a large group L>90cm. Each group corresponds to the maximum weights of 0.3, 8.5, and 100kg (see Carozza et al. 2016), and our tuning procedure is devised to capture this biodiversity coarsely in a catch (Carozza et al. 2017 and see item 4 section 4.2). The model might be sensitive to this parameter; however, tuning the model to a different set of groups would also require finer biodiversity observations and is beyond the scope of this study.

**Table A2: Temperature units. K or °C. Why use the 2 units?**

**Changed.** These two units acknowledge differences between how the forcing is provided (in Celsius), how temperature is applied to estimate the fraction of large phytoplankton (in Celsius), and how it is used in the model through the Arrhenius relationship (in Kelvin). We added more details on the different temperature units to be more transparent.

**References:**

Barnes, C., Maxwell, D., Reuman, D.C. and Jennings, S., 2010. Global patterns in predator–prey size relationships reveal size dependency of trophic transfer efficiency. *Ecology*, *91*(1), pp.222-232.

Braun, C.D., Della Penna, A., Arostegui, M.C., Afonso, P., Berumen, M.L., Block, B.A., Brown, C.A., Fontes, J., Furtado, M., Gallagher, A.J. and Gaube, P., 2023. Linking vertical movements of large pelagic predators with distribution patterns of biomass in the open ocean. *Proceedings of the National Academy of Sciences*, *120*(47), p.e2306357120.

Carozza, D.A., Bianchi, D. and Galbraith, E.D., 2016. The ecological module of BOATS-1.0: a bioenergetically constrained model of marine upper trophic levels suitable for studies of fisheries and ocean biogeochemistry. *Geoscientific Model Development*, *9*(4), pp.1545-1565.

van Denderen, D., Gislason, H., van den Heuvel, J. and Andersen, K.H., 2020. Global analysis of fish growth rates shows weaker responses to temperature than metabolic predictions. *Global Ecology and Biogeography*, *29*(12), pp.2203-2213.

Galbraith, E.D., Le Mézo, P., Solanes Hernandez, G., Bianchi, D. and Kroodsma, D., 2019. Growth limitation of marine fish by low iron availability in the open ocean. *Frontiers in marine science*, *6*, p.439120.

Gislason, H., Daan, N., Rice, J.C. and Pope, J.G., 2010. Size, growth, temperature and the natural mortality of marine fish. *Fish and Fisheries*, *11*(2), pp.149-158.

Gloege, L., McKinley, G.A., Mouw, C.B. and Ciochetto, A.B., 2017. Global evaluation of particulate organic carbon flux parameterizations and implications for atmospheric pCO2. *Global Biogeochemical Cycles*, *31*(7), pp.1192-1215.

Hansen, B., Bjornsen, P.K. and Hansen, P.J., 1994. The size ratio between planktonic predators and their prey. *Limnology and oceanography*, *39*(2), pp.395-403.

Hatton, I.A., Heneghan, R.F., Bar-On, Y.M. and Galbraith, E.D., 2021. The global ocean size spectrum from bacteria to whales. *Science advances*, *7*(46), p.eabh3732.

Moore, C.M., Mills, M.M., Arrigo, K.R., Berman-Frank, I., Bopp, L., Boyd, P.W., Galbraith, E.D., Geider, R.J., Guieu, C., Jaccard, S.L. and Jickells, T.D., 2013. Processes and patterns of oceanic nutrient limitation. *Nature geoscience*, *6*(9), pp.701-710.

Nuno, A., Guiet, J., Baranek, B. and Bianchi, D., 2022. Patterns and drivers of the diving behavior of large pelagic predators. *bioRxiv*, pp.2022-12.

Scherrer, K. and Galbraith, E., 2020. Regulation strength and technology creep play key roles in global long-term projections of wild capture fisheries. *ICES Journal of Marine Science*, 77(7-8), pp.2518-2528.

The paper "BOATSv2: New ecological and economic features improve simulations of High Seas catch and effort" describes a set of significant advances in the BiOeconomic mArine Trophic Size-spectrum (BOATS) model. The main improvements are that BOATSv2 now resolves a distinct benthic pathway for delivering energy to demersal and benthic fisheries and includes spatially variable fishing costs and catchability. They also integrate iron-dependent fish growth rates and fishing effort targets that were developed in previous work. After re-calibrating the model, BOATSv2 performed similarly for coastal systems showed marked improvement in its representation of high seas fisheries catch, which was significantly over-estimated by BOATSv1. The addition of the benthic pathway was the primary driver of this improvement, with spatially variable fishing costs and catchability providing secondary yet notable further improvement.

I found the BOATSv2 improvements documented in this paper and the resulting improvements they yielded in high seas catch to be significant. The approaches for parameterizing the array of ecosystem, fishing and economic factors should be of general interest to the community, as should the approach to evaluation and optimization. I did, however, find some aspects of the presentation challenging. While I was generally convinced that the approaches were reasonable, there were cases where limited and/or gaps in the model description left me with questions. I also felt that the organization could be improved. My two main comments are thus:

**1) The presentation of BOATSv1 in Section 2.1 needs improvement. I understand that the authors don't want to spend too much time reviewing previously published model dynamics but the reader still needs to understand the BOATSv1 foundation to follow the rest of the analysis herein. My advice is to give yourself an extra ~50 lines or so and put yourselves in the shoes of someone who has not read the Carozza article. Clearly define all of the parameters and quantities you mention and provide enough narrative to give the reader a quantitative and qualitative understanding of the model dynamics. I have tried to provide specific suggestions below that I hope are useful.**

**Changed**. We significantly reformulated and expanded Section 2 about the description of BOATSv1 (+35 lines while we also moved parts of the description to Section 4). We also added a table (now Table 1) summarizing the main parameters and quantities of the model discussed in the main manuscript. We added details to Fig 1 and added references to the figure within the text to illustrate the contribution of different parts of the model. See following our answers to your specific comments for more details, and the highlighted manuscript.

**2) While the conclusions were ultimately clear, the presentation of Results needs improvement. For example, when I reached the Results section (section 5) I was**

surprised because the primary results of the study (e.g., the improvement in fidelity with high seas fish catch) had just been presented in Section 4. I also thought the results may contain too many detours and details that risked blunting the main conclusion. Please consider describing the optimization methodology in the methods, but moving the optimization results to the results section. The intermingling of methods and results may have contributed to my sense that there were too many detours and details, but a final round of editing for brevity and focus would be beneficial. Page 18-19, for example, had a lot of material that, while interesting, was secondary to the primary messages. Again, I have tried to provide specific suggestions for your consideration below.

**Changed**. Thank you, we strived to modify the manuscript in several ways to improve the clarity following your suggestions:

1- The model parameterization and optimization results are now split between a section describing the optimization method (Section 4: *Parameterization procedure*), and the results of the optimization are now detailed separately in the results section (Section 5.1). We also updated Fig 2 to better reflect the steps of the parameterization procedure.

2- Paragraphs describing parameter uncertainties in the model description sections (Sections 2 and 3) have been moved to Section 4 ("*Parameterization procedure")*.

3- We have rewritten parts of the description of the results for the optimization to be clearer.

See our answer to specific comments for more details.

Overall, I think this is a substantive paper that documents meaningful model advances and skill improvements that will be of general interest to the modeling community and serve as a valuable reference for future BOATS applications. I also think, however, it would benefit from a final round of edits for clarity and brevity to ensure that it has the impact it should. Please view my comments in that constructive spirit.

Specific comments:

**Abstract: Clarify that the novel features described starting on line 7 are novel to BOATS but not necessarily novel in the field. You need to define large marine ecosystems for the uninitiated. Also, there seems to be a wonderful opportunity to**

**state the factors that were responsible for the model improvements after line 13. Please take it!**

**Changed.** We specified that these are "*Features added to BOATS here for the first time…*" (L7), clarified that large marine ecosystems are "(*66 commonly adopted coastal ocean ecoregions)*" (L10, see also next comment), and added a sentence to be more specific about the factors responsible for model improvement:

L14-15 "*Improvements mainly stem from separating pelagic and demersal energy pathways, complemented by spatially variable catchability of pelagic fish and depth- and distance-dependent fishing costs.*"

**Line 41: Define LMEs and include a reference so that readers understand this definition.**

**Changed.** We added a sentence and reference:

L46-47: "*Note that, Large Marine Ecosystems (LMEs) are 66 coastal ocean regions defined by ecological criteria (Sherman and Duda, 1999).*"

**Section 2: As described in general comment 1, I found this section challenging.  I would allow yourself more space to present these core dynamics clearly. I have provided a few specific suggestions to improve it that I hope are useful:**

**Changed.** We have significantly reformulated Section 2 and added about 35 lines to communicate the rationale of BOATSv1. Please see the updated section 2 (L74-202).

**Table 1 currently includes only a limited subset of the parameters and quantities discussed in this section and a lot of detail on optimization procedures that aren't described until much later.  The parameter/quantity definitions are what the reader needs now.  Please provide them for all the parameters/quantities discussed in this section and save the additional details of the optimization for when the reader needs them.**

**Changed.** Table 1 now includes a list of the parameters and quantities discussed in the model description (Sections 2 and 3). Former Table 1 (now Table 2) has been moved where we address the parameterization procedure (Section 4).

**I didn't understand why you chose to include the growth/recruitment function for the smallest size class with the MvF equation on line 80.  The general growth expression is given later and the most natural place to deal with the recruitment function seems to be around line 100 in the current text.  I suggest you start with MvF and then unpack the growth functions as they arise in the text in a consistent manner.**

**Changed.** Our presentation of Eq. (1) was meant to summarize the biomass conservation equation along with its initial and boundary conditions, as is often done when presenting ordinary differential equations, where these conditions are integral parts of the solution. To avoid any confusion, we have now splatted the differential equation (L90) and the boundaries (L98), we have also added more details:

L95-97: "*This first-order partial differential equation in time and size, requires both a boundary condition, here prescribed at the smallest size class $m_0$ and representing recruitment, and an initial condition at t=0, representing the initial biomass distribution for each group*"

Note that also provide more details and lay out the main principles of the model based on this system of equations. Growth, mortality and recruitment are further discussed in the following paragraphs. We also added more details on the definition of recruitment.

L91-94: "*The first term in Eq. (1) represents the rate of change in time of the fish biomass spectrum for each group. The second term is the divergence of the growth flux, i.e., the transfer of biomass to increasing size as fish grow. The third term encapsulates the biomass accumulation due to the increase of individual size as fish grow. The fourth and fifth terms represent losses from natural mortality and catch respectively.*"

Please refer to the manuscript for all the modifications of Section 2.

**Equation (2): should the (1-Phik) be carried over to the right-most quantity?**

**Changed.** Thanks for noticing this mistake, we corrected it (L115).

**Line 90-99: Improved representation of the energy flow between phytoplankton and fish ends up being one of the major required improvements between BOATSv1 and BOATSv2.  Fully understanding this important change requires a clearer description of the BOATSv1 parameterization.  The productivity symbols need clearer definitions, the shape and rationale for the energy spectrum and the trophic scaling need to be more clearly described.  I could not find any description of how the characteristic size of the phytoplankton was determined.  It is unclear which groups NPP is being partitioned across and why assuming that it is even is sensible (e.g., if one group has much higher biomass than another, wouldn't it make sense for more NPP to go to the one with higher biomass?).  Please expand this section as needed so that the reader understands how growth and energy flow constraints were handled in COBALTv1 so they can fully understand how these change in COBALTv2.**

**Changed.** We improved the explanation of growth and the main model components (see Section 2):

L109-114: "*In BOATS, the growth rate at a given size occurs either at the maximum physiological rate when food is not limiting (gray area in the central panel Fig. 1) or proportionally to primary production $\Pi_\psi$ when food is limiting (green area in the central panel Fig.1). Accordingly, the growth rate is proportional to the minimum of two quantities: (1) the energy provided by primary producers that reaches a given size class, given trophic transfer across the food-web $\xi_{P,k}$ ,divided by the number of fish in that size class, and (2) the maximum production potential for a fish of that size, based on an individual-level allometric growth rate that follows a von Bertalanffy formulation $\xi_{VB,k}$ (in $g\ s^{-1}$):*"

Regarding the determination of the characteristic size of phytoplankton, we clarified that it is based on an empirical model published in Dunne et al. 2005:

L122-123: "*The representative size $m_\psi$ is determined from the empirical phytoplankton size structure model of Dunne et al. (2005) and depends on temperature (T in °C) and primary production $\Pi_\psi$.*"

L266: "*We use an empirical phytoplankton size model to account for this variation (Dunne et al. 2005).*"

Finally, about the partition of primary production, we added the details:

L123-126: "*To ensure coexisting fish groups, and because of the scarcity of data available to constrain resource allocation, primary production is equally partitioned across the groups, i.e., $\varphi_{C,k} = 1/n_k = 1/3$ . While this is a first-order assumption that allows realistic simulation of catches by group, it should be revised as new observational constraints become available.*"

**Line 111-119: Would this paragraph be better saved for a discussion of the optimization procedure?**

Changed. This paragraph about the undetermined parameters of the model has been reformulated and moved to Section 4.2 (L381-390).

**Line 112: Are these natural logs or base 10?**

Changed. These are indeed natural logs. For clarification, we replaced mentions of natural logs with "*ln*".

**Line 135-138: It is difficult to understand what is being described here without seeing the relationship and where the parameters sit within it.**

Changed. To clarify the description of the fishing selectivity we now include the equation in the manuscript (Eq. 8) and clarified its description (L175-182 in Section 2).

**Please ensure that each parameter in eqs. (5) and (6) is clearly defined.**

**Changed.** In order to be clearer, we added a table with the list of all parameters and quantities of the model discussed in the manuscript (Tab. 1). We also added the equation of the fishing selectivity function (see previous comment) and included numerical values for the price and costs:

L195-196: "*In BOATSv1, the ex-vessel fish price $p_k$ is generally assumed to be constant in space and time (1.264 $10^{-3}$ \$ $g^{-1}$), since observations suggest small historical variations (Sumaila et al., 2007; Galbraith et al., 2017). Similarly, cost $c_k$ is also assumed constant (1.852 $10^{-7}$ \$$W^{-1}$ $s^{-1}$).*"

**Figure 1: Please expand this caption so that the reader understands what is plotted. It also seems like colorbars are needed in several places.**

**Changed.** We expanded the caption and provided more details:

L91: "*Schematic diagram of the main modules, components, and processes of BOATSv2. Environmental forcings, shown in the left panel ("pelagic" for BOATSv1; "pelagic" and "demersal" for BOATSv2), drive an ecological module that solves for the evolution in time of fish biomass as a function of fish size, for multiple groups with different maximum size, shown in the central panel. These fish biomass spectra interact with the dynamic of fishing, controlled by an economic module and economic forcings, shown in the right panel. Economic forcings are spatially uniform in BOATSv1, but can be spatially variable in BOATSv2. Environmental forcings include the spatial distribution of resources at low trophic levels ($\Pi_\psi$ or $\Pi_\beta$) and representative habitat temperatures ($T_{75}$ or $T_{bot}$). Fish biomass spectra for multiple groups emerge from the balance of environmentally controlled growth ($\Gamma$, linked with $\xi_P$ or $\xi_{VB}$), recruitment (R), natural mortality ($\Lambda$), and fishing mortality (H). Economic forcings, which include spatially uniform ex-vessel prices (p) and spatially variable fishing costs (c) and catchability (q), influence the dynamic of fishing effort (E) for each fish group. Color shadings of forcings illustrate spatial variations, from low (light) to high (dark) values. This figure is updated from the schematic for BOATSv1 in Carozza et al. (2017).*"

Note that we prefer not including colorbars for simplicity of the schematic. Instead, we specified:

L91: "*Color shadings of forcings illustrate spatial variations, from low (light) to high (dark) values.*"

**Line 164, Section 3.2: "Novel" is a tricky word to use. Do you mean new features relative to prior versions of BOATS? Be more precise with what you mean here.**

**Changed.** Here we refer to the newly added features that added to BOATSv1, which along with *previously-published features* (Section 3.1) constitute BOATSv2.

We retitled L240: "*Newly added features*"

**Eq. (8): What does the superscript "corr" indicate? This occurs later in the text as well in association with other quantities, but I was never completely sure what it was meant to indicate.**

**Changed.** The superscript "*corr*" indicated corrected quantities and parameters compared to BOATSv1. We changed to "*v2*" and included a clarification:

L222-223: "*Note that here and in the following sections, the superscript "v2" indicates corrected quantities compared to the initial formulation in BOATSv1.*"

**Eq. (10): Are the costs additive?**

**Changed.** Yes. Initially, we were unsure if both would apply, or if instead one would be dominant over the other, so we tested them separately first, hence the description of the processes independently. Then they were added. We now clarify:

L294-296: "*Given the uncertainty over whether distance or seafloor depth have a greater impact on costs in pelagic and demersal fisheries, we first tested distance and depth-dependent costs separately, and then added them to determine their combined impact.*"

**Line 215-217: How was the size of the phytoplankton set? I don't think this was ever mentioned in Section 2.**

See previous comment, where we clarified the approach and references.

**Line 235: Was there a rationale for choosing 370 km?**

**Changed.** It accounts for 200nm, the width of Exclusive Economic Zones, meant to separate coastal and High Seas according to the United Nation Convention on Law of the Sea. We clarified:

L285: "*Here, we adopt x\* = 370km (or 200nm), the limit of Exclusive Economic Zones separating coastal regions and High Seas.*"

**Figure 2: As in figure 1, please expand the caption to make the meaning of this figure clear. It gives the impression, for example, that the selection of 11 parameters comes from BOATSv1. I don't believe this is the case.**

**Changed.** We have updated the figure as well as expanded the caption to be clearer:

L347: "*Schematic diagram of the parameterization procedure starting from BOATSv1 (in blue), with two steps. (1) Ecological update (in green): a Monte Carlo tuning procedure with 5 selection criteria is applied on a modified version of BOATSv1 that allows separate pelagic and demersal pathways and growth limitation in iron-limited regions. An ensemble of 20000 simulations is carried out only for coastal regions with various parameter sets, and*

*we identify a set of 5 (or 10 extended) best parameter sets. (2) Economic update (in red): with 3 selection criteria, we identify the best economic parameterizations applied on the optimized intermediate BOATSv2-Bio version to determine BOATSv2. We use simulations that include the High Seas for 5 best parameter sets. Observations used for the parameterization of both are shown in gray."*

**Line 359-377: The description here related to the calibration results and the performance of the calibrated model against observed patterns seems like a Result (see general comment 3). Line 376-377, for example, reveals perhaps the most prominent result – the improvement in high seas catch. I would consider describing the calibration procedure in the methods and moving the Results to the Results section where they can be concisely presented after digesting the methodology.**

**Changed.** As previously detailed, these calibration results are now in the results section, in the "Parameterization" subsection under the "*Economic parameters*" subsection:

L70 "Section 4 describes a revised model optimization procedure. *Section 5 justifies the selection of an ensemble of 5 optimal parameters, compares the old and new model versions, highlighting improvements in the representation of global fisheries in BOATSv2, and discusses insights from the new formulation."*

**Section 4.3 also suffers a bit from this mix of Methods and Results. Also, following general comment 2, there are many details and detours on pages 18-19. While each may be interesting, there is a risk of pulling attention away from the main messages. Part of this may be addressed with a clearer separation of Methods and Results, but I would also encourage the authors to think carefully about how to present the results they feel are most critical as concisely as possible. There also seem to be a number of small discrepancies between values listed in the text and those in Table 3. Please ensure that these are synchronized.**

**Changed.** We split this section into a subsection under "Parameterization procedure" named "Economic update: sensitivity to cost and catchability", where we detail the method, and a subsection in results named "Economic parameters" where we present the results of the parameterization (see respectively sections 4.3, L420-440, and 5.1.2, L472-511). Note that we also have partly reformulated the description of the selection of optimized economic parameters (now in section 5.1.2) to strengthen the main message. We now stress that (1) economic parameters have little influence on the fidelity of simulations to observation along the coast, (2) they mainly improve global indicators, i.e. High Seas catch fraction and mean fishing depth, with modulations between pelagic and demersal. This helps us pick the best combination of spatial cost and catchability parameterizations kept for BOATSv2.

**Line 393-394: I'm not sure what you mean by: "This suggests that analogous parameterizations of heterogeneous costs and catchability will generate comparable variability in LME catches."**

**Line 395: Assuming this is relative to Watson, should BOATSv2 be 0.64?**

**Changed.** Regarding the two previous comments. We apologize these lines were unclear. This was partially caused by the columns of the table being improperly labeled. This has now been fixed. The values and coefficients of correlation reported (for max catch) are correct (i.e. 0.46 for pelagic catch, and 0.69 for demersal catch). These correlations for maximum catch, and the fact that they are independent of cost or catchability profile selected, indicates that these latter variables are not impacting the maximum catch. Instead, maximum catch is controlled by ecosystem features. However, cost and catchability influence the timing of the development of fisheries. We reformulated the unclear sentence.

L484-487: "*Heterogeneous costs or catchability show no effect on the variability of maximum pelagic and demersal catch yields, $r^{SAUP}_{LMEmax}$=0.46 vs. $r^{SAUD}_{LMEmax}$=0.69 (Table 4); these should instead influence the timing of the development of fisheries. Both correlations suggest that, along the coast, catches are independent of economic parameterizations and are instead controlled mainly by the environment.*"

**Line 404-405: The effect of heterogenous costs seems quite small on the fraction of high seas catch. I found this surprising, and I could not find this result in Table 3. It looks like the high seas fraction with heterogenous costs is ~0.14, not 0.3?**

Yes, heterogenous costs have a relatively small effect on the fraction of High Sea catch, compared to the effect of reduced biomass production when resolving separate pelagic and demersal pathways. This is illustrated by the Table 4 (formerly Table 3) column reporting $R_{C90s}$. Note that the ~0.3 value was referring to our discussion of the fraction of High Seas pelagic catch (Table 4) that are insufficiently corrected with just heterogeneous cost profiles, suggesting that cost alone is not a dominant mechanism. This secondary detail has been removed to simplify the presentation of the results (see updated Section 5.1.2, and previous comment).

**Results and Discussion section: This seems out of place. Haven't most of the primary results have already been revealed in the prior section (see general comment 3).**

**Changed.** The "*Results and discussion*" section now includes the results of the parameterization (Section 5.1) before more focused discussion of various features of catches, effort and biomass. We have relabeled the subsections (Sections 5.2-4) accordingly "*Features of simulated catch/fishing effort/biomass*".

**Line 445: The difference in the mortality constant seems to merit some additional discussion. How should the reader interpret this ecologically? Perhaps you could close the loop on this issue on line 464 where compensating effects are discussed? Finally, you may want to clarify that a negative value of this parameter, which I understand is an exponent, does not imply negative mortality?**

**Changed.** We realized the description of mortality was not clear and simply describe the relationship with (b) without further interpretation:

L538: "*Conversely, the mortality parameter $\zeta_1$ decreases when the growth scaling exponent (b) increases (Fig. 6d, r = -0.49), instead of decreasing, because of indirect impacts on the asymptotic size ($m_\infty$).*"

We also clarified the negative value:

L520: "*Note that the negative value for $\zeta_1$ does not indicate a negative mortality, since $\Lambda_k \propto e^{\zeta_1}$, Eq. (4).*"

**Line 467-469: Should the covariance of the pelagic and demersal temperature dependence be interpreted as an indicator that two parameters may not be needed (i.e., demersal and pelagic species exhibit the same response?). I was curious why you chose to allow a different temperature dependence for pelagic and demersal.**

**Changed.** Pelagic and demersal communities have covarying sensitivities of growth to temperature, however we think it is important to account for different rates of increase between the two communities. This is based on an observational meta-analysis from van Denderen et al. 2020 that shows that between the two communities temperature dependences are significantly smaller for demersal. Note that the sentence that explain this choice was confusing, we have reformulated and highlighted:

L269-271: "*We keep most food-web parameters the same for pelagic and demersal fish, with the exception of the activation energy for growth $\omega_{a,A}$ and mortality $\omega_{a,\lambda}$, since observations of growth rates suggest significant differences between the two communities (van Denderen et al. 2020).*"

L389-390: "*We keep the same parameters for both pelagic and demersal communities, except for the temperature dependence of growth and mortality*".

**Line 488-89: A reference to Patrick Lehodey's SEAPODYM work seems like it could be useful here?**

**Changed.** We added reference to SEAPODYM, especially linked with the representation of movement, along other references:

L564-566: "*Processes not included in the model, such as habitat alteration by bottom-trawling gears, additional constraints on habitats such as dissolved oxygen (Deutsch et al. 2020), fish migrations and movement (Lehodey et al. 2008, Watson et al. 2015, Guiet et al. 2022, Barrier et al. 2023), management and regulation, likely play a role in these biases.*"

**Line 502-504: Do you think that unresolved effects of hypoxia may play a role in the eastern tropical Pacific**

**Changed.** Indeed it is a possibility. However, here this is likely an indirect effect of our representation of the influence of iron limitation in HNLC regions. Presence of HNLC regions in the eastern pacific, as currently parameterized, can prevent accumulation of pelagic fish biomass, leaving demersal fish biomass, even very small, dominant. Further analysis and testing of other plausible processes is beyond the scope of the analysis for this BOATSv2 version, but the effect of oxygen on habitats should be more thoroughly assessed for next studies. We added the following:

L585-586: "The latter bias could reflect the parameterization of iron limitation, which reduces accumulation of pelagic biomass in the Eastern Topical Pacific, an HNLC region *(see Appendix B)."*

**References:**

Barrier, N., Lengaigne, M., Rault, J., Person, R., Ethé, C., Aumont, O. and Maury, O., 2023. Mechanisms underlying the epipelagic ecosystem response to ENSO in the equatorial Pacific ocean. *Progress in Oceanography*, *213*, p.103002.

van Denderen, D., Gislason, H., van den Heuvel, J. and Andersen, K.H., 2020. Global analysis of fish growth rates shows weaker responses to temperature than metabolic predictions. *Global Ecology and Biogeography*, *29*(12), pp.2203-2213.

Deutsch, C., Penn, J.L. and Seibel, B., 2020. Metabolic trait diversity shapes marine biogeography. *Nature*, *585*(7826), pp.557-562.

Guiet, J., Bianchi, D., Maury, O., Barrier, N. and Kessouri, F., 2022. Movement shapes the structure of fish communities along a cross-shore section in the California Current. *Frontiers in Marine Science*, *9*, p.785282.

Lehodey, P., Senina, I. and Murtugudde, R., 2008. A spatial ecosystem and populations dynamics model (SEAPODYM)–Modeling of tuna and tuna-like populations. *Progress in Oceanography*, *78*(4), pp.304-318.

Sherman, K. and Duda, A.M., 1999. Large marine ecosystems: an emerging paradigm for fishery sustainability. *Fisheries*, *24*(12), pp.15-26.

Watson, J.R., Stock, C.A. and Sarmiento, J.L., 2015. Exploring the role of movement in determining the global distribution of marine biomass using a coupled hydrodynamic–size-based ecosystem model. *Progress in Oceanography*, *138*, pp.521-532.